# Extensive variation between chromosomes of North American and European hop

Sandip Mallikarjun Kale [1,15,16], Heidrun Gundlach[2,16], Oliver Gericke [1,16], Nadia Kamal [2,3], Aldo Almeida [1], Nicholi Pitra [4], Nicholas Price[4], Georg Haberer [2], Thomas Lux [2], Flavia Krsticevic[1], Oliver Kemp [1], Louise de Bang [1], Axel Himmelbach [5], Sudharsan Padmarasu[5], Mark-Timothy Rabanus-Wallace [5], Lucie Horáková [6], Václav Bačovský[6], Kasper Nielsen [1], Nanna Bjarnholt [7,8], Nikola Micic [7,8], Isabella Kruse-Andersen [1,7], Birger Lindberg Møller [7,8], Christian Janfelt [9], Birgitte Skadhauge [1], Paul D. Matthews[4], Klaus F. X. Mayer[2,10], Nils Stein [5,11], Martin Mascher [5,12], Manuel Spannagl [2,13] ✉, Alexander Feiner [14] ✉ & Ilka Braumann [1] ✉

Hop is an essential ingredient in brewing, providing beer with its characteristic bitterness and aroma. Most modern hop cultivars are hybrids between European and North American hop lineages, but how these ancestries contribute to bitter acid content, the most important trait in hop breeding, remains unclear. Here, we report chromosome-scale, haplotype-resolved assemblies of the hybrid hop cultivar Apollo, assign European and North American ancestry across the genome, and identify varying levels of recombination suppression between chromosomes of either origin. Using this reference, we uncover genetic and chemical diversity in core bittering pathways between European and North American hops. We further show additive effects of beneficial European and North American alleles on bitter acid content, providing a foundation for genomics-assisted hop breeding.

Female plants of *Humulus spp.* L. (hop), member of the Cannabaceae family and sister to the genus *Cannabis*, have been cultivated for their cones (hops) since at least A.D. 859 when the practice of adding hops into beer developed in medieval Central Europe[1]. Hops contain α-acids that lend a characteristic bitter taste and, more importantly, made beer a durable food item suitable for transport and trade. After the mid-14th century, the demand for beer increased. It was preferred over water, as the latter was often polluted. Brewing became a professional craft and

[1]Carlsberg Research Laboratory, Copenhagen, Denmark. [2]Helmholtz Munich – German Research Center for Environmental Health, Plant Genome and Systems Biology (PGSB), Neuherberg, Germany. [3]Technical University of Munich, TUM School of Life Sciences, Computational Plant Biology, Freising, Germany. [4]Hopsteiner, S.S. Steiner, Inc, New York, NY, USA. [5]Leibniz Institute of Plant Genetics and Crop Plant Research (IPK) Gatersleben, Seeland, Germany. [6]Department of Plant Developmental Genetics, Institute of Biophysics of the Czech Academy of Sciences, Brno, Czechia. [7]Department of Plant and Environmental Sciences, Section for Plant Biochemistry, University of Copenhagen, Frederiksberg, Denmark. [8]Department of Plant and Environmental Sciences, Copenhagen Plant Science Center, University of Copenhagen, Frederiksberg, Denmark. [9]Department of Pharmacy, University of Copenhagen, Copenhagen, Denmark. [10]School of Life Sciences, Technical University Munich, Freising, Germany. [11]Institute of Agricultural and Nutritional Sciences, Martin Luther University Halle-Wittenberg, Halle, Germany. [12]German Centre for Integrative Biodiversity Research (iDiv) Halle-Jena-Leipzig, Leipzig, Germany. [13]Centre for Crop and Food Innovation, Food Futures Institute, Murdoch University, Perth, WA, Australia. [14]Hopsteiner, Simon H. Steiner, Hopfen, GmbH, Mainburg, Germany. [15]Present address: Department of Agroecology, Aarhus University, Slagelse, Denmark. [16]These authors contributed equally: Sandip Mallikarjun Kale, Heidrun Gundlach, Oliver Gericke. ✉e-mail: manuel.spannagl@helmholtz-munich.de; alexander.feiner@hopsteiner.de; ilka.braumann@carlsberg.com

subsequent municipal regulations like the Bavarian purity law (1516) limited beer ingredients to barley, water, and hops[2]. This shaped the conventional understanding of what is a beer and aided its rise to the most popular alcoholic beverage globally[3] with $555 billion gross value-added contribution to global Gross Domestic Product[4].

The utility of hops however extends well beyond their role as a bittering agent, as they offer a plethora of highly valuable metabolites with diverse functionalities. These include flavor- and aroma-contributing terpenes, natural preservatives (β-acids), and the anti-inflammatory and anti-cancer prenylflavonoid xanthohumol[5,6], along with its derivative 8-prenylnaringenin, the most potent phytoestrogen known[7]. Despite the obvious potential to broaden hop applications due to existing efficient fractionation technology, the absence of varieties with enhanced metabolite compositions for dual cropping limits hop use predominantly to beer production. These unmet breeding demands highlight the need for improved genetic resources to support hop improvement.

The hop originally cultivated for brewing purposes was *H. lupulus*, a species native to Europe and parts of Asia. Beyond *H. lupulus*, the genus *Humulus* comprises the five additional perennial species *H. cordifolius* (Japan), *H. neomexicanus* (south and western North America), *H. lupuloides* (central and eastern North America), *H. pubescens* (midwestern North America), and *H. yunnanensis* (southcentral China), as well as one annual species, *H. scandens* (syn. *H. japonicus*, East Asia)[8]. The Asian-North American and European hop lineages separated more than a million years ago, likely in China[9].

After the introduction of European *H. lupulus* to North America in the 17th century, spontaneous hybrid offspring of European (Eu) and native North American (NAm) hop, termed "Cluster", proved superior to Eu landraces in bittering compound content. Cluster varieties were widely cultivated and in the early 20th century imported into Europe[10]. With the establishment of the first scientific hop breeding program in the United Kingdom in 1904, breeders began to hybridize imported NAm with Eu germplasm. The first hybrid cultivars were released in 1934, 1939, and 1944[10]. More recent work shows that wild NAm hops contain higher α-acid levels than Eu hops. However, α-acid content of NAm-Eu hybrid cultivars exceeds that of wild hops of either origin by far[11]. Consequently, although traditional Eu landraces remain valued for their distinct and traditional aromas, most current cultivars are monoploid hybrids of Eu and NAm hop or their descendants. Still, current draft hop genome assemblies[12–14] leave the impact of this interspecific breeding strategy on genome structure largely imperceptible, because they lack phasing and haplotype resolution and miss information on species ancestry for regions associated with beneficial traits.

Assembling reference-quality, phased haplotype-resolved assemblies of complex plant genomes, such as the highly heterozygous dioecious hop genome (haploid genome size:~2.5 Gb, ♀ $2n = 18 + XX$, ♂ $2n = 18 + XY$), typically relies on long-range sequencing combined with high-throughput chromosome conformational capture (Hi-C)[15–17]. Partitioning of haplotypes by parental origin in interspecific hybrids can further be achieved through trio-binning[18], the use of specific parental k-mers to sort chromosomes or scaffolds, which has recently been adapted for plant genome assemblies[19–21]. Trio-binning, however, requires additional parental sequence datasets and only allows assignment of species origin in F1 but not in interbred hybrids such as most modern hop cultivars.

In this work, we generate chromosome-scale assemblies of both haplotypes of the hop cultivar (cv.) Apollo, a high α-acid producer derived from repetitive interbreeding of *H. lupulus* (Eu) and *H. lupuloides* (NAm) hybrids[22,23], and assign European (Eu) and North American (NAm) ancestry across haplotypes using k-mers isolated from species-specific repeat families within the Apollo genome. We identify varying levels of recombination suppression between chromosomes of either origin. We further show that beneficial Eu and NAm haplotypes have additive effects on the agronomically most relevant trait in hop, α-acid content.

## Results

### Tracing European and North American ancestry in the cv. Apollo genome

We used a binning-free approach relying on the hifiasm[24], ALLHiC[25], and TRITEX[26,27] pipelines to assemble the diploid genome of hop cv. Apollo from a total of 120 Gb circular consensus sequence (CCS) reads corresponding to 24-fold coverage of the diploid genome (Supplementary Figs. 1 and 2, Supplementary Table 1). First, we developed a haploid assembly using the hifiasm and TRITEX pipelines with manual editing to remove allelic contigs (Supplementary Tables 2 and 3, Supplementary Fig. 3). To facilitate intra- and interspecific genome comparisons through a common chromosome nomenclature across the Cannabaceae family[28], pseudomolecules were renumbered based on their collinearity with *Cannabis sativa* chromosomes[29] (Supplementary Fig. 4). Similar as in cereal genomes[30], we found the gene density to be increased towards the distal region of all pseudomolecules (Supplementary Fig. 5), except chr01, which harbors the 45S rDNA and corresponds to the sixth largest Apollo chromosome (Fig. 1a). Chr01 is submetacentric and lacks the signal for the *Humulus* subtelomeric repeat (HSR) tandem repeat array on the p-arm. This likely explains the observed deviation in gene density from that of the other Apollo chromosomes.

Subsequently, we generated a phased assembly of cv. Apollo using hifiasm in combination with ALLHiC and TRITEX, with Hi-C data and gene models from the haploid assembly guiding separation of contigs into distinct phases followed by the generation of a phased chromosome scale assembly (Supplementary Tables 4 and 5, Supplementary Fig. 6). Phase switch error estimation analysis identified only 2.085% (122,753) single nucleotide polymorphisms (SNPs) out of 5,886,045 SNPs showing phase switches indicating proper phase separation in the final assembly (Supplementary Table 6). The strong contacts between the neighboring contigs in the Hi-C contact matrix plot (Supplementary Fig. 7a) and the presence of ~98% complete Benchmarking Universal Single Copy Orthologs (BUSCO) in both phases (Supplementary Table 4) indicate high contiguity and completeness of the phased assembly.

At this stage, phase assignment in the assembly was still random. We therefore developed an approach to separate genomes of distinct parental origin in phased assemblies by leveraging subtle divergences in repetitive regions. Guided by the distinct chromosomal distributions of certain smaller repeat families, we identified differentially enriched high copy k-mer markers for two lineages (Supplementary Fig. 8), which were used for final haplotype partitioning (Supplementary Fig. 9). Mapping genotyping-by-sequencing (GBS) reads from selected feral and wild Eu and NAm *Humulus* genotypes (Supplementary Figs. 7b and 10) to the final assembly revealed that the two distinct lineages identified through the k-mer approach correspond to Eu and NAm ancestry. The cv. Apollo genome comprises one copy of either Eu or NAm origin for chromosome pairs chr01, chr03, chr04, chr05, chr06, chr09 and chrX. Both homologs of chr02 and chr07 are of Eu origin while both copies of chr08 are of NAm origin only. Further, one copy of both chr01 and chr08 were found to carry large introgressions of the other phase. A graphic representation of this haplotype-resolved phased assembly is shown in Fig. 1d. Homologous pseudomolecules with contrasting Eu and NAm phase assignment display substantial sequence divergence (~median identity 75–78%), which is consistent with the recognized species-level separation between Eu and NAm lineages described in the introduction. In contrast, pairs with either NAm–NAm or Eu–Eu assignment only are almost identical (> 98.5%) (Supplementary Fig. 7b, c). Approximately, 60% of phase switch errors were identified from chromosomes with homologous pairs with either NAm or Eu ancestry only (Supplementary Table 6).

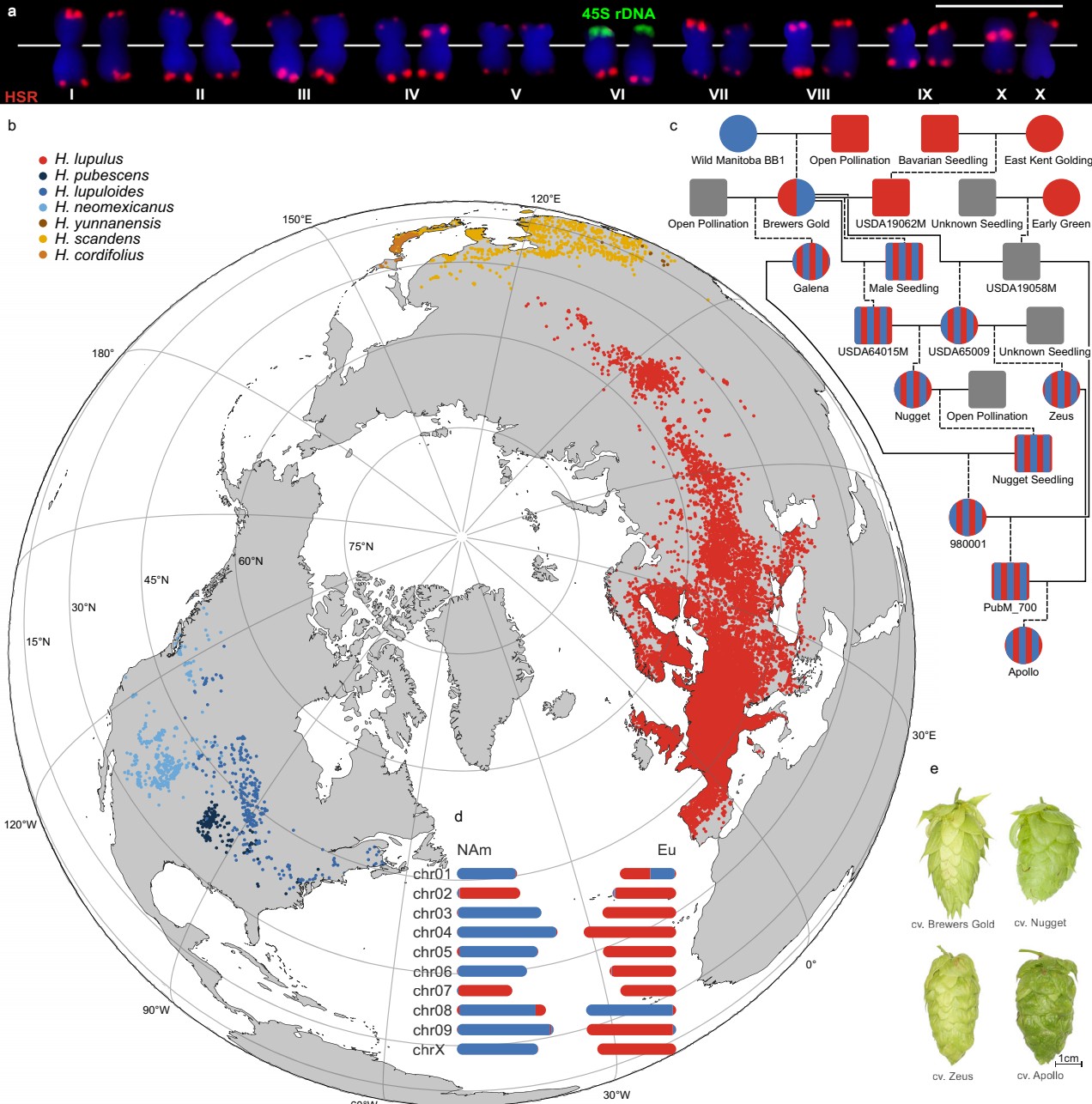

**Fig. 1 | Chromosome scale, haplotype-resolved phased genome assembly of cv. Apollo. a** Apollo karyotype ordered according to chromosome size and localization of the HSR subtelomeric tandem repeat (red) and 45S rDNA (green) on mitotic metaphase chromosomes. 45S rDNA sites are also labeled directly next to the corresponding signals. Chromosomes were counterstained with DAPI (blue). The sixth largest chromosome (VI) carries the HSR subtelomeric repeat on the q-arm and the 45S rDNA on the opposite p-arm as previously described[179]. The localization of the HSR tandem repeats differs between the two X chromosomes within one female cell. One X chromosome displays the expected pericentromeric HSR signal on the p-arm[180], the other displays a weaker HSR signal in the pericentromeric region on the q-arm. Scale bar = 10 μm. **b** Geographic distribution of *Humulus lupulus, Humulus lupuloides, Humulus neomexicanus, Humulus pubescens*, and *Humulus cordifolius*, the five species previously classified as botanical varieties[181] of *Humulus lupulus* (https://powo.science.kew.org/). Each dot displayed corresponds to a reported occurrence data point found on the Global Biodiversity Information Facility (GBIF). Red dots represent European, blue shaded dots North American and yellow and orange shaded dots represent Asian hop species. **c**) Known pedigree of hop cv. Apollo. Squares indicate males, while circles indicate females. Blue symbols represent North American (NAm) and red symbols European (Eu) germplasm. Gray symbols indicate unknown ancestry, while red and blue striped symbols depict Eu−NAm mixed genotypes and half red half blue is first generation hybrid of Eu−NAm. **d** Graphical overview of the haplotyped resolved phased chromosome scale assembly of Apollo illustrating regions in the genome of NAm and Eu ancestry in blue and red color. **e** Mature cones of hop cvs. Brewers Gold, Nugget, Zeus, and Apollo. Source data are provided as a Source Data file.

## Suppressed recombination in interspecific *Humulus* hybrids

The parent specific high copy k-mer analysis (Supplementary Figs. 8 and 9) unexpectedly revealed that despite the presence of interspecific multiple generation hybrids in six generations of the cv. Apollo pedigree[22,23], the original Eu and NAm haplotypes remain well-separated in its genome, with phase 1 being predominantly of NAm and phase 2 of Eu origin. This contrasts with the expected intermixed mosaic pattern of parental species linkage blocks in monoploid hybrids from the F2 generation onwards[31–33]. We therefore tested for suppressed recombination between Eu and NAm chromosomes in

Eu–NAm hybrids utilizing three bi-parental mapping populations: Apollo × PubM_740, Zenith × USDA21058M, and Cascade × HL-19-060-002M (Supplementary Data 1, Supplementary Table 7). While Apollo, Cascade, and USDA21058M comprise chromosomes of both Eu and NAm origin, PubM_740, Zenith and HL-19-060-002M are *H. lupulus* (Supplementary Fig. 11).

Linkage mapping following a pseudo-test cross strategy using markers heterozygous in the hybrid parents and showing a 1:1 segregation showed no or strongly suppressed recombination, in the Apollo × PubM_740 and the Zenith × USDA21058M population (Supplementary Fig. 12). Consequently, no linkage maps could be generated.

This observed pronounced reduction, up to an apparent absence, of recombination is striking given the essential role of crossovers in ensuring proper homolog pairing and chromosome segregation during meiosis. In a recent investigation of recombination landscapes in flowering plants comprising 665 chromosome maps from 57 species only 2% of chromosomes showed less than one crossover per chromosome[34] and, in most species, crossover assurance is necessary to achieve proper segregation during meiosis[35]. Still, Supplementary Fig. 13 shows very low or undetectable recombination rates across all ten chromosomes in the Apollo × PubM_740 mapping population. Apparent increase in recombination rates in a few regions, such as an Eu introgression on chr08.1, likely reflects high haplotype heterogeneity and sparse marker intervals rather than true meiotic exchange. Gene density in Apollo increases towards chromosome ends (Supplementary Fig. 5) while transposon density remains high along most of the chromosomal span (Supplementary Fig. 13), consistent with the presence of very large repetitive pericentromeric regions as also described by Akagi et al.[36]. In contrast to this study however, which analyses the *H. lupulus* landrace Saaz, we do not observe an increase of recombination towards chromosomal ends. This difference may arise because our study concerns an Eu–NAm hybrid with extensive divergence between ancestral haplotypes (Supplementary Fig. 7c), that are estimated to have diverged around 1.05 Mya[9]. Namely, crossover genetic determinants and/or changes in histone occupancy and distinct chromatin marks that potentially add further constraints to crossover formation along chromosome arms, though still broadly consistent with the typical range of crossover numbers observed across eukaryotic species[37]. Still, when considering genome-wide nucleotide diversity ($\pi$), or haplotype divergence ($d_{XY}$), we observe no consistent correlation of recombination suppression with local sequence divergence across chromosome arms (Supplementary Fig. 13).

We note that suppressed recombination is not the only atypical meiotic feature reported for hop. Significant deviation from expected Mendelian segregation patterns were observed in genetic studies[38,39]. Further, cytological studies supported previous observations, revealing severe abnormalities during meiosis I from synapsis onwards. These were reported to continue through the second division[40,41] with some genotypes exhibiting more meiotic abnormalities than others. A recent study revealed unbalanced segregation due to exclusion of bivalents from the metaphase plate, further corroborating meiotic abnormalities that are functionally disruptive in some genotypes, even in Eu landraces[42]. Further, Zhang et al.[41] found the expected pattern of ten bivalents resulting from correct pairing of the 20 hop chromosomes in less than 5% of the diakinesis-stage nuclei studied in two wild type and two hybrid crosses.

As outlined above, meiotic abnormalities are not restricted to hybrids but also occur in wild type crosses, indicating that diverged Eu–NAm ancestry is unlikely to be the sole contributing factor to suppressed recombination. Consistent with this suggestion, Horáková et al.[42] show a high intraspecific degree of diversity and complexity in the centromere organization of European *H. lupulus*, which was also found in cv. Apollo (Supplementary Fig. 14), and show that aberrant chromosomal segregation is linked to unusual centromere organization. The authors propose that unique centromeric structures contribute to the non-Mendelian segregation patterns they observe in *H. lupulus*.

In the context of severe but varying degree of meiotic abnormalities, it is noteworthy that linkage mapping following the same strategy as above in the Cascade × HL-19-060-002M population using 8751 markers heterozygous in Cascade resulted – in contrast to the two populations mentioned above – in the construction of a genetic map spanning 2864.69 centimorgans (cM) with an average marker density of 4.23 markers cM⁻¹ (Supplementary Table 8). In this case visual inspection of segregating markers and the comparison of genetic and physical maps showed no indications of suppressed recombination (Supplementary Fig. 15a, b). We detected an average of one or two recombination breakpoints per chromosome (Supplementary Fig. 15c).

Despite the observed lack of recombination in our current study and the reported extreme segregation defects during meiosis I and II in some hybrid crosses[40], most modern hop cultivars are the result of interbreeding Eu and NAm *Humulus* species. This is because such lines deliver higher overall yields as well as increased content of α-acids compared to traditional landraces and wild hops[11]. Even though the cause of the suppressed recombination described above remain to be finally resolved, our results imply that the advances of interspecific hop breeding to a large extend relate to the new assortment of chromosomes rather than actual recombination of traits. Therefore, we investigated preferences for certain Eu–NAm chromosome combinations in hop breeding. To that end, we genotyped 243 accessions using GBS (Supplementary Data 2 and 3), that were assigned as breeding lines, *H. lupuloides*, *H. neomexicanus*, commercial cultivars, and *H. lupulus*. We observed an increased proportion of Eu chromosome pairs, possibly reflecting the use of *H. lupulus* male genotypes in successive breeding generations (Supplementary Fig. 16). However, no preferences for any Eu–NAm chromosomal combinations were found suggesting that the improved performance of modern hop cultivars might rather be due to a heterotic effect than the result of selection.

A recent study on the climate sensitivity of European hop production predicts a decrease in hop yield by 4–18% and in α-acid content by 20–31% by the year 2050[43]. This calls for urgent adaptive measures including the breeding of new hop varieties. Modern plant breeding largely relies on the crosses between elite varieties. Lack of recombination however limits the progress that can be realized by breeding[44]. The same limitations consequently apply for the fine mapping of traits and the development of molecular markers which are until today not used routinely in *Humulus* breeding. To exemplify this, we carried out GWAS to identify loci associated with downy mildew resistance (DMR) (Supplementary Data 4) and powdery mildew resistance (PMR)[45] in the Apollo × PubM_740 and the Zenith × USDA21058M populations, respectively. We identified a 1.8 Mb region on chr02 associated with DMR (Supplementary Fig. 17), while the entire chromosome chr03, and a 1.56 Mb region on chr05 were strongly associated with PMR in the studied populations (Supplementary Fig. 18). The region on chr03 is consistent with the QTL reported previously[45]. The association of the entire chr03 with PMR aligns with the previously mentioned suppression of recombination, highlighting a fundamental challenge in both trait mapping and molecular marker development for hop breeding.

## Population genomics of European and North American hop

We further assessed the diversity present in the 243 hop genotypes mentioned above using 33,178 high quality SNPs (Supplementary Data 3, Supplementary Table 9). Comparison of $F_{st}$ values between male and female genotypes identified a pseudoautosomal region on chrX (Supplementary Fig. 19). Further, principal component analysis (PCA) grouped the genotypes based on their geographical origin (Fig. 2a). This is in concordance with a previous study that identified

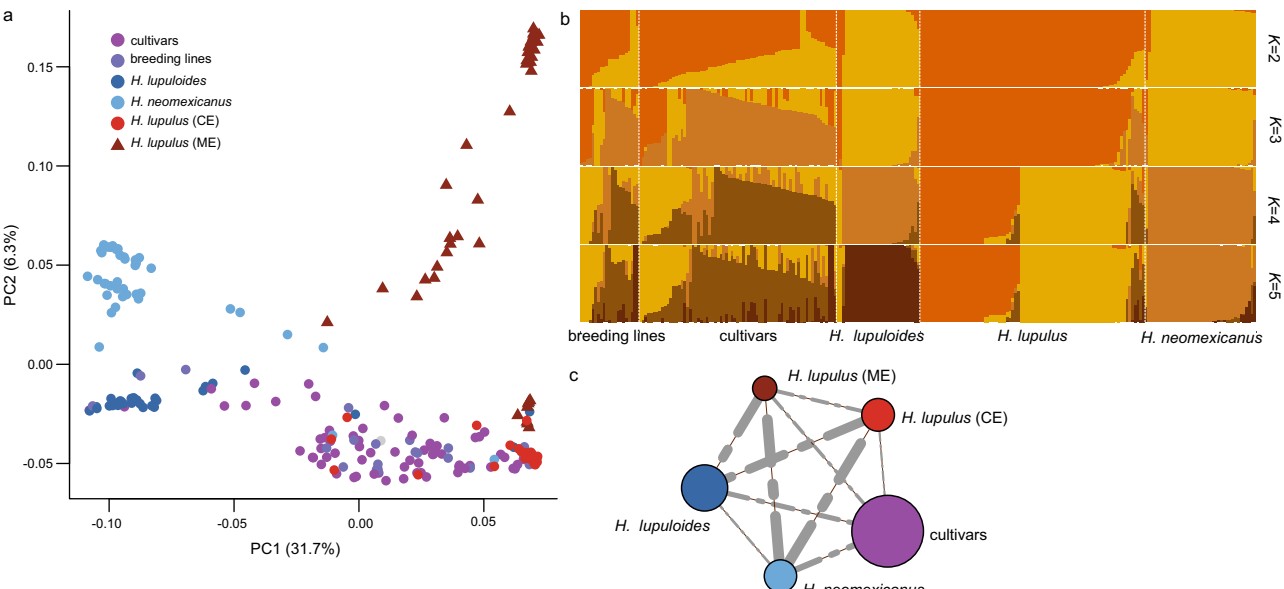

**Fig. 2 | Population genomic studies indicate emergence of a distinct lineage within European *H. lupulus*. a** Principal component analysis (PCA) showing diversity within 243 diverse hop genotypes, clustered by geographic origin. We observed two clusters within the *H. lupulus* genotypes, one constituted of genotypes from Central Europe (CE) and one of genotypes from the Middle East (ME). **b** Bar plot showing the distribution of ancestry coefficients from population structure analysis for $K = 2–5$. Each vertical bar represents an individual, colors denote inferred genetic clusters, and segment lengths reflect the proportion of ancestry assigned to each cluster. For most pure European and North American lines, little admixture was observed at $K = 5$, indicating distinct lineages. Two clusters within *H. lupulus* were identified at $K = 5$, supporting the emergence of a distinct lineage within European hop. **c** Nucleotide diversity and $F_{st}$ analysis to study diversity within and between populations. The circles represent nucleotide diversity within the populations, while the dotted lines represent $F_{st}$ values between the populations identified from PCA analysis. Circle size and line thickness are proportional with the nucleotide diversity and $F_{st}$ values. Highest nucleotide diversity was observed within cultivars. The $F_{st}$ between the two sub-clusters within *H. lupulus* was higher than that between the two North American species. Colors correspond to the key in (**a**). Source data are provided as a Source Data file.

NAm *Humulus* lineages as a monophyletic group distinct from European and closer to Asian hop[9]. The cultivars and breeding lines were clustered close to Eu genotypes likely due to the use of *H. lupulus* males in successive generations during breeding.

We observed two sub-clusters within the Eu genotypes included in our analysis: One specific to landraces and feral genotypes from Central Europe, while genotypes from the second cluster were mainly collected in the Middle East. Genotypes from both clusters did not show admixture, indicating that they have not interbred (Fig. 2b). Further, an $F_{st}$ analysis indicated higher divergence between these two sub-clusters within the *H. lupulus* lineage than between *H. lupuloides* and *H. neomexicanus*, respectively, suggesting the emergence of a distinct lineage within *H. lupulus* (Fig. 2c, Supplementary Table 10). *H. lupulus* genotypes have previously been reported to comprise low levels of genetic differentiation[9]. So far only in the Caucasus region a genetically isolated population of wild *H. lupulus* was identified[46,47]. Combined with our findings, this calls for a thorough taxonomic revision of *Humulus* genotypes sampled in Europe and the Middle East to investigate if currently undescribed distinct *Humulus* lineages exist on the Eurasian continent. Utilizing the full gene pool of *Humulus* will open broader opportunities for breeding more climate tolerant cultivars with optimized metabolite content.

### Genome size expansion and large structural rearrangements in *Humulus*

For a comparative analysis of gene content between hop and hemp (*C. sativa*), gene models in the haplotype-resolved, phased assembly of Apollo were annotated by an established pipeline[48] using ab-initio prediction, protein homology as well as RNA-Seq and Iso-Seq data from multiple tissues across various genotypes (Supplementary Data 5). After a consolidation step to correct for potentially missed genes in one phase the pipeline identified 30,920 (phase 1) and 30,398 (phase 2) high-confidence (HC) gene models with high completeness and minimal duplication (Supplementary Table 5). The differences in duplicated BUSCO genes and total gene count between both phases were attributed to a large extent to cis-translocations highlighting the impact of small-scale translocations in shaping gene content (Supplementary Fig. 20).

We observed an overall high collinearity and synteny in gene order among the two phases of *Humulus* cv. Apollo and the haploid genome assemblies of hop cvs. Cascade[13] and Saaz[36] as well as *Cannabis* cv. Pink Pepper[49] (Fig. 3a), but identified large translocations between *Cannabis* chromosomes chr04, chr08, and chrX and *Humulus* chr08, chr09, and chr04, respectively. The comparison between the two phases of the Apollo assembly identified an inter-chromosomal translocation between chr03 and chr05. We further observed a largely co-linear gene order between Apollo chromosomes and an available haploid genome assembly of European hop (NCBI GCA_963169125.1) further supporting the high quality and contiguity of our Apollo reference genome assembly (Supplementary Fig. 21).

In hop, gene density in proximal regions was lower (Fig. 3a) and no evidence for a whole genome duplication (WGD) was found (Supplementary Fig. 20). Transposable elements (TEs), which are highly abundant in large plant genomes, collectively occupy 56% (0.49 Gbp) of the hemp genome, while TE content in the Apollo genome is about 4.4-fold increased to 2.19 Gbp and 2.05 Gbp in phase 1 and 2, respectively. The expansion of the most prevalent family, *gypsy* LTR retrotransposons, alone contributes 77% to the ~3-fold genome size increase of 1.74 Gbp between hemp and hop (Fig. 3b, Supplementary Table 11). Hence, based on our analysis, the increased genome size of *Humulus* compared to *Cannabis* is explained by TE proliferation in *Humulus* rather than by WGD, which has been suggested in an earlier analysis[14]. TE insertion time analysis indicates the *Humulus* NAm and Eu lineages diverged about 1 Mya ago (Fig. 3c). This timepoint has been suggested

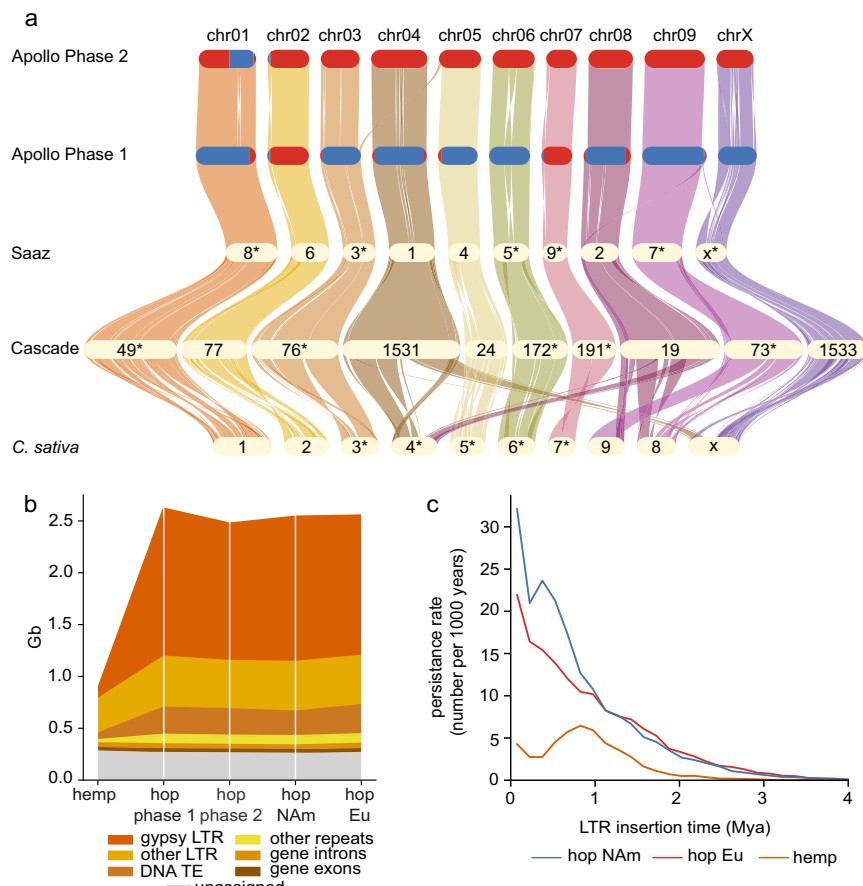

**Fig. 3 | Comparative analysis between the assemblies of *Humulus* cv. Apollo and *Cannabis sativa*. a** Gene based syntenic and orthologous relationships between *Cannabis sativa*, the haplotype-resolved phased assembly of hop cv. Apollo, as well as the hop cvs. Cascade and Saaz. Red and blue color indicate European (Eu) and North American (NAm) origin of the respective genomic regions in the Apollo genome. Chromosomes are labeled by original published identifiers where relevant, but gene order has been inverted for some contigs (*) to minimize crossings in the plot. Chromosomes are colored for visual distinction. Colors do not represent additional variables. **b** Transposon content in hop is increased 4.4-fold compared to hemp, while differences in transposon content between phases and ancestry in hop are marginal and related to assembly size differences. The genome size increase from hemp to hop is mainly driven by the expansion of *gypsy* LTR-retrotransposons. **c** A comparison of LTR-retrotransposon persistence rates reveals a strong increase in hop compared to hemp starting about 3 Mya. The higher and still ongoing transposon activity in hop leads to its genome size expansion. The divergence of the NAm and Eu rates starts about 1 Mya reflecting the geographic isolation on different continents and subsequent differential transposon histories. Source data are provided as a Source Data file.

in an earlier study[9] as well. The slight increase in size (155 Mb) between Apollo phase 1 compared to phase 2 parallels with an increase in *gypsy* LTR retrotransposon content occurring after the divergence of Eu and NAm hop lineages (Supplementary Table 11). Comparison of both assembly size and gene content revealed that the interspecific hybrid origin of the haplotype-resolved phased Apollo assembly essentially amounts to a reciprocal admixture of approximately 10% Eu origin in phase 1 and vice versa 10% NAm origin in phase 2 (Supplementary Table 12).

Investigation of the structural variations (SVs) between Apollo phase 1 and phase 2 (Fig. 4) identified a total of 52,593, 41,990, 4438, and 215 presence or absence variations (PAV), translocations, and inversions in phase 2 relative to phase 1, accounting for 1.8 and 1.67 Gb in phase 1 and phase 2, respectively (Supplementary Table 13). Most SVs occur between phases of contrasting Eu and NAm origin, while the lowest number of SVs were identified on chr02, which correlates with the common ancestry and consequently low divergence between its two phases.

The largest SV detected is an ~85 Mb inversion containing around 500 genes on chr06 (Supplementary Fig. 22a), confirmed in silico by mapping Hi-C reads from Apollo and Cascade (Supplementary Fig. 22b, c) against the haplotype-resolved phased assembly, and experimentally via PCR (Supplementary Fig. 22a, e, f). This inversion is present in Apollo phase 1 but absent in Cascade and could be traced back to cv. Brewer's Gold, a hybrid cultivar resulting from crossing a European *H. lupulus* male and a wild NAm female (BB1) collected in Manitoba, Canada, in 1916 within the native range of *H. lupuloides*[50] (Supplementary Fig. 22d, e). Released in 1934, Brewer's Gold was chosen for its high α-acid content. Many modern high α-acid hop varieties descend from Brewer's Gold[51] (Supplementary Fig. 22g), and the widespread occurrence of the inversion highlights the narrow genetic base utilized in developing contemporary hop cultivars (Supplementary Table 14, Supplementary Fig. 22f).

## Extensive variation in biosynthetic pathways between haplotypes

The extensive structural variation and sequence divergence between Eu and NAm chromosomes in cv. Apollo described above prompted a more detailed investigation of phase divergence at gene level. Because hybridization between Eu and NAm hops has been driven by the pursuit of higher bittering and aroma content, our analysis focuses on gene families involved in the related key specialized metabolic

## Apollo chromosomes

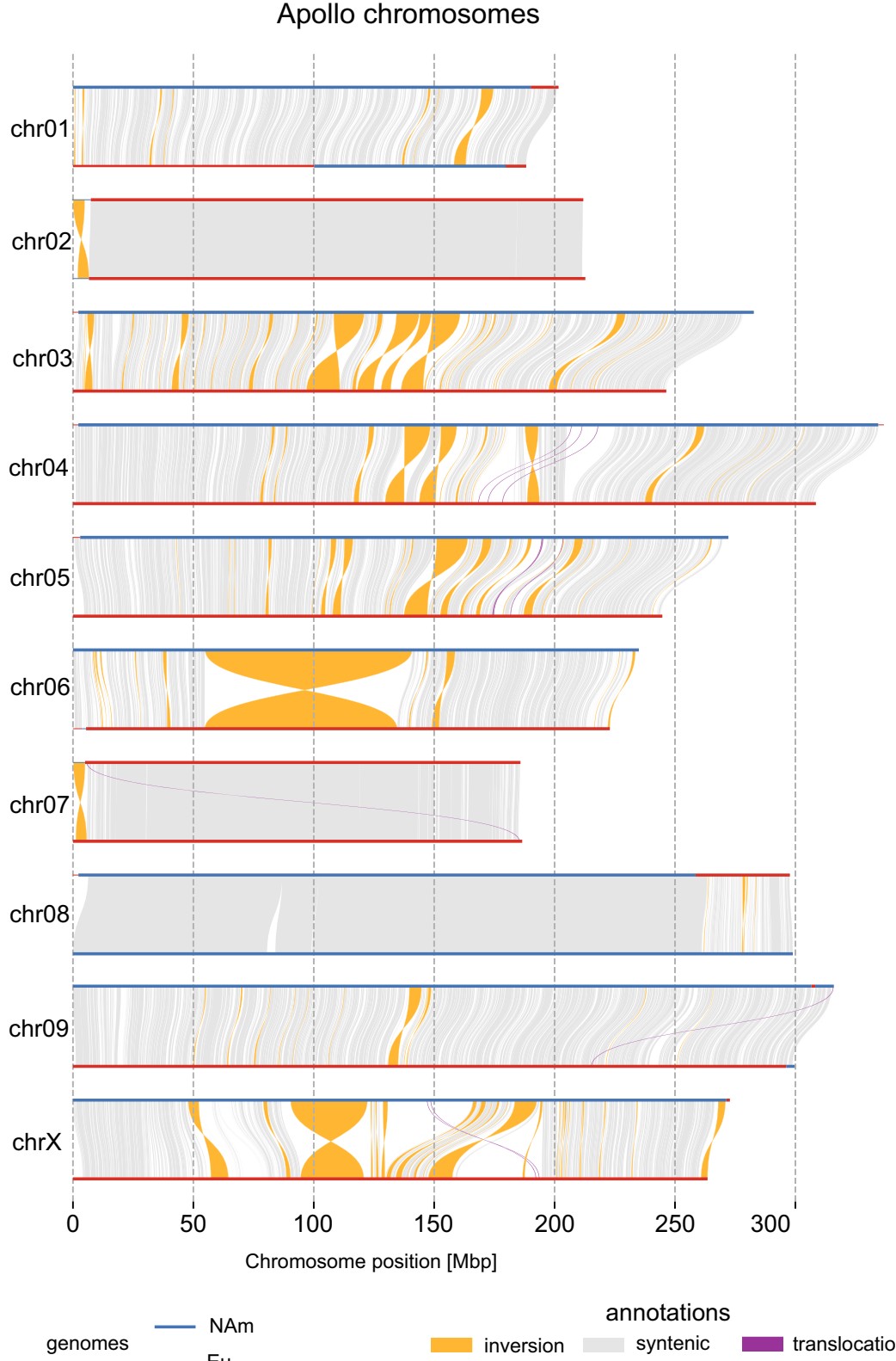

**Fig. 4 | Large structural rearrangements between the two phases of the cv. Apollo genome.** Assembly-to-assembly collinearity plot showing structural variations between the two phases in cv. Apollo. Structural variants were identified from the whole genome alignment and classified as presence or absence relative to phase 1. An -85 Mb inversion was identified between two phases of chr06. Abbreviations: NAm North American, Eu European. Source data are provided as a Source Data file.

pathways leading to flavor- and aroma-contributing terpenes and prenylated polyketides such as brewing-relevant bitter acids and pharmaceutically significant xanthohumol and 8-prenylnaringenin: terpene synthases, prenyl-transferases and polyketide-synthases[52–56].

Due to its lineage resolution our genome provides a framework to examine how these central metabolic pathways differ on sequence level, gene copy number, and expression dynamics between the Eu and NAm haplotypes.

A gene ontology (GO) enrichment analysis across the Rosales (Supplementary Fig. 23, Supplementary Data 6) and Cannabaceae (Supplementary Fig. 23, Supplementary Data 6–8) revealed a Cannabaceae specific expansion in gene families involved in the biosynthesis of commercially relevant metabolites named above. The enrichment indicates a pronounced diversification of specialized metabolic processes within the Cannabaceae leading to the biochemical complexity found in hops.

Figure 5 displays phylogenies of the terpene synthase (TPS) (Supplementary Data 9), aromatic prenyl-transferase (PT) (Supplementary Data 10) and polyketide-synthase (PKS) (Supplementary Data 11) gene families in the Rosales. Analysis of PTs revealed pronounced diversification within the Cannabaceae. The Apollo genome encodes 10 PT loci in phase 1 and 8 in phase 2, with *hlpt1* and *hlpt2*, encoding key enzymes in bitter-acid biosynthesis, located on chrX. Our analysis shows that the plastoquinone-related subfamily is expanded in the Cannabaceae and includes a well-supported clade containing hop and hemp enzymes involved in bitter-acid and cannabinoid biosynthesis. Among type III polyketide synthases[52] (PKS) a previously undescribed duplication of the chalcone synthase (*chs*) gene was identified on chr09 of phase 2 (Eu), together with four additional *chs* homologs (*hlchs*2–5)[52], of which three are expressed in glandular trichomes.

Within the terpene synthase (TPS) family, specifically the TPS subfamilies TPS-a, TPS-b, and TPS-g are involved in the biosynthesis of volatile flavor- and aroma-contributing terpenes[57]. The TPS-b and TPS-g subfamilies were expanded in Apollo. The TPS-g subfamily, synthesizing linear mono- and sesquiterpenes underpinning the resinous aroma of hops, showed increased copy numbers on phase 1 (Fig. 5a) and evidence for multiple tandem duplications on chr01 (Fig. 5d). Supplementary Data 12 shows a comprehensive summary of the genes included in the phylogenetic analysis shown in Fig. 5a–c in cv. Apollo along with their expression.

To complement this genomic perspective with transcriptional information, we next examined whether one of the two genomic phases showed expression dominance in the developing cone. We compared the proportion of expressed genes per phase at each developmental stage across chromosome pairs. This analysis revealed no consistent genome-wide dominance in gene expression. Overall, expression was largely balanced between the two phases. Only chromosomes chr03, chr05, and chrX exhibited a statistically significant bias favoring phase 1 (NAm) over phase 2 (Eu), and even in these cases the difference was minor, with an average of 51.3% of expressed genes originating from phase 1 (Supplementary Fig. 24).

Building on these genome- and phase-level analyses, we next explored how gene regulation shapes metabolite accumulation during cone development. A co-expression analysis across five developmental stages of the cone identified four clusters of differentially expressed genes (DEGs) with distinct temporal trends (Fig. 6b, Supplementary Note 1). These clusters were correlated with metabolite-abundance data from an untargeted analysis of the same samples (Fig. 6c, Supplementary Note 1). The resulting correlation network illustrates a dynamic chemo-transcriptional landscape during cone development, with DEGs from clusters 2 and 4 increasingly correlated with diverse volatile (VC) and non-volatile compound (NVC) families. These associations trace back to individual genes, showing that key steps in the biosynthetic pathways producing flavor- and aroma-contributing terpenes and bittering acids are driven by increasing gene expression over the course of cone development, peaking 2–3 weeks after flowering (Fig. 6d). This transcriptional activation is followed by an increase in the abundance of the respective compounds, indicating transcriptional regulation of specialized metabolite formation in hop. While the expression of certain terpene synthases peaks around weeks 2–3 after flowering and subsequently declines, associated terpene levels continue to increase toward late developmental stages, in contrast to the bittering acids humulone and lupulone, which stabilize as gene expression decreases.

Gene expression levels and metabolite abundance are not necessarily correlated, since transcripts following a distinct pathway are often co-regulated, metabolites are often transformed and relocated to fulfill their roles[58]. However, an observation in our dataset linked to primary metabolism provides interesting leads for further investigation: We explored gene expression in the mevalonate (MVA) and methylerythritol phosphate (MEP) pathways (Supplementary Data 13). Those produce isopentenyl diphosphate (IPP) and dimethylallyl diphosphate (DMAPP), the essential C5 building blocks for all terpenes and bitter acids in hops[56,59]. Genes of the plastid-localized MEP pathway, which colocalizes with monoterpene biosynthesis and bitter-acid prenylation, were differentially expressed during cone development, often following the transient rise-and-fall pattern of DEG 3. In contrast, genes of the cytosolic MVA pathway, colocalizing with sesquiterpene formation, showed stable expression. This divergence may contribute to the continued accumulation of sesquiterpenes such as α-humulene and β-caryophyllene during late cone development (Fig. 6d) by maintaining a more continuous precursor supply. This correlation is further displayed within the chemo-transcriptional network presented in (Fig. 6c), which highlights the alignment of individual sesquiterpene abundances with declining or slowly increasing gene expression patterns of DEG 1 and 4, in contrast to the more fluctuating nature of other compounds, particularly of the non-volatile lipid-like class, involving oxidized terpenes as well as bitter acid derivatives.

## The impact of NAm and Eu haplotypes on α-acid content

The pronounced divergence between haplotypes described above not only reflects their distinct evolutionary histories but also suggests that they may contribute differently to key agronomic traits. Yet, previous population-based studies investigating α-acid content have not resolved from which ancestral backgrounds beneficial alleles originate[60,61]. To address this, we performed GWAS to identify genomic regions linked to elevated α-acid content and to trace their Eu or NAm ancestry using the haplotyped resolved Apollo assembly.

Although a single haplotype reference for GWAS is preferable to minimize redundant or inconsistent variant calls, likely neither of the two phases in the haplotype-resolved Apollo assembly provides a universally optimal reference for association mapping: Due to its pedigree the pool of segregating chromosomes in the Apollo × PubM_740 population has an approximate NAm:Eu ratio of 1:3 (Fig. 1; Supplementary Fig. 11). Using phase 2 (predominantly Eu) therefore provides a closer match to the genetic background of this population. However, NAm specific variants, which already occur at lower frequency in the population, map less efficiently to phase 2 (Supplementary Fig. 10), potentially further reducing statistical power for their detection in association mapping. Therefore, we performed two separate GWAS using each of the phases as reference respectively.

The phase 2 referenced GWAS revealed a ~30 Mb region on chr08, containing a 26 Mb Eu introgression in the otherwise NAm chr08.1, to be associated with increased α-acid content in a GWAS using 21,756,972 SNPs across female individuals (Fig. 7a) as a single major association. No recombination was detected across the introgression. The phase 1 referenced GWAS detected the same region as well as additional significant associations on chr05 and chr09 (Fig. 7b), confirming that the two haplotype references contribute complementary information. GWAS using both phases sequentially as reference to detect genome regions associated with β-acid content also produced complementary results (Supplementary Fig. 25). These findings support our initial rationale that both references are required to reveal a complete picture of genomic regions associated with beneficial traits.

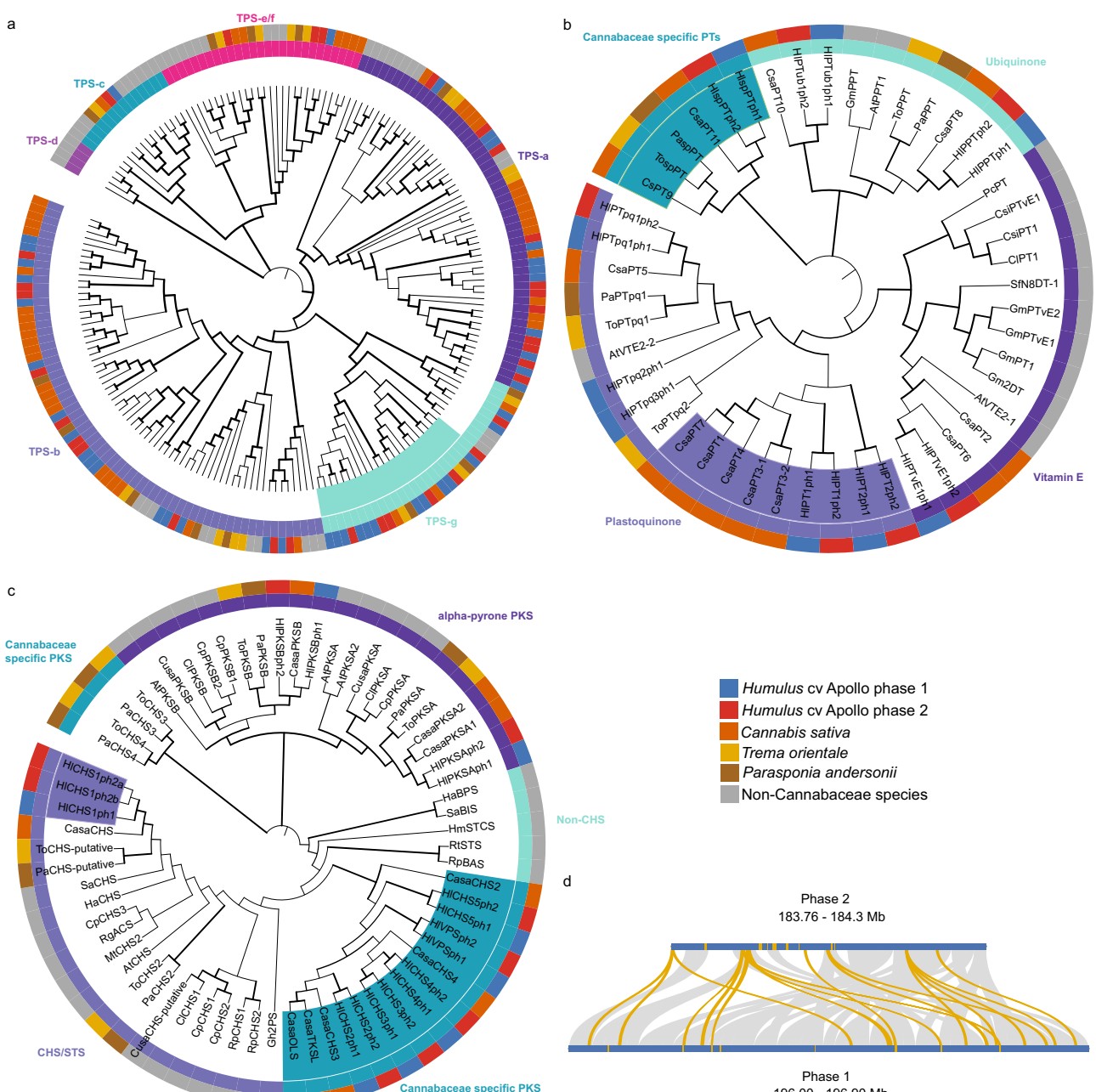

**Fig. 5 | Phylogenetic analysis of protein families involved in high value compound biosynthesis. a** Terpene synthase (TPS) phylogeny based on amino acid sequences reveals expansion of a subclade of the TPS-g subfamily in phase 1 of the cv. Apollo assembly. The respective protein sequences are highlighted in turquoise next to the inner circle. **b** Prenyltransferase (PT) phylogeny based on amino acid sequences. PTs involved in α- and β-acids biosynthesis and cannabinoids are high-lighted in purple next to inner circle, whilst those from the Cannabaceae specific clade are highlighted in teal next to inner circle. **c** Plant type III polyketide synthase phylogeny based on amino acid sequences. The bona fide chalcone synthase (CHS),

synthesizing the backbone structure of xanthohumol, shows a duplication in cv. Apollo phase 2 (Eu), highlighted in light purple next to the inner circle. The enzymes for the biosynthesis of α- and β-acids, and cannabinoids are grouped in a Cannabaceae-specific clade highlighted in teal next to inner circle. For **a**–**c** Coloring of the inner circle corresponds to gene family depictions given next to the phylogeny. Outer circle colors indicate species as shown in the key. **d** Synteny analysis of the TPS-g subfamily (yellow) on chr01 between both phases revealed that the expansion of TPS-g subfamily members in phase 1 has included tandem gene duplications and chromosome rearrangements. Source data are provided as a Source Data file.

We next examined the associated regions in more detail to characterize how they contribute to α-acid content. For the Eu intro-gression identified on chr08, a k-mer-based analysis of the individuals in the Apollo × Pub_M740 population identified three distinct geno-type clusters corresponding to individuals carrying zero, one, or two copies of the Eu introgression, following a 1:2:1 segregation pattern. Clusters with one or two introgression copies exhibited significantly higher α-acid levels, indicating dominant inheritance of this trait (Supplementary Table 15, Supplementary Fig. 26, Fig. 7c). An overview

of the gene content within the chr08.1 introgression is provided in Supplementary Data 14. The region contains 514 predicted genes. Among those, 73.5% depict orthologs between the two phases, while 25.5% represent para-orthologs or diverged loci, indicating evolu-tionary conservation while displaying signatures of recent or ongoing diversification in this region. No genes known to be directly involved in the biosynthesis of α-acids were found in the introgression. However, the region contains several regulatory genes, among those HUM-LU.APOL.r2.08P1G2339100. This gene encodes the transcription

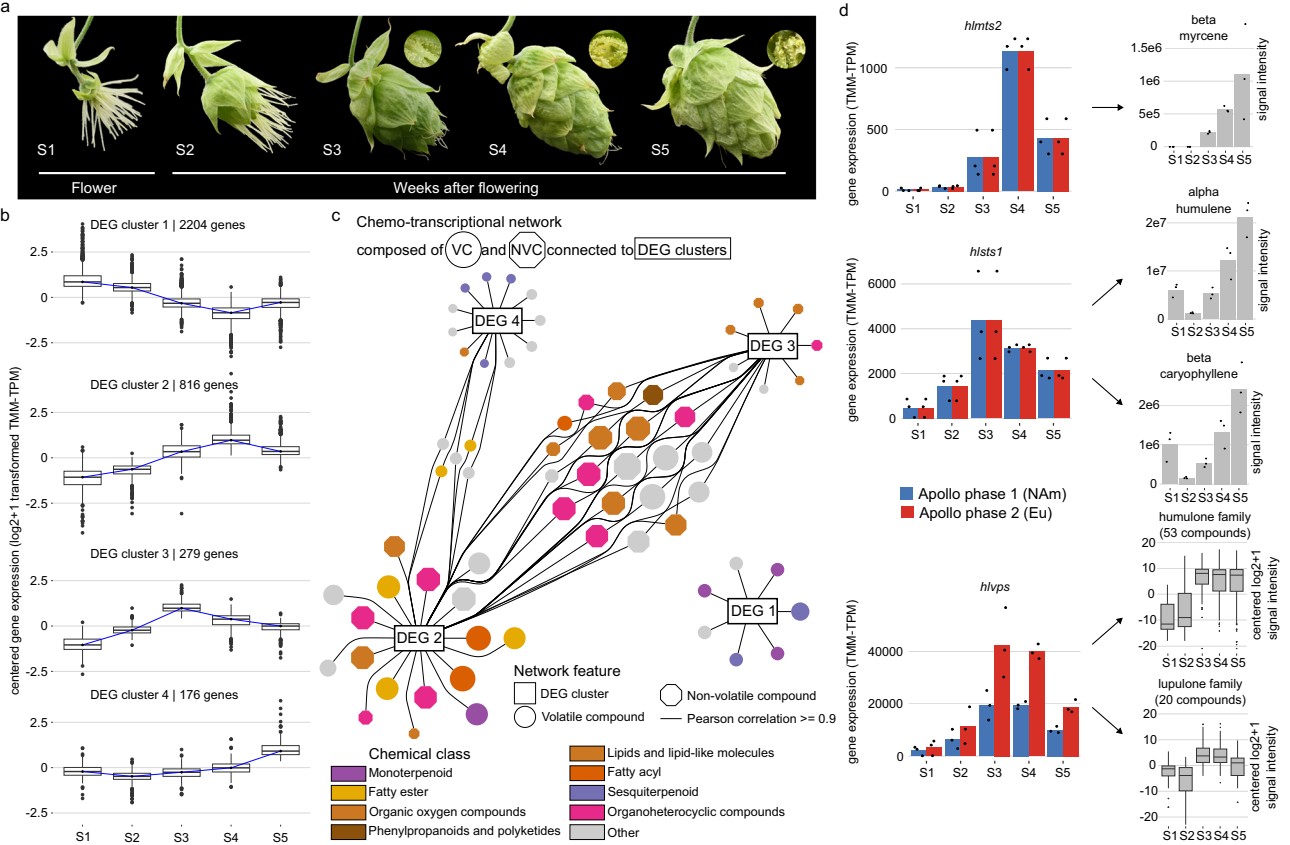

**Fig. 6 | Chemo-transcriptional framework in developing cones of cv. Apollo.**
**a** Five stages of hop cone development corresponding to time of flowering (S1) and 1 to 4 weeks after flowering (WAF, S2 to S5). Insets show glandular trichomes (lupulin glands), the main sites of specialized metabolite biosynthesis and storage. For transcriptomic analysis, cone tissue was separated into glandular trichomes and bracts/bracteoles. **b** Clusters of differentially expressed genes (DEGs) with similar temporal profiles during cone development. Gene expression is shown as centered $\log_2(x+1)$ transformed TMM-corrected TPM values (TMM-TPM), gene numbers are indicated on top of each graph. ($n = 3$ biological replicates). **c** Chemo-transcriptional network showing correlations between four DEG clusters (rectangular node; DEG 1–4) with volatile compounds (VC, circle node) and non-volatile (NVC, octagon node) chemical families. Edges indicate ≥ 5% of DEGs positively correlated (Pearson ≥ 0.9) with a given feature. **d** Correlation of chemistry and

transcription of genes during cone development ($n = 3$ biological replicates), illustrated for genes involved in the biosynthesis of major VCs, *hlmts*2 (*Humulus lupulus myrcene synthase* 2) and *hlsts*1 (*Humulus lupulus humulene synthase* 1), and NVCs *hlvps*1 (*Humulus lupulus valerophenone synthase* 1). Both, *hlmts*2 and *hlsts*1 are homozygous, resulting in identical measured expression between alleles, as allele-specific expression cannot be distinguished. In contrast, *hlvps* is heterozygous, allowing allele-specific differences in expression to be observed. Boxplots shown in panel b and d are defined with center line indicating the median, box limits represent the 25th and 75th percentiles (interquartile range, IQR), while whiskers extend to the smallest and largest observations within 1.5 × IQR of the lower and upper quartiles. Observations beyond the whiskers are plotted as individual outliers. Abbreviations: NAm North American, Eu European. Source data are provided as a Source Data file.

---

factor HlWRKY1, which forms a binary complex with HlWDR1 that acts as an activator of genes involved in the terminal steps of prenyl-flavonoid and bitter-acid biosynthesis. The complex further promotes the formation of a ternary MBW transcription factor complex, which regulates more upstream genes in the same pathways[62,63].

Given that the chr08.1 Eu introgression shows as a consistent and major association in both haplotype-resolved GWAS analyses and contains a gene encoding a regulator with a well-established role in bitter-acid and prenylflavonoid biosynthesis, our data indicate that this region plays a key role in determining α-acid content. Earlier mapping studies identified the same region to be associated with stable elevated α-acid content in cv. Magnum. The identified marker sequences (AY588355.1[60], AF515719.1[61]) localize to the same genomic region as the Eu introgression on chr08. The cv. Magnum is derived from cv. Galena[23], which contributed to the Apollo lineage (Fig. 1c), supporting a central role of the chr08.1 Eu introgression in breeding for optimized α-acid content in modern high α hop cultivars.

Additional genomic regions associated with α-acid content were found on chr05 and chr09 spanning the respective full chromosome. Individuals of the Apollo × PubM_740 population with NAm–Eu

genotype for chr05 have a higher α-acid content than those with Eu–Eu genotype (Fig. 7c). Given the size of this association, further investigations are necessary to identify high confidence target genes. However, this chromosome contains the genes encoding monooxygenases HlHS1 and HlHS2 (Supplementary Fig. 27). Recent experimental evidence[64] confirms the involvement of the HlHS1 enzyme in the oxidation of deoxyhumulone towards humulone, the final step in α-acid biosynthesis. We found NAm dominant expression of the *hlhs1* gene (Fig. 7d), and an additional RNAseq experiment of the developing cone in nine F1 individuals of the Apollo × PubM_740 population with Eu–NAm genotype at the *hlhs1* locus confirmed dominant expression of the NAm allele (Fig. 7e).

We did observe differences in α-acid content in individuals with a NAm–Eu genotype for chr09 compared to individuals with an Eu–Eu constitution (Fig. 7c), however in this direct comparison the effect was not significant. This might be due to the absence of extreme NAm–NAm genotypes in the population, due to the parental genotypes of cv. Apollo (NAm–Eu) and PubM_740 (Eu–Eu), or because of the presence of other beneficial alleles in the control group. The chr09 association is consistent with previously reported SRR marker 3a88-219

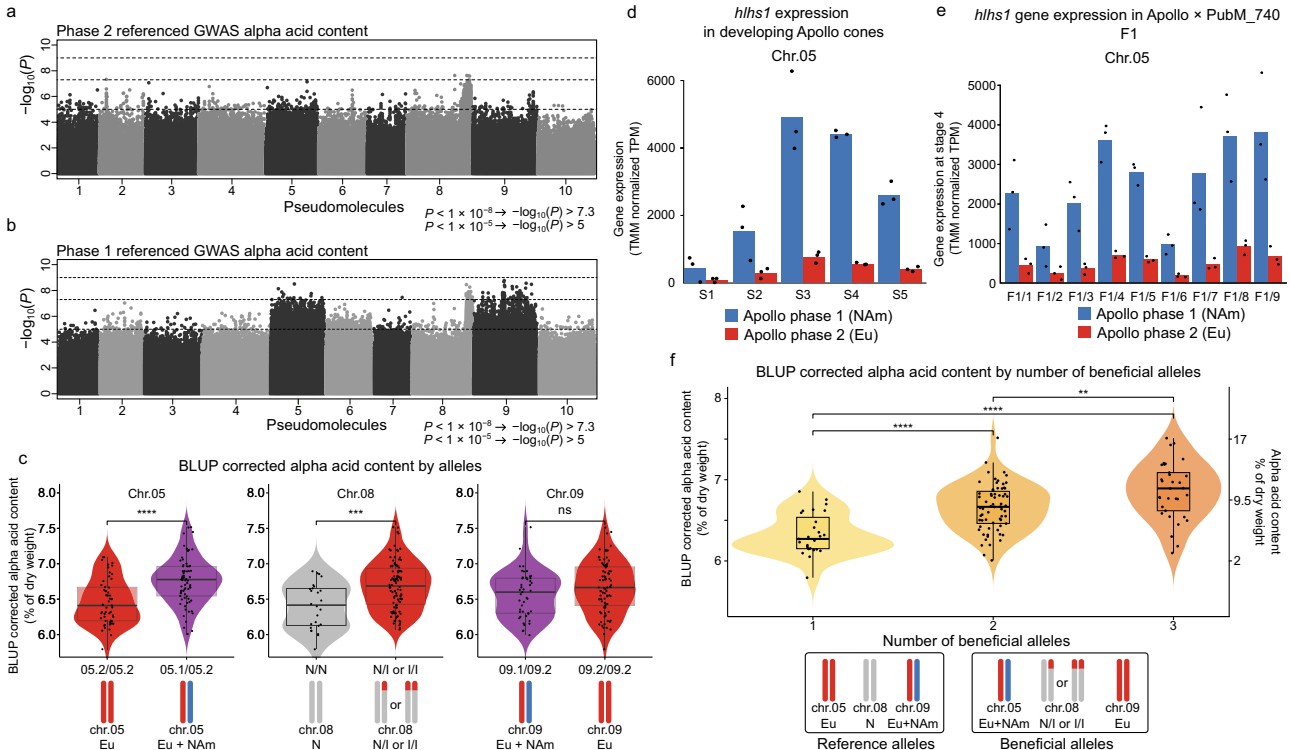

**Fig. 7 | European and North American haplotypes impact α-acid content in the Apollo × PubM_740 F1 population.** GWAS of α-acid content referenced with **a** Phase 2 (n = 131) and **b** Phase 1 (n = 126). Linear mixed models (two-sided tests) were applied with an α = 0.05 Bonferroni threshold (dotted line). **c** Differences in α-acid BLUPs were assessed using two-sided Wilcoxon rank-sum tests with Benjamini-Hochberg false discovery rate (BH-FDR) correction (chr05: n = 55 vs 73, P = 1.3 × 10⁻⁵; chr08: n = 26 vs 102, P = 7 × 10⁻⁴; chr09: n = 45 vs 83, P = 0.114). **d** Allele-specific expression of *hlhs1* (*Humulus lupulus humulone synthase* 1) during cone development (S1-S5) shows higher expression of the NAm compared to the Eu allele (n = 3 biological replicates). **e** This expression pattern is maintained in female individuals of the Apollo × PubM_740 F1 population (n = 3 biological replicates).

**f** Grouping individuals by number of beneficial alleles (1, 2 or 3) shows a significant effect (Kruskal–Wallis P = 3.6 × 10⁻⁸). Pairwise differences were assessed using two-sided Wilcoxon rank-sum tests with BH-FDR correction (1 vs 2: n = 27 vs. 66, P = 4.2 × 10⁻⁶; 1 vs. 3: n = 27 vs. 33, P = 1.2 × 10⁻⁷; 2 vs. 3: n = 66 vs. 33, P = 0.006). **c**, **f** Violin plots show data distribution; dots represent individual samples. Box plots show the median (center line), box limits indicate the 25th–75th percentiles and whiskers extend to 1.5 × the interquartile range, coinciding with minima and maxima for all groups shown. Violin plots are colored for visual clarity. **c–f** Genotype abbreviations: Eu European, NAm North American, N Non-introgressed, I Introgressed. Source data are provided as a Source Data file.

(AY382854.1) and AFLPs associated with increased α-acid content cv. Nugget[39]. Chr09 harbors the gene encoding the VPS1 enzyme, which catalyzes the first committed step of the bitter acid biosynthetic pathway[56].

In this context, we examined whether the genomic regions identified in the two GWAS exert an additive influence on α-acid content in the Apollo × PubM_740 population. Our results confirm the presence of such an additive effect as a significant increase in α-acid content is observed across the population dependent on whether one, two or three beneficial alleles are present in an individual (Fig. 7f). This finding provides evidence demonstrating how the combination of advantageous NAm and Eu alleles directly impacts α-acid content, the commercially most relevant trait in hop breeding. Moreover, our results highlight the essential role of determining species origin for specific haplotypes to better comprehend how interspecific hybridization influences key agronomic traits in breeding programs.

## Discussion

The haplotype-resolved phased assembly of the *Humulus* cv. Apollo represents a pivotal advancement in the application of contemporary genomic tools in hop research and underscores the immense value of haplotype resolution in plant genome assemblies. In contrast to prior haploid draft assemblies[13,14], our haplotype-resolved framework enabled us to link the Eu and NAm ancestry of individual chromosomes or genomic regions to elevated α-acid content, the key agronomic trait in hop. By showing that interspecific breeding has primarily reshuffled whole chromosomes and by identifying specific Eu and NAm haplotypes on chr08 and chr05 that additively contribute to increased α-acid content, our study highlights how targeted selection of defined lineage segments can be used to optimize agronomic traits in hops. Further, the narrow genetic base of modern cultivars and the pronounced differentiation between two *H. lupulus* sub-clusters in Europe and the Middle East argue for a broader exploitation of the *Humulus* gene pool to breed more climate-resilient, high-yielding varieties. Consequently, the cv. Apollo genome provides both a conceptual and a practical foundation for designing future hop breeding strategies around haplotype-informed selection rather than marker discovery alone.

Through the haplotype resolution of the Apollo assembly, we have further gained deeper insights into the intricate structural and genetic architecture of the complex hop genome. This will facilitate substantial enhancements in the genetic understanding of agronomic traits and ultimately allow the exploration of allelic variation, copy number variation, gene presence/absence variation and allele specific expression. We have been able to gauge the extent of genetic diversity between individual hop genotypes within genes and pathways involved in the biosynthesis of valuable metabolites, which are of paramount importance in hop. The thorough annotation of the genes present in the two *Humulus* species encompassed in the Apollo genome will expedite subsequent integrated pan-genome, pan-transcriptome, and pan-metabolome analyses designed to elucidate and discover the aforementioned pathways[65]. This will ultimately lead to an

improved understanding of how to breed for optimized content of valuable metabolites such as xanthohumol and 8-prenylnaringenin, among others. The targeted exploitation of *Humulus* genetic diversity for improved agronomic and biochemical traits will enable a significant stride towards the utilization of hops beyond the brewing industry.

## Methods

### Plant material
The chromosome-scale haplotype-resolved phased assembly was constructed for *Humulus* cv. Apollo[22]. The plants used for sequencing were grown and maintained under standard agronomic conditions in an experimental nursery at the Golden Gate Ranches, S.S. Steiner, Inc., Yakima, WA, USA (46.30211 N, 120.076365 W).

### Karyotyping
For mitotic chromosome preparation young leaves (2–5 mm long) were collected from intensively grown Apollo plants. Leaves were pretreated with 0.002 M 8-hydroxyquinoline for 4 h (2 h at room temperature and 2 h at 4 °C in dark), fixed overnight in Clarke's fixative (ethanol:glacial acetic acid (GAA), 3:1 (v/v)) at 4 °C, before using for chromosome preparation. Fixed leaves were washed 1 × 5 min in distilled water, 1 × 5 min in 45% acetic acid, 2 × 5 min in 0.001 M citrate buffer and macerated in 1% enzyme mix (1% pectolyase; P3026, Sigma-Aldrich, 0.75% cellulase R-10 Onozuka; C8001.0005, DuchefaBiochemie, 0.75% cellulase; 219466, Sigma-Aldrich, 1% cytohelicase; C8274, Sigma-Aldrich) diluted in 0.001 M citrate buffer for 50 min at 37 °C. The leaves were then squashed in 60% acetic acid. Well preserved metaphases were frozen in liquid nitrogen and coverslips were removed using forceps. Prepared slides were incubated 1 × 5 min in freshly prepared Clarke's fixative and used for fluorescence in situ hybridization (FISH) or stored at −20 °C in 96% ethanol until use.

*Humulus lupulus* subtelomeric repeat (HSR; GU831574.1), and 45S rDNA sequences (AF223066.1) were amplified from cv. Apollo genomic DNA using specific primers (HSR: HSR_FTAAGGCTCCGTTTC-GACTCC + HSR_RAACTTTTGTGCCCATGACCC; 45S rDNA: 45S_rDNA_F TGCCCGTTGCTCTGATGATT + 45S_rDNA_R TCCACCAACTAAGAACG GCC)[66]. PCR products were purified following QIAquick (Qiagen) PCR purification kit protocol. Purified DNA was labeled following fluorescent nick translation labeling kit (Jena Bioscience, Germany) instructions using Cy3 and fluorescein dyes for HSR and 45S rDNA probes, respectively.

FISH was performed on mitotic metaphase chromosomes[67] using 77% stringency. Chromosomes were counterstained with 4´,6´-diamidino-2-phenylindole (DAPI) and mounted in Vectashield Antifade Mounting Medium. Images were captured using an epifluorescence microscope Olympus AX70 (Olympus, Japan) equipped with a cooled cube camera and images were processed using Adobe Photoshop 2023.

### Geographic distribution of *Humulus* species
Six occurrence datasets were downloaded from Global Biodiversity Information Facility (GBIF) GBIF.org (https://doi.org/10.15468/dl.w53rkc, https://doi.org/10.15468/dl.cr37wu, https://doi.org/10.15468/dl.6ye7ug, https://doi.org/10.15468/dl.8jnu9g, https://doi.org/10.15468/dl.mn7b6d, and https://doi.org/10.15468/dl.rzwt9h) and filtered based on coordinate data to generate a derived dataset. The filtered and combined dataset is available at Zenodo[68] and registered as a GBIF Derived Dataset[69]. Only data points that matched native distributions as described by Plants of the World Online[8] (powo.science.kew.org) for each species were kept. To reduce overplotting and complexity, geographic coordinates were rounded to the nearest 0.25° in both latitude and longitude and location points with duplicate coordinates were removed. The geographic map was generated in

Python using Cartopy (v0.25.0)[70], and the base map was obtained from Natural Earth.

### Estimation of genome size
Paired-end (PE) 250 bp reads generated from sequencing of 450 bp PCR-free library of cv. Apollo (please see bi-parental population section) were used to estimate genome size using k-mer based approach. The 101 bp k-mer sequences were extracted from the raw reads using KMC (v3.2.1)[71] software and histogram of k-mer sequence frequency was generated using kmc_tools from KMC (v3.2.1) which was later used to estimate genome size using findGSE package with model fitting for heterozygosity[72] from R statistical environment.

### PacBio HiFi library construction and sequencing
For DNA extraction young leaves were collected from a single Apollo individual, and flash frozen in liquid nitrogen and transferred to the service provider (Polar Genomics, Boyce Tomson Institute, Ithaca, NY, USA). High molecular weight (HMW) genomic DNA (gDNA) from plant nuclei was prepared as described[73]: Nuclei were isolated from ~20 g frozen leaf tissue homogenized in ice-cold homogenization buffer (10 mM Trizma base, 80 mM KCl, 10 mM EDTA, 1 mM spermidine, 1 mM spermine, pH 9.4−9.5) supplemented with 0.5% Triton X-100 and 0.15% β-mercaptoethanol, followed by filtration, differential centrifugation and repeated washing. Purified nuclei were embedded in 1% low-melting-point agarose and lysed for 24 h at 50 °C in 0.5 M EDTA (pH 9.0−9.3), 1% sodium lauryl sarcosine and 0.1 mg mL⁻¹ proteinase K. Agarose-embedded DNA was subsequently washed in EDTA and stored at 4 °C until further use.

DNA quantity and quality parameters were assessed using both spectrometry (ND-1000; NanoDrop, Thermo Scientific, USA) and fluorometry (Qubit 2.0 Fluorometer, Invitrogen, USA) methods with 260/280 and 230/260 ratios ranging from of 1.82 to 1.84 and 1.7 to 2.16, respectively.

The libraries for PacBio high-fidelity (HiFi) circular consensus sequencing (CCS) were constructed at the Earlham Institute, Norwich, UK using the SMRTbell Express Template Prep Kit 2.0 (Pacific Biosciences, USA). Briefly, 26 μg of gDNA was manually sheared with the Megaruptor 3 instrument (Diagenode, USA) according to the operations manual. Each aliquot of sheared gDNA underwent AMPure PB bead (Pacific Biosciences, USA) purification and concentration before undergoing library preparation using the SMRTbell Express Template Prep Kit 2.0 (Pacific Biosciences, USA). The HiFi libraries were prepared according to the HiFi protocol version 03 (Pacific Biosciences, USA) and the final library was size fractionated using the SageELF system (Sage Science, USA), 0.75% cassette (Sage Science, USA). The resulting libraries were quantified by fluorescence (Qubit™ 3.0, Invitrogen, USA) and the size of fractions and libraries were estimated from a smear analysis performed on the FEMTO Pulse System (Agilent, USA).

The loading calculations for sequencing were completed using the PacBio SMRTLink Binding Calculator v9.0.0.92017. Sequencing primer v2 was annealed to the adapter sequence of the HiFi libraries. The libraries were bound to the sequencing polymerase with the Sequel II Binding Kit v2.0 (Pacific Biosciences, USA). Calculations for primer and polymerase binding ratios were kept at default values for the library type. Sequel II DNA internal control was spiked into each library at the standard concentration prior to sequencing. The sequencing chemistry used was Sequel II Sequencing Plate 2.0 (Pacific Biosciences, USA) and the Instrument Control Software v9.0.0.92233.

Libraries were sequenced across five Sequel II SMRT 8 M cells using the following parameters: diffusion loading, 30-hour movie, 2-h immobilization time, 4-h pre-extension time, 30−80 pM on plate loading concentration. CCS reads were obtained using PacBio CCS software (https://github.com/PacificBiosciences/ccs).

## Hi-C library construction

The chromosome conformation capture sequencing (Hi-C) libraries were prepared as described[74] with modifications during the isolation of nuclei and the final size selection of the Hi-C library. 1.5 g leaves were harvested and crosslinked using formaldehyde as described[74]. For the isolation of nuclei, crosslinked leaves were ground to a fine powder using a precooled (−80 °C) mortar and pestle. The powder was transferred to a 50 mL plastic tube and 30 mL homogenization buffer (10 mM EDTA, 10 mM Tris-Cl, 80 mM KCl, 0.5 M sucrose, 0.1 mM PMSF, 0.1% (v/v) 2-mercaptoethanol, 1 μg mL$^{-1}$ apronitin, 1 μg mL$^{-1}$ leupeptin and 1 μg mL$^{-1}$ pepstatin, pH 9.4) was added. The suspension was kept on ice and mixed using a 5 mL pipette. 2 mL of a buffer containing 10 mM EDTA, 10 mM Tris-Cl, 80 mM KCl, 0.5 M sucrose and 20 % (v/v) Triton X-100 (pH 9.4) was added. The sample was mixed by using a 5 mL pipette and kept on ice for 20 min. Nuclei were isolated in a cold room (4 °C) by filtration through Miracloth and Sefar Nitex filter placed in a funnel as described[74]. Nuclei were collected by centrifugation (3000 g, 15 min, 4 °C). The supernatant was discarded and the pellet was resuspended in 350 μL ice-cold NIB-PD[74] using wide-bore pipette tips. In cases where 350 μL of buffer was insufficient to completely resuspend the pellet, an additional 350 μL of ice-cold NIB-PD was added. A 2 mL Eppendorf tube was provided with 1 mL ice-cold NIB-PD cushion[74], and 350 μL nuclei suspension were slowly layered on top. Nuclei were purified by centrifugation (16,000 g, 1 h, 4 °C) and resuspended in 500 μL RE buffer[74] by swirling with wide-bore tips. Nuclei were collected by centrifugation (1900 g, 5 min, 4 °C) and resuspended in 150 μL 0.5% SDS using wide-bore tips. 50 μL aliquots were distributed into three 2 mL tubes using wide-bore tips and incubated at 62 °C for 7 min. Each tube was provided with 145 μL water and 25 μL 10 % (v/v) Triton X-100 and incubated at 37 °C for 15 min with 350 rpm shaking[74]. Restriction with DpnII and subsequent steps were as published. The size selection of the Hi-C library was performed using AMPure XP beads. Briefly, the final PCR reactions were pooled, and the DNA was precipitated by the addition of 1 volume AMPure XP beads following standard manufacturer's protocols (Beckman Coulter Inc., USA). Hi-C libraries were eluted from the beads using 27 μL EB buffer. The libraries were quantified and sequenced (paired-end, 2 × 100 bp) using HiSeq 2500 (Illumina, USA) sequencing platforms and standard protocols from Illumina (Illumina, USA)[74]. This work was conducted at the Leibniz Institute of Plant Genetics and Crop Plant Research (IPK) in Gatersleben, Germany.

## Haploid genome assembly construction and collinearity analysis

The CCS reads were assembled using hifiasm (v0.15.1)[24] with default parameters setting and the primary contigs were used for haploid assembly construction. In a highly heterozygous species like hop, most of the homozygous and identical sequences are collapsed in primary contig assembly, however, highly diverged contigs from heterozygous regions called as allelic contigs cannot be merged. This is evident from larger contig assembly size than the expected haploid genome size. The aim of constructing a haploid genome assembly was to remove such allelic contigs and assemble the remaining contigs into pseudomolecules. The pseudomolecule construction was carried out using the TRITEX pipeline[26,27]. Initially, the guide map was constructed by mapping the markers from hop genetic map against the primary contig assembly using minimap2 (v2.17)[75]. The Hi-C data was then used for identifying allelic contigs, chimera breaking and contig ordering to generate haploid pseudomolecules. The contig ordering involves manual editing wherein the Hi-C contact matrices were visualized using HiC-inspector[76] and rearranged based on Hi-C links. BUSCO (Benchmarking Universal Single-Copy Orthologs, v5.1.2[77,78]) analysis was used to assess assembly completeness using single copy orthologs from embryophyta_odb10 database. The HiC reads were mapped against the final assembly and

HiC contact matrix was visualized as a heat map using HiCPlotter[79] to assess contiguity.

The pseudomolecules of the haploid assembly were named as per *Cannabis sativa* genome assembly (cs10). This was done by aligning the transcript sequences from the *C. sativa* cs 10 genome[29] against the haploid hop assembly using gmap (downloaded on 2019-09-12)[80]. The maximum proportion of gene models from each *C. sativa* chromosome mapping on any given pseudomolecule of the haploid hop assembly was determined, and the corresponding number was assigned to the respective hop pseudomolecule.

The collinearity between the haploid assemblies of cv. Apollo (present study) and cv. Cascade was assessed by assembly-to-assembly comparison. First, the pseudomolecules from the Cascade assembly were aligned to the chromosomes of the haploid Apollo assembly using minimap2 (v2.24). The primary alignments with length ≥2000 bp were extracted and visualized as a dot-plot in R statistical environment[81].

## Generation of phased hop assemblies

We used hifiasm (v.0.16)[24], ALLHiC[25], and TRITEX[26,27] pipelines to generate a phased assembly of hop cv. Apollo. To reduce false positive duplicates that stem from bias due to low quality reads, CCS reads were extracted from PacBio subreads using ccs (v.6.0.0) software with more stringent parameters (--min-rq 0.99, --min-passes 3). The HiFi reads along with paired end Hi-C data were used to generate phased contig assemblies using hifiasm (v.0.16)[24] with default parameter settings. It has been reported that the contigs in phased contigs assembly are often mixed and must be separated as per the phase before using it for pseudomolecule construction[24]. The ALLHiC pipeline was therefore used to generate a phase separated contig assembly. Initially, the contigs from both the phases generated using hifiasm were merged. To separate phases, the gene models of the haploid hop assembly (see genome annotation section) were therefore aligned against the combined phased contig assembly using gmap and the top two hits (contigs) for each gene model were extracted. These contig pairs were considered allelic for that gene. This allelic contig information along with Hi-C data then served to partition contigs into two phases for each chromosome using the ALLHiC pipeline. The Hi-C contact matrices of contigs from both the phases for each chromosome were generated using "run-assembly-visualizer.sh" script from 3D-DNA pipeline[82] and visualized using Juicebox[83]. The contigs were manually separated into two phases based on Hi-C links. This involves visualization of Hi-C contact matrix heatmap using Juicebox and shuffling of contigs in such a way that to have contiguity along the diagonal. Finally, all the contigs from each phase were combined and pseudomolecule construction was carried out separately for each phase using TRITEX[26,27] pipeline as described earlier.

## Estimation of switch-errors in phased assembly

The phase switch error is defined as the incorrect assignment of single base from one phase to another. The percentage of switched bases in the phased assembly was determined following the calc_SwitchErr pipeline[25]. Briefly, the PE reads from a 450 bp PCR free library of Apollo were mapped to phase 1 of the assembly using minimap2 (v2.17) and variant calling was performed using bcftools (v1.11) pipeline. In parallel, the PacBio HiFi reads were also mapped against phase 1 of the assembly using minimap2 (v2.17) with parameter --secondary = no to select the best alignment for each read. The resulting BAM file was used for phasing the SNPs identified from PE read mapping using WhatsHap[84]. The phased SNPs with the 'PS' label were extracted for further comparison. This set is considered as truly phased SNPs. Further, the two phases of the assembly were aligned using minimap2 (v2.17), and the Pairwise mApping Format (PAF) file was converted to delta using paf2delta function from RagTag[85] which was further filtered using delta-filter from MUMmer (v4.0)[86] using parameters (-i 90,

-l 1000). The filtered delta file was then used for SNP calling using show-snps function from MUMmer (v4.0) with parameter -Clr. The SNPs from the phased assembly were compared with truly phased SNPs and the percentage of inconsistent SNPs were identified.

## Tandem repeat annotation and clustering into families

Tandem repeats were identified with TandemRepeatFinder (v4.09.1) using the default parameter settings[87]. The resulting output contains some overlapping annotations due to multiple reporting of repeats at different unit sizes if the longer units had a higher score (e.g., 20, 40, 80 unit length). Such overlapping annotations were removed by prioritizing longer annotation intervals and shortening the following smaller sized intervals. A representative unit sequence was derived from each consensus unit reported by TandemRepeatsFinder. This was necessary because the exact start positions can vary between tandem repeat stretches of basically identical sequences. A simple illustration would be the trimer unit TGC-TGC-TGC-... which gives by 1 bp shifts the three possible units TGC, GCT and CTG. The reverse complement GCA-GCA-GCA-... results to GCA, CAG, AGC. The first item of an alphabetically sorted list of all possible units was chosen as representative unit: AGC for the given example. Further, the representative unit sequences were clustered using vmatch dbcluster (http://www.vmatch.de/, https://anaconda.org/bioconda/vmatch, v2.3.0). For each one of the top 200 largest clusters their member locations were plotted in 500 kb bins along the chromosomes. For most of the clusters all of its members were distributed equally between the two phases, some clusters were centromere specific, and a few clusters showed a distinct opposite specificity for one or the other parent.

## K-mer based parent identification

The chromosomal locations of the above-described parental ancestor specific tandem repeat clusters were used to roughly partition the genome assembly into the two parental origins. To get a denser coverage with specific parental marker tags, these sequence bins in turn were used as input for the identification of high copy k-mer sequences which are enriched in one or the other parent. After determining the frequencies of all k-mers of size 20, 30, 40 and 50 with tallymer[88], a frequency filter of $\geq 400$ for k-mers derived from parent 1 sequence bins and $\geq 250$ for parent 2 was applied. Further, only k-mers with absolute $\log_2$-fold change value $\geq 4.25$ were used to assign chromosomal regions to one of the parents in three steps: (1) mapping of the $a_1$- and $a_2$-k-mer libraries to the assembly to annotate $a_1$ und $a_2$ specific locations (2) binning of each chromosome into 500 kb bins and calculation of the $\log_2$-fold ratio between the annotated $a_1/a_2$ base pairs per bin. (3) annotation of approximate coordinates (accuracy ~ bin size) for the parental origin by picking up relevant zero crossing from the $\log_2$-fold values.

The mapping in step 1 was done separately for each phase with vmatch allowing only perfect matches of the phase enriched k-mers. To make the matching process more efficient vmatch masking to a non-existing base 'Z' was used. Subsequently the annotation coordinates were extracted from the 'Z' base pairs. In step 2, each chromosome was partitioned into equally sized bins close to the given bin size to avoid edge effects. The original log2-fold ratio resulted in positive ($a_1$ bp > $a_2$ bp), negative ($a_1$ bp < $a_2$ bp), $\infty$ ($a_2$ bp = 0), $-\infty$ ($a_1$ bp = 0) or 0 ($a_1$ bp = $a_2$ bp) values. The remaining case $a_1$ bp = 0 and $a_2$ bp = 0 did not occur for bin sizes $\geq 500$ kb. For the visualization the $\pm\infty$ values were transformed to slightly above max and min values per sequence by int(max $\log_2$ fold + 0.5) + 0.1 or int(min $\log_2$ fold $-$ 0.5) $-$ 0.1 respectively. With the aim to filter for larger homogeneous stretches of parental origin in step 3, the zero crossings were only considered if the corresponding bins contained at least 50 bp of one $a_1$ or $a_2$ sequence tags. With the described bioinformatic approach it was possible to assign chromosomal regions of the Apollo genome assembly to their parental origin without the usual trio binning which needs additional sequencing data of the parents.

## Determination of identity between phases

The identity between homologous chromosomes of the phased assembly was determined using a similar strategy as described previously[89]. Briefly, the pseudomolecules from phase 2 were aligned to phase 1 pseudomolecules of the Apollo assembly using minimap2 (v2.22). Percentage of sequence identity was calculated for primary alignments with size $\geq 10$ kb using the following equation:

$$\begin{aligned} &\text{Percentage of sequence identity} \\ &= (\text{Number of mismatches}/\text{Lengths of alignment}) \times 100 \end{aligned} \tag{1}$$

The median percentage of sequence identity over 500 kb bin was plotted across the chromosomes using R (v4.3.1) statistical environment.

## Determination of the origin of phases in Apollo

A depth of coverage analysis[90] was carried out to determine the origin of phases in Apollo. For this, representative accessions from *H. lupulus*, *H. lupuloides*, *H. neomexicanus*, and modern cultivars were sequenced using genotyping-by-sequencing (GBS). The reads were trimmed to remove adapters and low-quality bases with minimum read length of 30 bp using cutadapt (v3.5 with Python 3.9[91]). Further, we observed high similarity between phases for chromosomes chr02, chr07 and chr08 and therefore a combined assembly was generated by using all the chromosomes from phase 1 and all but chr02, chr07 and chr08 chromosomes from phase 2 of the Apollo assembly. The high-quality reads from each accession were then mapped to the combined assembly of Apollo using minimap2 (v2.24)[75]. The output of minimap2 (v2.24) was piped to samtools[92] (v1.15) for its conversion to BAM format which later was sorted using NovoSort (v3.01.06). The number of reads in the 5 Mb window were calculated from the BAM file of each accession. Reads from each window were first normalized to sequencing coverage and then to the number of reads from the same window for Apollo. The logarithm of normalized count was plotted to create a genome-wide coverage plot in R (v4.0.2) statistical environment. For the top rows in Supplementary Fig. 10 the average values of the normalized counts per bin were calculated separately for four North American (Neomex, 14, 15, US wild 15, 16) and two European lines (Eu feral, 13 and 14) and displayed as log2-fold difference of the NAm/Eu ratio.

## Syntenic and orthologous framework for *Humulus* and *Cannabis*

To establish a syntenic and orthologous framework for cv. Apollo, the genome assemblies of two publicly available hop cultivars Cascade[13] and Saaz[36] and *Cannabis sativa*[49] were downloaded and used. We extracted genomic coordinates and selected the longest protein isoform at each gene locus as the representative sequence. Orthology inference, synteny detection, and all visualizations were performed with GENESPACE v1.3.1 using 10 CPU cores and otherwise default parameters. Plots were rearranged following the GENESPACE plotting tutorial (https://htmlpreview.github.io/?https://github.com/jtlovell/tutorials/blob/main/riparianGuide.html). All input and results, including R workspace files, are available[93] to ensure full reproducibility and facilitate reuse of the analysis.

## Identification and validation of structural variations (SVs)

The SVs relative to pseudomolecules of phase 1 of the Apollo assembly were identified as described[94]. First, the pseudomolecules from phase 2 were aligned to the chromosomes from phase 1 using minimap2 (v2.24). The paf file was converted to delta using "ragtag_paf2delta.py" script from ragtag[85] while the small and low-quality alignments were filtered using "delta-filter" function from MUMmer (v3.23)[95]. The delta

file was converted to tsv format using "show-coords" function from MUMer before SV identification using SyRI pipeline[96]. Further, the CPL, DEL, DUP/INVDP (loss) variants and the phase1 sequences in NOTAL and TDM were converted as Absence SVs (relative to phase 1). The CPG, INS, DUP/INVDP (gain) variants and the query sequences in NOTAL and TDM were converted as Presence SVs (relative to phase 1). The INV variants were regarded as inversions while the TRANS and INVTR were both regarded as translocation SVs. The SVs were visualized using the plotsr package[97].

To validate an 85 Mb inversion between the two phases of chromosome chr06, a PCR based approach was used. For this, five kb nucleotide sequences flanking the approximate initial and distal points of the inversions were retrieved. The coordinates for these points were 54,872,166 bp and 140,780,292 bp as well as 54,764,504 and 134,556,775 bp for the initial and distal points of phase 1 and phase 2, respectively. Next, the four retrieved sequences were aligned to each other using blastn[98], and primers were designed to specifically amplify a 2.107-kb fragment containing the inversion breakpoint on phase 1 (Phase1-ini06-F3137- TTTATGGCAAGGAAGCAGTCGTTT, Phase1-ini06-R5243- TGGAGGGAAAGTATTGCAAATGAGA).

PCR reactions were performed in a UNO thermal cycler (VWR, Denmark) using GoTaq G2 DNA Polymerase kit (Promega, Germany) following the manufacturer's instructions with gDNA extracted from dried leaf or cone material of accessions in the pedigree using the DNeasy Plant Mini kit (Qiagen, Germany) as template. The amplification program applied was as follows: 95 °C for 3 min; 30 cycles of 95 °C for 1 min, 55 °C for 1 min, 72 °C for 1 min and 30 s; and a final extension step of 72 °C for 5 min. Gel electrophoresis of PCR reactions was performed on a 2% agarose gel (Invitrogen, USA). The amplified fragment from Apollo DNA was once purified using GenElute™ PCR Clean-Up Kit (Merck, Germany) and the sequence was confirmed by Sanger sequencing through external service provider (Macrogen, Europe). A positive control to confirm isolation of intact DNA was performed using primer pair contig18 (F: TCATCAG-CAGGTGGGTCAGGCA and R: TCCGCACTTCTCTCACAGGCGA)[99] for each accession.

## Annotation of protein coding genes from haploid assembly

Evidence from transcriptome data as well as homology information from related species were incorporated in the Plant Genomics and Systems Biology (PGSB) gene annotation pipeline in a first step. For the homology-based gene annotation, protein sequences from Uniprot (21,802 proteins from Rosids in addition to *Fragaria vesca*, octaploid *Fragaria, Prunus persica, Rosa chinensis, Cannabis sativa*, and *Arabidopsis thaliana* (Tair10)) were combined into a database (AnnoDB) and mapped to the haploid assembly reference genome sequence using the splice-aware mapper GenomeThreader (v1.7.1[100], parameters: -startcodon -finalstopcodon -species rice -gcmincoverage 70 -prseedlength 7 -prhdist 4). As part of the evidence-based step, multiple RNA-seq datasets (Supplementary Data 5) were used for the genome-guided prediction of gene (exon-intron) structures. The RNA-seq reads were trimmed using Trimmomatic (v0.39.2)[101]. Quality control was performed with FASTQC (v0.11.9)[102] before and after trimming. All RNA-seq data were mapped to the genome reference with STAR (v2.7.8a)[103] parameter alignIntronMax = 50,000) and assembled into transcripts with Stringtie2 (v2.1.5, parameters -m 150 -t -f 0.3[104];). Transdecoder (v3.0.0) (https://github.com/TransDecoder/TransDecoder) was used to identify potential open reading frames (ORFs) and predict protein sequences. The predicted protein sequences were compared against a protein reference database (UniProt Magnoliophyta, reviewed/Swiss-Prot) using BLASTP (v2.13.0-max_target_seqs 1 -evalue 1e-05[105]) and hmmscan v3.3.2[106] was used to identify conserved protein family domains for all proteins. BLAST and hmmscan results were then processed by Transdecoder-predict and the best translation per transcript sequence was selected. Finally, results from the two gene prediction approaches were combined, and redundant protein sequences were removed.

Protein sequences with an open reading frame were then extracted from the annotation using gffread (v0.12.1[107];) and Biopython[108]. BLASTP (v2.13.0, e-value 1e-5[105,108];) was used to compare these to AnnoDB, and a database containing validated magnoliophyta proteins (UniMag) downloaded from Uniprot on 2017-02-20, and PTREP (release 19). PTREP contains hypothetical proteins with deduced amino acid sequences and was used to identify divergent TEs without significant similarity at the DNA sequence level. The two former databases were filtered for sequences containing both start and stop codons. Hits with a subject and query coverage above 80% were selected, and protein sequences that were complete and had a subject- and query-coverage above the threshold in the UniMag database or no blast hit in UniMag but in AnnoDB and not in PTREP were selected.

For the ab initio annotation of protein coding genes with AUGUSTUS[109], a hop model was first trained using available short read genomics data from hop (NRGENE, Ness Ziona, Israel). An AUGUSTUS (v3.4.0) prediction (options --UTR=on --alternatives-from-evidence=true --allow_hinted_splicesites=atac) was performed using the trained hop model. To avoid potential over-prediction, guiding hints were generated using transcriptome data, predicted protein sequences of HC gene.

## Annotation of protein coding genes from phased assembly

The annotation of protein coding genes from each phase of the haplotype-resolved phased assembly was carried out following aforementioned steps. Afterwards, EVidenceModeler9[110] (EVM, git commit 73350ce, partition options --segmentSize 900000 --overlapSize 50,000) was used to merge predictions and alignments from the non-redundant annotation based on the homology information and AUGUSTUS. All inputs were provided as gene predictions to EVM, and all inputs except from AUGUSTUS were also provided as protein alignment evidence to EVM. Evaluation of the different weight settings was performed using BUSCO (v5.3.2, eudicots_odb10 protein mode). The resulting BUSCO statistics served as the main guide for choosing which EVM weights to use. PASApipeline10[111] (v2.4.1) was used to update the annotation resulting from EVM to incorporate UTRs and splice variants. Confidence classification was performed on the final PASA pipeline output as described above. Several coverage thresholds were evaluated based on BUSCO results (v5.3.2, protein mode) and blastn11 (v2.9.0 +, options -max_target_seqs 1 -evalue 1e-10) hits against Iso-Seq transcripts, with a threshold of 65% set for both query and subject coverage across all three databases. Predicted proteins were classified into their confidence levels as described previously[48]. Representative transcripts for genes with several isoforms/splice variants were selected by performing a blast all vs. all search with blastp against AnnoDB. The best transcript was chosen based on coverage and identity. If no hit was found against the database, the longest transcript was selected. Functional annotation of transcripts as well as the assignment of GO terms, and Pfam- and InterPro-domains were performed with the tool Mercator[112].

## Gene model consolidation

To identify and consolidate potentially missed genes between both phases in the evidence-based annotation, we employed a mapping scheme similar to the method described in Jayakodi et al.[113]. Briefly, gene models were aligned to the genome using blast[114] (v34), exonerate[115] (v2.4) and gmap[80] (version 2021-08-25). Each match was subsequently scored by a pairwise protein alignment to the source model that triggered the match. Global alignments were generated using the AlignIO module of Biopython[108] with the blosum62 substitution matrix and gap open and extension penalties of 10.0 and 0.5, respectively. We only retained complete matches with a contiguous ORF and a start and stop codon. In addition, transfer was

limited to allelic matches complementing the gene annotation between homologous regions of the two phases. Such allelic blocks between phases 1 and 2 were derived from the initial evidence-based gene annotation using an all-against-all blastp comparison and McScanX[116] with default parameters. Starting from the highest scoring match, each block was progressively consolidated by insertion of missing allelic gene models if their coding region did not overlap any previously annotated CDS.

Finally, the consolidated gene models were corrected for potential chimeras and mis-calls of alternative splice variants using the Mikado pipeline[117]. The pipeline was run according to manual with standard parameters and SwissProt protein blast database filtered for Viridiplantae and Mikado pick command was run in reference-update mode.

## Transposon annotation

The transposon sequences from both the Apollo and the *Cannabis sativa* (cs10) (GCA_900626175.2) assemblies were annotated using EDTA[118] using default parameters. To determine the insertion age of each complete LTR-retrotransposon instance the divergence between its 5′ and 3′ long terminal repeat sequences (1-ltr_identity from the EDTA gff file) was used with the following formula:

$$Age = Distance/(2 \times Mutation\ rate) \times 10^{-6} \qquad (2)$$

The used random mutation rate was $1.3 \times 10^{-8}$ [119].

## Gene family expansion study

Gene sequence evolution and gene family expansion and contraction analysis was carried out using the following species: *Humulus lupulus* (phase1); *Cannabis sativa* (cs10; GCA_900626175.2); *Trema orientale* (GCA_002914845.1); *Morus notabilis* (GCA_000414095.2) *Ziziphus jujuba* (GCA_000826755.1); *Prunus persica* (GCA_000346465.2) and *Vitis Vinifera* (GCA_000003745.2). Initially, the coding sequences were extracted from the genome assembly of the respective species using the co-ordinates from GFF file. The transposable elements were predicted using RepeatModeler2[120] and Repeatmasker[121] and coding sequences with 90% overlap with the repeat sequences were considered as transposable elements and discarded from the following analysis. The longest transcripts of a gene were used to identify orthologous genes among phase 1 of hops and the six remaining species using OrthoFinder[122] with default parameters. Subsequently, we identified significant gene family expansions/contractions using the program CAFE5[123] among two sets of Orthogroups. The first set were orthologous groups with at least one copy of a gene in all species examined. The second set were orthologous groups with at least one copy in the three species of the Cannabaceae family (*Humulus lupulus*, *Cannabis sativa*, and *Trema orientale*). Rooted trees for the two sets that were used as input by CAFE5 were downloaded from Timetree of Life (https://timetree.org/)[124].

For all genes used within gene family expansion and contraction analysis (besides hops), annotation of gene ontology (GO) terms was conducted using eggNOG version 2.1.12[125]. GO enrichments were conducted to enable an in-depth biological interpretation of the inherent genetic content for each set of expanded orthogroups, while pooling genes for each significantly expanding orthogroup set and subsequently utilizing associated GO term data. For this, the R package 'topGO' v2.44.0 was used[126]. TopGO (topology-based Gene Ontology scoring) accounts for the topology of the hierarchical GO graph structure when testing GO terms for significance, facilitating the GO analysis when dealing with many GO groups. GO enrichment for all three domains Cellular Component (CC), Molecular Function (MF), and Biological Process (BP) was achieved based on count data using the 'weight01' GO graph algorithm and the Fisher's exact test, while setting the gene universe to all genes in the dataset (*Humulus* cv.

Apollo plus above-mentioned species). In addition, the parameter 'nodeSize' was set to 5, thus pruning the GO hierarchy from the terms which have less than five annotated genes. Upon completion of the analysis, the most relevant GO terms were isolated (test statistic $P < 0.05$) and enriched in significant genes.

## Phylogenetic analysis

The accession number and corresponding database for the sequences of Terpene synthases (TPSs), Polyketide synthases (PKSs) and aromatic Prenyltransferases (aPT) are given in Supplementary Data 9–11. Sequences were retrieved from the genome databases using BLASTp and selecting those with more than 50% coverage and 30% identity to functionally characterized TPSs, PKSs, and aPTs from other Rosid. Only sequences containing the pfam domains PF01397 and PF03936 for the TPSs, PF08392 for the PKSs and PF01040 for aPTs were included in the analysis. The evolutionary histories for each family were inferred separately from protein sequences by using the maximum-likelihood method based on the JTT matrix-based model[127]. Multiple-sequence alignments were made in MEGA X using the ClustalW algorithm with default settings; for the PKS tree only the amino acid sequence encoded by second exon of each gene was considered. The tree with the highest log likelihood was selected (−69400.37, −12299.76 and −19508.47 for the TPS, PKS and aPT tree, respectively). Initial trees for the heuristic search were obtained by applying the neighbor-joining method to a matrix of pairwise distances estimated using a JTT model. The analysis involved 198, 66 and 48 amino acid sequences for the TPS, PKS and aPT trees, respectively. All positions with less than 80% site coverage were eliminated (partial deletion option). In total, there were 462, 327 and 366 positions in the final data sets for the TPS, PKS and aPT trees, respectively. The statistical significance of each node was tested by the bootstrap method using 1000, 10,000 and 1000 iterations for the TPS, PKS and aPT trees, respectively. The evolutionary analyses were conducted in MEGA X[128]. Visualization of the trees was made through the interactive Tree of Life online tool[129].

To study the expansion of TPS between phases, the TPS regions were extracted from chr01 of both the phases. The regions were split into 20 kb bins and used for synteny analysis along with the TPS gene models. The synteny analysis and visualization was carried out as described in MCScan[130] (https://github.com/tanghaibao/jcvi/).

## Phylogeny of PKS gene families

The Apollo assembly PKS protein sequences, representing the HlCHS and HlVPS gene families, were used as queries to identify orthologous sequences in the genomes of *Cannabis sativa* (cs10 assembly), *Parasponia andersonii* (GCA_002914805.1), and *Trema orientalis* (GCA_002914845.1). tBLASTn was employed against each genome database using default parameters with an e-value threshold of $1 \times 10^{-50}$. For each significant hit, protein identifiers were retrieved and the corresponding amino acid sequences extracted. The collected protein sequences were aligned using MAFFT, a program optimized for multiple sequence alignments of amino acid sequences. Ambiguously aligned regions were subsequently removed using BMGE (Block Mapping and Gathering with Entropy) with default settings. The resulting curated alignment was then used for phylogenetic inference. Phylogenetic trees were reconstructed using FastTree via the NGPhylogeny.fr platform, employing maximum-likelihood estimation with 100 bootstrap replicates to assess branch support. This analysis elucidated the evolutionary relationships among PKS gene family members across the four species.

## Mapping populations and sequencing

Three different bi-parental mapping populations at F1 generation were used for the construction of genetic maps:

1. The Apollo × PubM_740 mapping population of size 184 was produced in 2012 by crossing Apollo (female) with PubM_740 (male).

Approximately 50 mg fresh leaf material from each individual progeny was collected and dried on silica gel and transferred to the service provider (LGC Genomics, Germany) for DNA extraction and whole genome re-sequencing (WGRS) using 150 bp PE chemistry (Illumina, USA) at ~5 × coverage of diploid hop genome ($2n = 5$ Gb). In the case of Apollo, PCR free library with an average insert size of 470 bp was created and sequenced using 250 bp PE WGRS method at external service provider (NRGene, USA).

2. The Zenith × USDA21058M mapping population of size 128 was generated from a cross of Zenith (female) with USDA21058M (male) and sequenced using genotyping-by-sequencing (GBS) approach[45]. The genotypic data was downloaded from NCBI (BioProject ID PRJNA906612) while BLUPs estimates of phenotypic data form powdery mildew resistance (PMR) was downloaded from Havill et al.[45].

3. The Cascade x HL-19-060-002M mapping population of size 182 was generated from a cross of Cascade (female) with a *Humulus lupulus* HL-19-060-002M (male). Seeds were germinated and one leaf disk per two-week-old seedling was shipped to the service provider for DNA extraction and normalized-Genotyping by Sequencing (n-GBS) (LGC Genomics, Germany).

### Diversity panel
A diversity panel consisting of 243 samples was used for population genomic analysis. DNA extraction from young leaf tissue of each sample followed by sequencing using the *ApeK*1 restriction enzyme-based GBS approach was carried out externally (LGC Genomics, Germany). The GBS libraries were sequenced using 100 bp single end (SE) chemistry on Illumina sequencing platform at LGC Genomics, Germany.

### Variant calling
The reads from each sample were trimmed to remove adapters and low-quality bases with minimum read length of 30 bp using cutadapt[91] (v4.2 with python 3.9). The high-quality reads from individual accession were then mapped to phase 2 of the Apollo genome using minimap2 (v2.24)[75]. The output of minimap2 was piped to samtools[92] (v1.15) for its conversion to BAM format which later sorted using NovoSort (v3.01.06) and converted to CRAM using samtools[92] (v1.15). For WGRS data, duplicate reads were marked using "-md" command from NovoSort (v3.01.06). Variant calling was performed separately for each population using mpileup with minimum read quality (-q) cutoff of 20 and call function with -mv parameter from bcftools[131] (v1.15). The variant files were further filtered to remove insertion and deletions (InDels) and only bi-allelic SNPs with quality (Q) value > 25, depth ≥4 and less than 20% missing data were selected using vcftools[132] (v0.1.13).

For accessions from diversity panel, the barcoded GBS reads for each accession were extracted from sequencing file using the "fastx_barcode_splitter.pl" script from FASTX-Toolkit allowing one mismatch and variant calling was performed as mentioned above. The variant files were further filtered to remove insertion and deletions (InDels) and only bi-allelic SNPs with quality (Q) value > 25, depth ≥ 2 and less than 20% missing data were selected using vcftools[132] (v0.1.13).

### Linkage map construction
Linkage maps were constructed using Apollo × PubM_740 (186 progenies), Zenith × USDA21058M (130 progenies), and Cascade × HL19-060-002M (184 progenies) populations following pseudo-test cross strategy[133]. In the pseudo-test cross, markers heterozygous in one parent while homozygous in the other parent and showing 1:1 segregation are used for linkage map construction. Theoretically, two maps, one for each parent, can be constructed. However, out of six parental lines Apollo, USDA21058M, and Cascade were interspecific hybrids, while PubM_740, Zenith and HL-19-060-002M were Eu lines (Supplementary Fig. 11). We therefore couldn't identify sufficient markers for

map construction in pure Eu lines and only one map could be constructed for each population.

For the Apollo × PubM_740 population, SNPs with ≤ 5% missing data while for Zenith × USDA21058M and Cascade × HL19-060-002M populations SNPs with ≤ 20% missing data were used for linkage map construction following pseudo-test cross strategy. These filtration criteria were selected depending on the short-read sequencing strategy used for sequencing these populations. Briefly, the vcf2popNew.1.0.py[134] module from VcfHunter pipeline was used with a threshold of $P = 1 \times 10^{-5}$ to identify SNPs heterozygous in either parent and showing 1:1 segregation in F1 progenies. Later, a maximum of 1000 segregating SNPs from each chromosome were randomly selected and genetic maps were created for each chromosome separately using Rqtl[135]. Separate linkage maps were created for each parent of a population. Briefly, the marker data for each parent from a population was imported using the read.cross function and setting crosstype to "bc". The distorted markers ($P = 1 \times 10^{-4}$) were removed and linkage map was constructed by setting maximum recombination fraction of 0.35 and minimum LOD score of 10. The graphical genotypes of markers used for linkage map were generated to study recombination patterns in each population.

### Phenotyping for downy mildew resistance (DMR)
Female and male individuals of the Apollo × PubM_740 population grown in the Hallertau growing area, were inoculated and assessed as described before[136]: Disease severity was evaluated using a standardized ordinal scoring system according to the German Federal Plant Variety Office (Bundessortenamt, 2000), based on the percentage of infected leaf area. Infection levels were classified into five categories: 1 (highly tolerant, no sporulation), 3 (tolerant, 1–20% of leaf area infected), 5 (moderately infected, 21–50%), 7 (susceptible, 51–80%), and 9 (highly susceptible, 81–100%).

### Absolute quantification of alpha and beta bitter acids
Fresh inflorescences (cones) from 137 female individuals of the Apollo × PubM_740 population grown in the Hallertau region were collected across two crop years (2018 and 2021). Sampling was performed at two time points per year: in 2018, on 06-09-2018, and 19-09-2018 (two replicates per time point) and in 2021 on 09-09-2021 on 23-09-2021 (two replicates per time point). All samples were collected on dry ice and stored at −80 °C until further processing. Frozen hop cones from each sample were dried separately at 55 °C for 12 h using a Dörrex dehydrator (Stöeckli, Switzerland) and subsequently ground using a Tube Mill 100 control (IKA, Germany). Ground cone material was collected in 50 mL falcon tubes and stored at 4 °C until chemical analysis. To determine moisture content after drying, ~2 g of plant material was transferred into a glass beaker, covered with gauze, and fully dried over night at 105 °C using a drying and heating chamber (Binder, Germany). Hereby, the determined weight loss was used to determine the residual moisture content after drying. For chemical analysis, 50 ± 2 mg of plant material was transferred into a 1.5 mL glass vial together with 1 mL 50% acetonitrile, while subsequently being sonicated for 30 min using a CPXH Series Ultrasonic Cleaning Bath (Branson Ultrasonics, USA) to extract hop cone metabolites for each sample. Extracts were centrifuged for 3 min at around 1500 $g$ and 50 μL solution transferred to 150 μL 50% acetonitrile already prepared in a well of a 96-microfilterplate (0.2 μm polyvinylidene fluoride membrane, Agilent, USA), which is subsequently centrifuged for 3 min at around 1500 $g$. Filtered extracts were transferred into a new 1.5 μL glass vial containing a glass insert and immediately applied to HPLC-UV/Vis analysis. For this, 10 μL of sample was injected into a 1260 Infinity II LC-System (Agilent, USA) with a Photo Diode Array detector (360 nm) fitted to a Kinetex (2.1 × 100 mm, 1.7 μm, XB-C18) column (Phenomenex, USA). Two eluents were used; eluents A (ddH$_2$O + 1% CH$_3$COOH) and B (acetonitrile + 0.1% ortho-phosphoric acid) were graduated with

20% eluent A and 80% eluent B for the first 4 min, following with increased eluent B proportions of 85% (2 min), 90% (2 min), 95% (1 min) and final decrease in eluent B to 80% for the remaining 2 min, with a total run of 11 min. The column temperature was kept at 25 °C, while samples were kept at 15 °C until analysis. Data were collected using OpenLab CDS ChemStation version 2.3.54 software (Agilent, USA). All samples were run over multiple batches of maximal 32 samples each to avoid signal loss, which was observed for bitter acids over time. Using the R (v4.2.1) package TIGERr version 1.0.0[137], generated signal intensity values for each targeted compound (co-, n- and ad-humulone and lupulone analogs) were corrected for batch effects based on a representative quality control pool sample that was run every 10 samples as well as before and after every sequence. Absolute quantification of alpha and beta bitter acids was achieved using internal standard curves applied to each batch using ICE-3 standard as reference (LaborVeritas, Zurich, Switzerland). Eventually, metabolite concentration as percentage of mg dry weight was calculated using the following equation:

Metabolite concentration as percentage of mg dry weight

$= $ Determined compound concentration(mgL$^{-1}$)

$\times$ Dilution factor in mL(0.004)/(Sample weight(mg) $\times$ dry weight(%))

$$(3)$$

The best linear unbiased prediction (BLUP) values were estimated from the replicated data using following model:

$$\text{Parameter} \sim (1|\text{PlantID}) + (1|\text{SEASON}) \qquad (4)$$

using lmer package in R (v4.0.2) statistical environment and used as input phenotypes for downstream GWAS analyses.

## Genome-wide association study (GWAS)
GWAS was carried out using linear mixed model (LMM) function from gemma[138] (v0.98.5) (for disease resistance and α-acid content using phase2 as reference) or GAPIT (version 3) using BLINK model[139] (for β-acids and α-acids using phase 1 as reference) using SNPs with less than 10% missing data and with 1% minor allele frequency (MAF). The kinship matrix for each population was generated based on the IBS values calculated using snpgdsIBSNum function from SNPRelate package[140] in R statistical environment. The results were visualized by generation of Manhattan plots using qqman R package[141] for each trait. Significant marker test associations (MTAs) were identified using Bonferroni correction at significance level of 0.05.

## Identification of k-mer based introgression
GWAS for bitter acid content identified an introgression on chr08. A k-mer i.e., unique stretch of DNA sequence of length "k", based approach was used to identify the introgressed region and study their segregation in Apollo × PubM_740 population. For this, the k-mers of size 31 bp specific to the introgressed region were identified from the phased assembly of cv. Apollo using KMC tool[71] (v3.2.1). The diagnostic k-mers were then searched against the quality checked reads of each individual separately and reads carrying k-mers were counted. The proportion of reads carrying the k-mers were plotted against the total number of reads to study segregation of introgressed region.

## Population genomics analysis
The principal component analysis (PCA) was performed using the FastPCA algorithm[142] implemented in snpgdsPCA function of the R package SNPRelate[140] (v.1.24.0). The population structure analysis was carried out using ADMIXTURE[143] (v.1.3.0). The program was run for different K values starting from 2 to 10 with tenfold cross-validations and 500 bootstrap replicates at each run. Nucleotide diversity and Fst

analyses were carried out using vcftools[132] (v0.1.13) and SNPRelate (v.1.24.0) to study within and between population diversity, respectively.

The chromosome-wise depth of coverage analysis was carried out using the GBS data of cultivars and breeding lines as described above (Determination of origin of phases in Apollo). The coverage information was used to determine presence/absence of particular phases on chromosomes in the given sample. The information was later used to determine proportion of chromosome combinations in the entire pool of breeding lines and cultivars used in the present study.

## Chemo-transcriptional analysis of cone development in cv. Apollo
Preparation of plant material cv. Apollo was grown in a hydroponic system in a climate chamber set to 16 h daylength for eight weeks to induce vegetative growth. After 8 weeks, daylength was reduced to 13 h to induce flowering. Hop cones were harvested and pooled in biological triplicates at five distinct developmental stages (Flowering, 1, 2, 3 and 4 weeks after flowering (WAF)), a staging scheme previously defined by Wang et al.[53]. Collected cones were immediately flash frozen in liquid nitrogen and stored at −80 °C. To resolve the downstream analyses towards glandular trichomes, hop cones were disrupted by stirring in liquid nitrogen using a spatula following sieving of material through a double layer of metal sieves (500 μm mesh size) multiple times until a glandular enriched fraction and residual bract/bracteole material (bract fraction) was achieved.

## MALDI mass spectrometry imaging (MALDI-MSI)
For sectioning for MALDI-MSI analyses, hop cones at stage 2 WAF were collected fresh and cut in half along their central axis (strig) immediately prior to embedding in 2% aqueous CMC (Sigma-Aldrich, USA), carried out as previously described[144] and adapted to plants[145] with a few modifications. In brief, the procedure was as follows: The embedding container from SECTION-LAB Co. Ltd., Japan, was 3.5 cm × 2.5 cm, and filled one-third with the CMC solution. When the bottom layer was frozen by partial submersion of the embedding container in a n-hexane/dry ice mix, the half cones were carefully inserted with the cut part facing up and fully covered with CMC. Due to the high viscosity of CMC solution, a syringe was used to apply CMC between the tightly packed bracts and bracteoles. The embedding container was then fully submerged in the n-hexane and dry ice mixture until completely solidified. Blocks with embedded plant material were stored at −80 °C until tissue sectioning. Embedded hop cones were cryo-sectioned on a Leica CM3050S Cryostat at −30 °C using the Kawamoto film method[144] likewise adapted for plant samples[145]. The frozen CMC blocks with embedded plant material were mounted on the pre-cooled sample stage using OCT (Optical Cutting Temperature) Mounting media (VWR International, Randor, USA), and sections cut at 10 μm thickness onto adhesive cryofilm (type 3 C [16UF] 3.5 cm (SECTION-LAB Co. Ltd., Japan), which was mounted on glass slides using double-sided carbon tape (Nisshin EM Co., Ltd., Tokyo, Japan). The sections were freeze-dried and kept in a vacuum desiccator until matrix application. The matrix 1,5-diaminonaphtalene (DAN) (Sigma-Aldrich, Burlington, USA) was applied by sublimation, using custom built glass chamber for sublimation[145], for 5 min at 150 °C. The matrix-coated samples were stored under vacuum until MSI analysis, carried out on the same day. MALDI-MSI was performed on a Thermo QExactive Orbitrap mass spectrometer (Thermo Scientific, Bremen, Germany), equipped with an AP-SMALDI5 ion source (TransMIT GmbH, Germany), in negative ion mode, using a scan range of $m/z$ 60–900 and pixel size of 65 μm. All samples were analyzed in duplicates, and all shown compounds were identified based on the accurate $m/z$ value of their deprotonated forms [M-H]- ± 10 ppm to the assigned $m/z$ value, using MSiReader[146] (v1.01) (Supplementary Fig. 28). Signal intensities were normalized to the total ion current (TIC).

## Volatile compound analysis

Approximately 25 mg (±9 mg) of frozen bract and gland samples over five developmental stages, collected and prepared as described above (Chemo-transcriptional analysis of cone development in cv. Apollo) were extracted in 1 mL 50% acetonitrile (+10 ppm 1-hexanol as internal standard) under sonication for 30 min at room temperature using a CPXH Series Ultrasonic Cleaning Bath (Branson Ultrasonics, Brookfield, USA). Extracts were subsequently centrifuged at around 1500 $g$ for 3 min, followed by transferring of 25 µL supernatant into a 20 mL glass vial. To minimize extract exposure to air and potential oxidation of compounds, all sample vials were purged with Argon gas before analysis.

A dynamic headspace sorptive extraction (DHS) combined with Thermal Desorption Unit (TDU) and coupled with gas chromatography-mass spectrometry (GC/MS) system with Pulsed Flame Photometric Detection (DHS-GC-MS/PFPD) was used to isolate aromatic compounds from complex mixtures. The DHS-GC-MS/PFPD analysis was performed with a thermal desorption unit (TDU) equipped with a Multi-Purpose Sampler autosampler with a Peltier-cooled sample holder. The GC is equipped with a GERSTEL (Gerstel GmbH & Co. KG, Germany) Cold Injection System (CIS4) programmed temperature vaporization (PTV) inlet, controlled by a Cryostatic Cooling Device installed in an Agilent 7890B gas chromatograph (Agilent, CA, USA) with a 5977 single quad Mass Spectrometry Detector and a Pulsed Flame Photometric Detector (Xylem OI-Analytical, USA). The DHS-GC-MS/PFPD configuration was equipped with an Agilent capillary flow technology (CFT) two-way splitter (with makeup gas line), which was controlled with an auxiliary pressure control module (PCM). The sample was extracted via DHS by purging the sample with 300 mL carrier gas at 55 °C onto a Tenax-TA trap (Buchem BV, NL) kept at 21 °C during trapping phase, followed by thermally desorbing the trap in the TDU by programming a temperature ramp from 15 to 230 °C (held for 3 min) at a rate of 7.2 °C per minute, while maintaining a 48 mL min$^{-1}$ desorption flow. Desorbed compounds were focused at 2 °C on a liner packed with Tenax-GR in the cooled PTV inlet for subsequent analysis. After desorption, the PTV inlet was programmed from 2 to 240 °C (held for 7 min) at 12 °C sec$^{-1}$ to inject trapped compounds onto the analytical column. The injection port was operated in split mode at a 1:10 rate, furthermore the setup with two detectors provided a distribution of 80% sample to the MSD and 20% to the PFPD. Separations were performed on an Agilent 122-7032UI DB-WAX column (30 m × 0.25 mm i.d., 0.25 µm film thickness) that utilizes He as the carrier gas. Following oven method was applied: 40 °C for 4 min, ramp to 165 °C at 6 °C min$^{-1}$, ramp to 230 °C at 30 °C min$^{-1}$, and hold for 5 min. The MS used electron impact (EI) ionization with the ion source voltage and temperature set to 70 eV and 300 °C, respectively. A solvent delay of 3 min was applied to avoid mass detection during solvent depletion.

MSConvert version 3.0.2 was used to convert the generated spectral data into mzML format[147]. Raw spectral data pre-processing was done using MZmine2[148] (version 2.5.3), with a noise cutoff of 1E3 signal intensity (SI) during mass detection followed with extracting ion chromatograms with at least three consecutive scans of 1E3 SI under a $m/z$ tolerance of 0.1 (or 5 ppm) using the 'ADAP chromatogram builder' tool. Individual peaks were deconvoluted from the generated chromatograms using the local minimum search with a minimum absolute peak height of 5E3 SI and chromatographic threshold of 95%. Subsequently, pseudo-fragmentation spectra with a minimum size of two ions were extracted during spectral deconvolution based on hierarchical clustering and the sharpness model. Peak list alignment was done using the 'ADAP Alignment' tool by applying a retention time tolerance of 0.1 min and $m/z$ tolerance of 0.1 (or 10 ppm). Isolated spectral features were matched to an in-house spectral database that includes spectra of 17 commercially sourced reference VCs (Merck, Darmstadt, Germany) (Supplementary Data 15) which were analysed

under the same analytical setup and considered present upon cosine score >0.7. Finally, spectral features found in blank samples (empty vial) as well as the internal standard were removed before exporting the peak list to csv and msp format. The latter was used to search the peak list against the NIST20 library using the MSPepSearch tool version 0.9.4.9 (National Institute of Standards and Technology, Gaithersburg, MD, USA). A detailed description of the spectral pre-processing parameters used can be found in Supplementary Table 16. Spectral data post-processing was conducted using R script and involved normalization of spectral data to median sample weight and internal standard signal intensity. The script is available in the Code Availability section.

## Non-volatile compound analysis

14 mg (±11 mg) samples of the bract and gland fraction derived from above-described five developmental stages were transferred into 1.5 mL glass vials and extracted in 50% acetonitrile (+4 ppm forskolin as internal standard) under 30 min sonication, while using a CPXH Series Ultrasonic Cleaning Bath (Branson Ultrasonics, Brookfield, USA). Acetonitrile extracts were filtered using a 0.2 µm 96-well filter plate (Agilent, Santa Clara, USA) and analysed in negative and positive ionization mode utilizing a LC-qToF-MS/MS analytical system[149]. Hereby, measurements were performed using an Ultimate 3000 UHPLC+ Focused system (Dionex Corporation, Sunnyvale, CA, USA) coupled to a Bruker Compact ESI-QTOF mass spectrometer (Bruker, Billerica, MA, USA). Chromatographic separation was achieved on a Kinetex XB-C18 column (100 × 2.1 mm, 1.7 µm particle size, 100 Å pore size; Phenomenex, Torrance, CA, USA) maintained at 40 °C, using a flow rate of 0.3 mL min$^{-1}$. The mobile phases consisted of water with 0.05% (v/v) formic acid (solvent A) and acetonitrile with 0.05% (v/v) formic acid (solvent B). The gradient program was: 0–1 min, 10% B; 1–23 min, linear increase to 100% B; 23–25 min, 100% B; 25–25.5 min, decrease to 20% B; and 25.5–30.5 min, re-equilibration to 10% B. Mass spectra were acquired in positive and negative electrospray ionization mode over an $m/z$ range of 50–1200. Instrument settings were as follows: capillary voltage 4000 V, end plate offset 500 V, dry gas temperature 220 °C, dry gas flow 8 L min$^{-1}$, nebulizer pressure 2 bar, in-source CID energy 0 eV, hexapole RF 50 Vpp, quadrupole ion energy 4 eV, and collision cell energy 7 eV. For MS/MS analysis, data were acquired in an untargeted manner using a collision energy of 27 eV. For quality control, a pooled sample representing the spectral set as well as an extraction solvent blank was run after every 10 samples as well as before and after the sequence. As presented in a previous study[149], pre-processing of raw spectral data was conducted using MZmine version 2.5.3 using a slightly adjusted workflow (Supplementary Data 16). For this, spectral features were identified at level 1 ($m/z$ and retention time match, MS2 similarity above cosine 0.7) using an in-house chemical database composed of 24 commercially sourced reference compounds (Merck, Darmstadt, Germany), which were analysed on the same analytical system (Supplementary Data 17). The same spectral library was further used during feature-based molecular networking (FBMN) workflow that was adapted from a previous study as well[149], to support level 2 chemical identification (MS2 similarity above cosine 0.7). For this, spectral data of reference compounds was made available for public use at the Global Natural Products Social (GNPS) platform under the CCMSLIB IDs given in Supplementary Data 17. The FBMN pipeline used in this study is also hosted by GNPS, which further provides a chemical classification toolbox[150,151]. Herein, network annotation propagation (NAP) was used to augment the insight into the chemical nature present in the observed chemical space of hop cone development[152]. In addition, a hop-specific in-house in silico fragmentation database (ISDB) was used that contains 265 relevant metabolite structures that have been characterized in hops and related species[153–156]. The in silico fragmentation-based dereplication results were categorized by their reliability, with SMILES from Fusion,

Consensus, and MetFrag algorithms corresponding to level 3a, 3b, and 3c identification, respectively[149]. Eventually, FBMN and NAP outputs were joined via the MolNetEnhancer version 22 workflow[157], while both spectral sets (derived from negative and positive ionization mode) were eventually merged using the 'merge networks polarity' tool version 22.1 from GNPS (Supplementary Figs. 29–32). All molecular networks were visualized using Cytoscape software[158] (version 3.8.2) and detailed parameters were listed in Supplementary Table 17. To reduce the number of chemical features for downstream correlation analysis, all subnetworks with at least five nodes were isolated, comprising 45 subnetworks with 583 features in total (of 987). Hereby, for each subnetwork the median of signal intensities derived from all nodes for a given developmental stage was taken to represent the abundance of a subnetwork's chemical features throughout hop cone development (Supplementary Fig. 33).

## RNA isolation and sequencing

For extraction of RNA from developing cones, fractions of enriched bract and trichomes were macerated using a Geno/Grinder. The extraction was performed using a modified CTAB extraction protocol[159]. 1 mL of extraction buffer (ES) was added to each fraction of macerated tissue and incubated for 15 min while occasionally vortexing. The samples were then centrifuged for 10 min at 14,000 g. The supernatant was transferred to a new tube and 500 μL of chilled isopropanol was added. The samples were stored at −20 °C overnight and then precipitated at 14,000 g at 4 °C for 10 min. The liquid was discarded, and the precipitated nucleotides were resuspended in 100 μL of RNase free water. The suspension was further cleaned up and concentrated using Zymo RNA Clean & Concentrator kit with the included DNase treatment steps. Purified RNA was stored at −80 °C. For the Apollo cone development dataset, non-stranded, poly-A tail selected RNASeq library preparation and sequencing using PE100 (2 × 101 bp) chemistry on DNBSEQ platform (MGI, China) was conducted by BGI TECH SOLUTIONS (Hongkong) and 8 Gb clean data was generated for each sample. Gland material from ten selected female individuals of the Apollo × PubM_740 population were processed as described above and isolated mRNA sequenced at Macrogen (Seoul, South-Korea), using a TruSeq Stranded mRNA library that was processed on a HiSeq 2500 sequencer (Illumina, USA) generating paired-end reads of 150 bp in length.

## Differential gene expression (DEG) analysis

Raw RNA-seq reads were processed using a previously described approach[160], which at first uses Rcorrector[161] (v1.0.4) to remove erroneous k-mers from Illumina paired-end reads, including removing read pairs with at least one unfixable error. Further, adapter and low-quality bases with qualities below phred 5 were trimmed using TrimGalore![162] (v0.6.6). To ensure full depletion of reads originating from rRNA, reads were mapped against an rRNA database that was based on concatenated archaeplastida SSUParc and LSUParc fasta files derived from the SILVA repository[163] using bowtie2[164] (v2.4.4). Reads that passed the previous steps were examined regarding quality metrics using FastQC[102] (v0.11.9) (Supplementary Fig. 34).

Differential expression analysis was adapted from the Trinity/Trinotate pipeline[165,166]. Transcript quantification from each sample was done using Salmon[167] (v1.9.0) using the decoy aware transcript index built using pseudomolecules from phased assembly as decoy sequences (index with option –keepDuplicates). As part of the trinotate pipeline, the generated gene counts matrix was processed within the 'DESeqDatasetFromMatrix' function (default settings) of the R package DESeq2[168] (v1.36.0) (design = -stage) to identify differentially expressed genes (DEG) that are at least 4-fold ($\log_2$ fold change $\geq 2$) differentially expressed and backed up with Benjamini-Hochberg FDR-adjusted $P$ values of smaller or equal than 0.001 in any of the pairwise sample comparisons, while adjusting for expression dispersion among given biological replicates[169].

To extract groups of similarly expressed genes during hop cone development, TPM (Transcripts Per Million) normalized gene counts were further cross-sample normalized using the Trimmed Mean of M-values (TMM) approach to adjust for any differences in sample composition, facilitated by the 'edgeR' R package[170,171] (v3.42.4). The generated TPM-TMM normalized expression count data of a total of 3612 DEG was applied to hierarchical cluster analysis. For this, data transformation to a base-2 logarithm with a pseudo count of +1 was performed, followed by centering each row (individual genes) to its respective mean. For hierarchical cluster analysis a distance matrix was created based on the transformed expression dataset, for which a centered Pearson correlation (1 − corr) metric was applied using the 'amap' R package[172]. A Pearson based distance metric was preferred to emphasize on correlated expression profiles rather than individual events within the time course experiment. Hierarchical clustering was conducted using 'Ward.D2' as agglomeration method to shape the cluster tree by utilizing the 'hclust' function from R package 'stats'[81], Cluster detection was done utilizing the 'dynamicTreeCut' R package[173] using the 'cutreeDynamic' function (minClusterSize = 100 and deepSplit = 2) that generated 4 DEG clusters (Supplementary Table 18). Dynamic tree cutting was tested by applying the average silhouette method, which assesses the overall quality of the clustering result by measuring how well an observation is clustered and further estimation of the average distance between clusters[174]. Optimal numbers of generated clusters were based on calculated maximum average silhouette values by using the 'cluster' R package[175] (Supplementary Fig. 35). Silhouette and cluster plots for the final clustering result were generated using 'factoextra'[176], 'cluster'[175] and 'ggplot2'[177] R packages. Generated clusters show groups of similarly expressed DEGs during hop cone development.

A thorough assessment of allele-specific expression (ASE) for homo- and heterozygous genes in corresponding DEG clusters was conducted based on gene ortholog information generated as described in detail below. For each DEG cluster, ASE was described in three categories: 'het category 1' – heterozygous (het) genes that are solely present in one phase; 'het category 2' – ortholog het genes that were found present for both phases; 'het category 3' – ortholog het genes that are further significant differentially expressed between both phases ($p < 0.05$), while showing an expression above 1 TPM in at least one stage. Correlation of DEG cluster information to determined metabolic profiles was based on $\log_2 +1$ transformed gene expression (after HCA quality control) and chemical data, with the latter comprising 45 NVC subnetworks and 49 VCs. Hereby, a positive correlation was accounted for if a chemical feature was showing a Pearson coefficient above or equal to 0.9 in at least 5% of DEGs of a given cluster (Supplementary Table 19). For this, correlation analysis was conducted using the cor() function from the 'stats' R package[81]. The resulting gene counts for each positive correlation event between a DEG cluster and a compound were used to generate a representative network using Cytoscape software[158] (version 3.8.2).

## Genome-wide allele specific expression analysis

Allele specific expression (ASE) analysis was carried out using ortholog gene pairs identified between the two phases of Apollo assembly. The gene pairs with 100% identity and coverage were classified as homozygous while heterozygous ortholog pairs of genes were allowed to have 5% sequence divergence as well as up to 0.05 tolerance in gene length based on the ratio of $\log_2$ transformed gene lengths for each pair. For the entire genome, 38913 genes were found within determined ortholog pairs, with additional 537 cases of ortho tandem repeats giving rise to 19,725 ortholog gene pairs overall. Hop cone development-derived replicated expression data of cross normalized TPM counts was pulled down for each ortholog gene pair and subsequently applied to differential expression analysis using R package DESeq2 v1.36.0 (design -stage+phase) to isolate gene pairs that differ

significantly in expression (adjusted $P < 0.05$) during cone development. Mean replicate expression data from hop cone development was used to create a TPM cutoff based on the resulting set of 38,913 genes. Hereby, the highest TPM value for each gene during cone development was isolated and the 1st quantile used as the TPM cutoff (1.05 TPM) to further refine the dataset to 24172 genes (6740 homozygous and 17432 heterozygous genes after removal of ortho tandem based redundancy in favor for heterozygous ortholog pairs) to be evaluated for a genome wide presence of ASE. To determine the proportion of expressed genes for every chromosome in the context of ASE, a counting scheme was implemented as follows: homozygous genes get each a point if expressed above TPM cutoff, heterozygous genes with significant ASE get a point for the dominant phase, while non-significant heterozygous genes expressed above TPM cutoff are treated like homozygous genes. Finally, the summed count was divided by total count of ortholog genes present in each chromosome and phase, respectively. Subsequently, two-proportions Z-tests were conducted on binary counts representing ASE within each chromosome and phase utilizing the prop.test() function from the 'stats' R package[81]. Thereby, $P$ for each two-proportions Z-test conducted between the phases of each stage as well as combination of all stages were calculated under the Yates continuity correction (Supplementary Table 20). To get a stage-resolved insight into ASE, the same set of 19725 ortholog gene pairs was utilized for a differential expression test in DESeq2 as described above while comparing phases for each stage separately (design = ~phase). Intersections of gene pairs that showed significant ASE (adjusted $P < 0.05$) were visualized for all possible stage combinations using an UpSet plot by applying the R package 'UpSetR'[178] (Supplementary Fig. 36).

### Reporting summary

Further information on research design is available in the Nature Portfolio Reporting Summary linked to this article.

## Data availability

The sequencing data generated in this study have been deposited in the European Nucleotide Archive (ENA) database under accession codes PRJEB64593 (scaffold sequence used for training AUGUSTUS gene annotations), PRJEB64169 (contig sequences and AGP file of the haploid cv Apollo assembly), PRJEB63995 (CCS, RNA-Seq, iso-seq and Hi-C reads used for the generation and annotation of the phased cv. Apollo assembly), PRJEB64122 (RNA-Seq data from the developing Apollo cone), PRJNA1082089 (phased assembly of hop cv Apollo), PRJEB63565 (linkage mapping Apollo × PubM_740 population), PRJEB63534 (linkage mapping Cascade × HL-19-060-002M population), PRJEB63136 (GBS data used for population genomic study). The annotation of the phased cv. Apollo assembly has been deposited to Zenodo [https://doi.org/10.5281/zenodo.18787309]. All input and results for the syntenic and orthologous framework for *Humulus* and *Cannabis* were deposited to Zenodo [https://doi.org/10.5281/zenodo.18802928]. The Raw DHS-GC – MS and LC-qTof-MS/MS spectral data generated on Apollo hop cone development, control and blank samples are available at the MassIVE repository database under ID MSV000095961. All derived molecular networking jobs can be publicly accessed at following links: FBMN job for positive ionization [https://gnps.ucsd.edu/ProteoSAFe/status.jsp?task=0f83c7cfad91444e855c1987f161e854], FBMN job for negative ionization [https://gnps.ucsd.edu/ProteoSAFe/status.jsp?task=19aa55bc1f564681b104ddd1b4858a82], NAP job for positive ionization [https://proteomics2.ucsd.edu/ProteoSAFe/status.jsp?task=025a463a22c54b2da226cf9ebe5c031a], NAP job for negative ionization [https://proteomics2.ucsd.edu/ProteoSAFe/status.jsp?task=7ff6e3f7657240c88bfc11f8e3b360fd], MolNetEnhancer job for positive ionization [https://gnps.ucsd.edu/ProteoSAFe/status.jsp?task=48dd84680d3c4e04b6f6b82e407157cd], MolNetEnhancer job for negative ionization [https://gnps.ucsd.edu/ProteoSAFe/status.jsp?task=bf995243b6e94a49ab702541dc26e1ce], Merged ionization network [https://gnps.ucsd.edu/ProteoSAFe/status.jsp?task=ed53ff8a61af457f8c9bf4de87153d89]. The following previously published data sets were used in the current study: occurrence datasets from the Global Biodiversity Information Facility (GBIF) for generation of Fig. 1b (https://doi.org/10.15468/dl.w53rkc, https://doi.org/10.15468/dl.cr37wu, https://doi.org/10.15468/dl.6ye7ug, https://doi.org/10.15468/dl.8jnu9g, https://doi.org/10.15468/dl.mn7b6d, and https://doi.org/10.15468/dl.rzwt9h); genome assemblies for *Humulus* cv. Cascade (Supplementary Fig. 4, https://hopbase.org) and for *Cannabis sativa* cv. Cs10 (Figs. 3b and 3c, https://www.ncbi.nlm.nih.gov/search/all/?term=GCA_900626175.2); gene annotation files for *Humulus* cvs. Saaz (Fig. 3a, https://plantgarden.jp/en/list/t3486/genome/t3486.G004), Cascade (Fig. 3a and Supplementary Fig. 20d, https://hopbase.org), and drHumLupu1.1 (Supplementary Fig. 21 https://www.ncbi.nlm.nih.gov/datasets/genome/GCF_963169125.1/) and *Cannabis sativa* cvs. Pink Pepper (Fig. 3a, https://www.ncbi.nlm.nih.gov/search/all/?term=GCA_029168945.1) and Cs10 (Supplementary Fig. 4b, 5, 20d, and 23, https://ftp.ncbi.nlm.nih.gov/genomes/all/GCF/900/626/175/GCF_900626175.2_cs10/), *Parasponia andersonii* (Fig. 5, https://www.ncbi.nlm.nih.gov/datasets/genome/GCA_002914805.1/), *Malus domestica* (Supplementary Fig. 20d and 23, https://ftp.ncbi.nlm.nih.gov/genomes/all/GCF/002/114/115/GCF_002114115.1_ASM211411v1/), *Prunus persica* (Supplementary Fig. 20d and 23, https://ftp.ncbi.nlm.nih.gov/genomes/all/GCA/000/346/465/GCA_000346465.2_Prunus_persica_NCBIv2/), *Vitis vinifera* (Supplementary Fig. 20d and 23, https://ftp.ncbi.nlm.nih.gov/genomes/all/GCA/000/003/745/GCA_000003745.2_12X), *Trema orientale* (Fig. 5 and Supplementary Fig. 23, https://ftp.ncbi.nlm.nih.gov/genomes/all/GCA/002/914/845/GCA_002914845.1_TorRG33x02_asm01/), *Morus notabilis* (Supplementary Fig. 23, https://ftp.ncbi.nlm.nih.gov/genomes/all/GCF/000/414/095/GCF_000414095.1_ASM41409v2/) and *Ziziphus jujuba* (Supplementary Fig. 20d and 23, https://ftp.ncbi.nlm.nih.gov/genomes/all/GCF/000/826/755/GCF_000826755.1_ZizJuj_1.1/).

Raw sequence data for the Zenith × USDA21058M biparental population (Supplementary Fig. 12b, https://www.ncbi.nlm.nih.gov/bioproject/?term=PRJNA906612). Source data are provided with this paper.

## Code availability

Code used for metabolomics data processing and analysis is available at GitHub (https://github.com/carlsberglaboratorium-publications/apollo-cone-development-metabolomics) and archived at Zenodo (https://doi.org/10.5281/zenodo.19134880).

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

## Acknowledgements

I.B. and B.S. thank Geoff Fincher for helpful advice before and during the project. S.P., A.H., and N.S. thank Ines Walde and Manuela Knauft for technical assistance with Hi-C sequencing. A.F. and N. Pitra thank Martin Waldinger and Rachel Bussey for technical assistance in generating the mapping family Apollo × PubM_740 and the hop plant material used for analysis. M.S. and K.F.X.M. acknowledge funding from the Bavarian Research Foundation (ID AZ-1577-22; M.S. and K.F.X.M.). N.B. and N.M. acknowledge financial support from the VILLUM Foundation, Denmark (grant no. 19151; N.B. and N.M.). Helmholtz Munich acknowledges funding support from the German Federal Ministry of Education and Research (de.NBI, 031A536B). C.J. acknowledges financial support from the Carlsberg Foundation (CF14-0214; C.J.) and the Danish Council for Independent Research | Medical Sciences (grant no. DFF – 4002-00391; C.J.). B.L.M. acknowledges financial support from the VILLUM Foundation through the VILLUM Center for Plant Plasticity (VKR 023054/00007523; B.L.M.).

## Author contributions

P.D.M. prepared Apollo DNA for sequencing and collected tissue for RNA-seq for annotation purposes. I.B. coordinated genome sequencing. N.S. coordinated and supervised HiC sequencing. S.P. prepared HiC libraries and A.H. performed sequencing of HiC libraries. A.F. provided the Apollo × PubM_740 mapping population and L.d.B and A.A. made the Cascade x HL-19-060-002M mapping population. N. Pitra provided GBS data for 243 hop accessions and coordinated RNA-seq for annotation

purposes. O.K. prepared plant material, isolated RNA for RNA-seq and Iso-Seq of the developing cone. K.N. coordinated Iso-Seq and RNA-seq for the developing cone. S.M.K. performed haploid and haplotype-resolved phased assemblies, ancestry analysis, genome comparison between cvs. Cascade and Apollo and collinearity analysis. S.M.K. and F.K. did GWAS and recombination analysis. F.K. identified candidate genes. S.M.K. and N. Pitra performed population genomic studies. H.G. conceptualized and developed sequence tags for phase and ancestry separation, performed assembly evaluations, TE and tandem repeat annotations and evaluation, analysed TE history via insertion age and executed hop-hemp, Apollo-Cascade, and Apollo phases comparisons. O.G. performed the metabolome analysis and transcriptomics of the developing cone and determined α acid content in the Apollo × PubM_740 population. I.K.-A. analysed genes of the humulone biosynthetic pathway. A.F. executed the downy mildew phenotyping in the Apollo × PubM_740 population. N.K. performed gene prediction and functional annotation, gene family analysis and provided the orthologous framework. G.H. and T.L. performed gene annotation and gene model consolidation. G.H. compared gene content between phases and performed comparative genomics analysis. A.A. conducted phylogenetic analysis and A.A. and L.d.B. did PCR analysis. L.H. and V.B. performed karyotype analysis. N. Price performed genome expansion analysis and gene family expansion study. M.-T. R.-W. provided advice on HiC assembly. N.B. coordinated and C.J. supervised mass spectrometry imaging. N.M. prepared samples and developed methods for mass spectrometry imaging and performed image analysis with input from N.B.. M.M. provided advice on genome assembly and interpretation of sequence data. M.S., P.D.M, A.F., and I.B. designed the study. S.M.K., O.G., A.A., H.G., M.S. and I.B. wrote the draft manuscript. S.M.K., H.G., O.G., A.A., O.K., I.B., F.K., N.K., N.B, N.M. and G.H. prepared figures. B.L.M. supervised I.K.-A. and provided input on metabolomics and the final manuscript. K.F.X.M. coordinated analytical work and K.F.X.M., M.M., and B.S. provided input for the draft and the final manuscript. M.S. coordinated gene prediction and annotation and comparative and TE analyses. I.B. wrote the final manuscript and coordinated the study. All co-authors contributed to and edited the final version.

## Competing interests

The authors declare the following competing interests: S.M.K., O.G., A.A., O.K., L.d.B., K.N., F.K., B.S., I.K.-A., and I.B. are current employees of, or were employed during the conduct of this study by, Carlsberg A/S. N. Pitra, N. Price, P.D.M., and A.F. are current employees of, or were employed during the conduct of this study by, Hopsteiner. B.L.M. is a consultant to Carlsberg Research Laboratory. Carlsberg A/S and Hopsteiner supported this work through salaries and other research support for authors affiliated with those organizations. Beyond the contributions of these co-authors, Carlsberg A/S and Hopsteiner had no additional role in study design, data collection and analysis, decision to publish, or preparation of the manuscript. All other authors declare no competing interests.
