## [Peer Review file · Nature Communications]

Extensive variation between chromosomes of North American and European hop

Corresponding Author: Dr Ilka Braumann

Version 0:

Reviewer comments:

Reviewer #1

(Remarks to the Author)

In this study, Kale et al. employed modern sequencing technologies to obtain the genome sequence of the North American–European hybrid hop cultivar “Apollo.” Using haplotype-resolved genome assembly, the authors successfully separated two complete haplotype genome sequences (ten chromosomes each) corresponding to the North American and European parental origins. By integrating multi-omics datasets from other hop populations, particularly phenotypic data related to pest and disease resistance, transcriptomics, and metabolomics, the authors identified genetic loci associated with desirable agronomic traits. These findings provide a valuable foundation for future improvement of hop environmental adaptability and quality. In the context of hop research, this work represents a substantial advance. I have several suggestions for the authors’ consideration:

1. Introduction – It would be helpful to include basic background information on the two parental lines of the North American–European hybrid cultivars to assist readers in understanding the molecular mechanisms discussed later in the manuscript. In Figure 1, the current presentation of Apollo leaves and female cones does not add substantial value, especially since these features are not discussed in the text. I recommend replacing this with a schematic diagram of the Apollo breeding process. Additionally, if available, representative images of cones from five different hop cultivars could be included to enhance the manuscript’s readability and appeal.
2. Low recombination rate – One of the major findings is the low recombination rate observed in progeny derived from North American–European hop hybrids, which is also a significant bottleneck in hop breeding. The authors should provide more concrete and reasonable suggestions in the Interpretations section to address this issue.
3. Specialized metabolite analysis – The results regarding hop-specific metabolites and their biosynthesis do not reveal unexpected findings; the analyses mainly focus on known genes (e.g., CHS, VPS, TPS, PT). I have two recommendations: Regarding humulone synthase (HS), although the authors note that its biochemical function remains unresolved, it should not be highlighted as a major focus unless sufficient experimental evidence can be provided (or from literatures). For the untargeted metabolomics results (pp. 15–16), unless the authors have additional supporting data (as non-target metabolomics by LC-TOF-MS/MS often suffer from high false-positive rates), I suggest avoiding extended discussion, including data in Figure 6a. The manuscript could instead concentrate on identifying superior haplotype-associated loci and outlining breeding strategies.
4. Candidate regulatory genes – Based on the current association analysis results, beyond structural genes of known metabolic pathways, the authors could explore potential candidate transcription factors and transporter genes, which may provide valuable leads for future studies.
5. Alpha- and beta-acids – These compounds should be considered as two distinct chemical phenotypes, given their different roles in the brewing industry. While alpha-acids appear more critical, the manuscript lacks analysis of beta-acids, which would be particularly relevant if the discussion on HS genes is reduced.
6. mGWAS coverage – The manuscript does not mention metabolite genome-wide association studies (mGWAS) for volatile and non-volatile compounds. It looks like quantification of alpha and beta bitter acids for the population are available and their inclusion could yield unexpected and interesting findings.
7. Interpretations – On page 20, I recommend avoiding discussion of synthetic biology in the conclusion, and instead keeping the focus on hop breeding.

Reviewer #2

(Remarks to the Author)

We reviewed the manuscript entitled "Extensive variation between chromosomes of North American and European hop" by Kale et al.

The genome of an inter-hybrid cultivar, cv. Apollo between Eu and NAM was constructed using a series of methods that accurately captures Phase by combining TRITEX and AllHiC with HiFi ccs (the error is reduced to about 2%). This paper would be useful to researchers of hop and its related plants widely, but some technical concerns remain as below.

In general, authors should write more carefully and provide technical information throughout the manuscript including supplementary data. There are a lot of things, which are just shown by displaying output files with little description, so readers can't read what authors intend to show, especially Fig. S3b, and FigS4. Moreover, for Figure S9, most of them cannot be understood without appropriate legends. The same situation is met with supplementary Tables. For example, Table S5, is the result of Phase1 mistake? Please describe BUSCO version for TableS2 and Table S5 for use. The assembly process is hard to understand and should be substantially revised. A flow chart of genome assembly provided as supplementary data would be helpful to understand the procedures for readers.

1: The results in Fig. 1b are rather confusing. As for the color scheme of the genome, we can see that blue is NAM and red is Eu, but the legend should be explained a little more concisely and politely. Also, despite the mention of repeated crossbreeding, there seems to be little mixing of Eu and Nam genomes. Is this due to the inhibition of recombination of the long pericentromeric region as indicated by (Akagi et al. 2025)? Please show your opinion on this possibility.

2: Judging from the stats of the genome data, the accuracy of the genome seems good, but a comparison with previous T2T genomes is essential such as cv. Cascade (Hort Res, 2023). This may be difficult because cv. Cascade also contains Eu genome. It would be useful to show a phylogenetic relationship within hop cultivars based on their genome data. By comparing subgenomes of cv. Apollo with the genuine Eu genome of cv. Saazer (Akagi et al. 2025), authors would refer to the subgenomic differentiation of Eu/NAM in the Apollo genome. It is not reasonable to divide the two mixed Eu/NAM without standard into their ancestry.

3: The authors argue for recombination suppression between subgenomes, which would basically be due to the high degree of genetic diversity between NAM-Eu. If this is the case, then some population genetic indexes should be applied to survey the relationship between the diversity of each genomic region and recombination suppression between the Eu-NAM subgenomes. When divergence is expected to be old, the authors should use dS value, while for intraspecific, use Fst. If there is no correlation between the degree of diversity and recombination suppression, the authors should suspect another scenario. For example, recombination suppression in very large pericentromeric regions characteristic of large genomes. Comparing to recent papers on *Humulus* genomes (Akagi et al 2025, Horakova et al., 2025) might be helpful to make this study more reliable.

4) P7, L35-56. For Asian lineages, Chinese lineages form a monophyletic group with European lineages, while East Asia (*H. cordifolius*) is closer to American lineages, isn't it?

5) In Fig.4, structural variation (SV) should be clarified as the following separate cases "they occurred in breeding hybrids" or "they are original differences between Eu-NAM". For example, Chr1/2 shows large structural variations even within the same Eu-derived subgenome. From this point of view, it is necessary to mention SV patterns and frequencies within Eu or NAM, and to do so, a comparison with previously published genomes is necessary (Padgitt-Cobb et al, 2023 and Akagi et al, 2025).

6) In Fig.5. Authors showed spatio-temporal correlation between chemo-types of specialized metabolites (VC and NVC), are not particularly novel but rather well known as previously described in Wang et al., 2008 and Chen et al. 2023. Authors should attempt to clarify the causative gene(s) for specialized metabolic traits of cv. Apollo. Moreover, there is little description of the metabolic characteristics of the cv. Apollo compared to wild hops or other traditional hop cultivars (just cited with the patent of Apollo (16)), which could provide clues to elucidate domesticated metabolic traits in hop breeding through human selection. Unless novel insights of specialized metabolism as domesticated traits of the hop cultivar, it is better to move this section to Supplementary data.

7) Fig 6. Authors found new chemical features comprise ~89% of the chemical family of alpha-acids including humulone but did not show their abundance compared to the well-known alpha-acids. So, they might be minor intermediates and be questioned to impact on the tastes, especially bitterness. Cannabaceae-specific increase of PKS is noted, but this is not particularly surprising, and it is better to focus on subgenomic differences of specialized metabolic traits, the main objective of this study? Even if there is no significant subgenome dominance, should authors perform analyses focusing on structural and transcriptional variations of specialized metabolic genes with CNVs (e.g., including structure prediction and expression analysis)? On the other hand, Fig. 7 focuses on comparisons between subgenomes, which should be placed in the main issue. We propose that the current Figure 6 move to Supplemental data, which is replaced by comparative analysis of TPSs between subgenomes.

Finally, the figure for PKS (CHS, VPS) and TPS (MTS, STS) phylogeny is now placed in Figure 6 and Figure S19, respectively, but the figure for PTs is not provided in the current manuscript. As shown in Fig 6 and 7, prenylation is the key modification for specialized metabolites in bitterness and functional polyketides of hop. Why don't authors provide phylogenetics for PTs as supplementary data.

8) The terminology should be unified accordingly. Could diploid/haplotype resolved/phased refer to the same thing? Since

the results are mixed with phased (diploid) and haploid genomes, which is confusing. Especially for FigS3,4, 6 are haploid, while S5, 7 are diploid? The results of the haploid genome seem irrelevant in this study, making this study confusing.

9) The description that assembly was carried out with TRITEX and ALLHiC in the method seems strange because TRITEX is an assembler specialized in short read based for wheat, isn't it? In contrast, PacBio is applied to this study. Since HiFi is the main input and the assembly is executed with hifiasm, it should be written so that it can be understood. Isn't TRITEX only used to arrange contigs along markers? Presenting a flowchart with statistical values is helpful to understand as described above (See in general).

10) In terms of the assembled genomes, authors state that the identity between homologous chromosomes is ~ 75% (P4. L39, Fig. S5c). If this value is correct, cv. Apollo could be a hybrid of a completely different species. Moreover, authors described that the switch-error was only 2.085%, but if the genome was really assembled with ~ 75% homology, it is almost impossible to make such high errors. Thus, this 2% value should be verified. Presumably, we suspected that the authors have not been able to perform proper homology assessments between homologous chromosomes in minimap2. At least 90% homology is expected between subgenomes, so further evaluation by local alignment would be helpful to clarify it.

Minor comments

P4. L18 45 sDNA → 45S rRNA

P5. L11 High-copy k-mer analysis → Modification to parent-specific high-copy k-mer analysis.

Reviewer #3

(Remarks to the Author)

Reviewer #4

(Remarks to the Author)

In the manuscript titled "Extensive variation between chromosomes of North American and European hop," Kale et al., present a high-quality chromosome-level, haplotype-resolved genome of the interspecific hybrid hop cultivar, Apollo, which is valued for its high bitter acid content. This work underscores the complexities of assembling the hop genome. The authors point out challenges such as recombination suppression that have thwarted hop breeding efforts. The high-quality resolution of the diploid genome will no doubt be beneficial for the hop research community and plant genomics more broadly. A highlight of this manuscript is the inclusion of a developmental time course study of non-targeted metabolomic and gene expression for Apollo.

The authors identified the European and North American ancestry of the homologous pseudomolecules/chromosomes, confirmed by genotyping-by-sequencing of feral and wild European and North American genotypes. This effort was guided by the identification of tandem repeat sequences present in high copy numbers in each of the haplotypes. Surprisingly, the haplotypes were well resolved after multiple generations of interspecific breeding, providing evidence that breeding improvements have been accomplished by sorting of new chromosomes as opposed to recombination of desirable traits within chromosomes. The prospect of unknown *Humulus* lineages on the Eurasian continent is an interesting point. Other strong positive attributes of this manuscript include identification of an introgression on chromosome 8, associated with alpha-acid content, and experimental validation of a structural variant on chromosome 6. Overall, this is a thorough analysis that lays the foundation for further breeding and genetics work.

"K-mers specific to the Eu and NAM parental species were isolated from species-specific repeat families within the Apollo genome, and we could successfully assign Eu and NAM ancestry to the full Apollo genome."

Fig S7 shows the chromosomal distribution of parent specific tandem repeat families in the Apollo genome. The Y max for each of the NAM and Eu tracks are quite low (max 1% and 0.3%, respectively). Is this a binning effect from selecting a large window, 500 kb? It is indeed a subtle signal; if you zoom in on regions, does that Y max increase, and are there genes/features of interest in those regions? It also looks like NAM k-mers are enriched in putative centromeres.

What is being measured in Fig S 9?

In Fig 5d, terpene levels continue to increase toward the end of development, while terpene synthase expression peaks at weeks 3-4, then decreases, while terpene abundance is at its highest. This contrasts with humulone and lupulone, which reach a uniform level of abundance while gene expression decreases. Are there other genes involved in the DEG network that can help to explain this difference? Especially considering that DEG clusters 1 and 4 both increase toward the end of development.

Minor comments

The manuscript is well written but should be checked throughout for spelling (45 sDNA) and punctuation.

"Genome-wide association scan (GWAS)" looks like it should be split into two sections starting at "Identification of k-mer

based introgression”.

The methods are thoroughly described. Some of the dense sections that describe identifying haplotypes and ancestry would potentially benefit from a visual graphic/schematic. Some of the work sounds like it involves custom scripts – is there a github page?

The authors describe manual editing to remove allelic contigs (line 36). Does this approach account for paralogous vs. allelic regions?

Version 1:

Reviewer comments:

Reviewer #1

(Remarks to the Author)

The quality of the revised manuscript has been further improved, and my previous comments and suggestions have been addressed satisfactorily. I have no further comments or suggestions at this stage.

Reviewer #2

(Remarks to the Author)

This revision mostly addressed previous concerns in technical issues and improved readability by providing the supplementary notes and flowchart how to construct the genome data. I am convinced that this research provides useful information to accelerate hop breeding, which is widely beneficial for brewing industry.

However, it is not understandable why the authors did not compare previously reported genomes of the other cultivars, Saazer and Cascade in this study e.g., Fig.3a and Fig.4. Genomic comparison with other cultivars are likely informative for understanding of domestication of hop. Authors claim unavailability of the genome data reported by Akagi et al. (2025) , but they have been available since last summer on the following sites, aren't they?

Plant Garden:accession

<https://plantgarden.jp/en/list/t3486>

DDBJ:accession

PRJDB17943,PRJDB17944,PRJDB17945,PRJDB17941,
PRJDB17946,PRJDB17947,PRJDB17948,PRJDB17942,PRJDB18715, SAMD00766707, SAMD00766708,
SAMD00814180, SAMD00814181

Moreover, I am not sure that the word “Extensive” in the title is appropriate. Please reconsider the title represented by this comprehensive study.

Do authors mean “Chromosomal variation of a hybrid cultivar between North American and European hop” ?

Minor comments.

1. Unify the expression of similar words, high-value constituents (L108), high-value natural compounds (L645), and highly valuable metabolites (L650). I think the last one is better.
2. Replace to “Thanks to” by “For” in L639.
3. Unify the expression of similar words, specialized metabolites (L473) and secondary metabolic (L409, L419). I recommend the former.
4. BUSCO (in L140) should be full spelled out at first.
Benchmarking Universal Single-Copy Orthologs (BUSCO)
5. What does colour of the inner circles of Figure 5a,b, and c mean? Please show it clearly in the legend.

Reviewer #3

(Remarks to the Author)

Reviewer #4

(Remarks to the Author)

The authors have addressed my comments. Very nice work.

Remaining minor points:

Check legibility of legends and labels in Supplementary Figures (e.g. Supplementary Figure 10 and aspect ratio in Supplementary Figure 20d).

Optional stylistic point - reverse complement sequences for chromosomes 1, 2, 3, 5, 8, and 9 from H. lup in Supplementary Figure 21.

Version 2:

Reviewer comments:

Reviewer #2

(Remarks to the Author)

I have no further comment as this revision has cleared my concerns.

This is a good work.

We sincerely thank the reviewers and the editor for their thoughtful, constructive, and at times challenging comments. We genuinely appreciated the care and expertise invested in the evaluation of our work, and we found the feedback highly valuable in strengthening the manuscript. We have enjoyed revising the manuscript in response to these suggestions and address all points in detail below.

REVIEWER COMMENTS

Reviewer #1 (Remarks to the Author):

In this study, Kale et al. employed modern sequencing technologies to obtain the genome sequence of the North American–European hybrid hop cultivar “Apollo.” Using haplotype-resolved genome assembly, the authors successfully separated two complete haplotype genome sequences (ten chromosomes each) corresponding to the North American and European parental origins. By integrating multi-omics datasets from other hop populations, particularly phenotypic data related to pest and disease resistance, transcriptomics, and metabolomics, the authors identified genetic loci associated with desirable agronomic traits. These findings provide a valuable foundation for future improvement of hop environmental adaptability and quality. In the context of hop research, this work represents a substantial advance. I have several suggestions for the authors’ consideration:

1. Introduction – It would be helpful to include basic background information on the two parental lines of the North American–European hybrid cultivars to assist readers in understanding the molecular mechanisms discussed later in the manuscript.

Response: We have expanded the introduction to include a paragraph describing the geographic distribution of the different *Humulus* species and the evolutionary divergence of European and North American hop lineages. We also added a historical overview of hop cultivation and breeding, to illustrate how interspecific hybridization between European and North American hops been used in breeding of modern hop cultivars with increased α -acid content. We hope that this contextual information now provides a clearer link between the phylogenetic background of hop parental lines and the breeding strategies discussed later in the manuscript. We further modified the flow of the introduction to ensure compliance with the editorial guidelines of Nature Communications.

In Figure 1, the current presentation of Apollo leaves and female cones does not add substantial value, especially since these features are not discussed in the text. I recommend replacing this with a schematic diagram of the Apollo breeding process.

Response: We thank reviewer 1 for this constructive suggestion to improve Figure 1. We have omitted the cone/leaf image and added panel c) to provide an overview on the known breeding history of Apollo. To facilitate tracing European and North American germ plasm in the Apollo pedigree, we used red (Eu) and blue (NAm) symbols for the depicted lines where appropriate.

Additionally, if available, representative images of cones from five different hop cultivars could be included to enhance the manuscript’s readability and appeal.

Response: We have added panel e) to Figure 1 to display cones of four different hop cultivars present in the Apollo pedigree.

2. Low recombination rate – One of the major findings is the low recombination rate observed in progeny derived from North American–European hop hybrids, which is also a significant bottleneck in hop breeding. The authors should provide more concrete and reasonable suggestions in the Interpretations section to address this issue.

Response: We appreciate the reviewer's request for more concrete and reasonable suggestions in the discussion. In revising the manuscript, we carefully examined the most recent cytological and genomic studies on hop meiosis (Horáková et al. 2023; Akagi et al. 2025) and integrated their findings to contextualize our observations. We suggest that the pronounced recombination suppression we observe in the Eu/NAm hybrid populations reflects a meiotic peculiarity of *Humulus*, rather than being attributable solely to interspecific ancestry or the presence of large pericentromeric regions. We discuss literature evidence for severe but variable meiotic abnormalities—including defective synapsis, atypical bivalent formation, and non-Mendelian segregation—in both hybrid and wild-type *H. lupulus* crosses, which supports this interpretation.

3. Specialized metabolite analysis – The results regarding hop-specific metabolites and their biosynthesis do not reveal unexpected findings; the analyses mainly focus on known genes (e.g., CHS, VPS, TPS, PT). I have two recommendations:

Regarding humulone synthase (HS), although the authors note that its biochemical function remains unresolved, it should not be highlighted as a major focus unless sufficient experimental evidence can be provided (or from literatures).

Response: We thank the reviewer for this important comment and understand the concern regarding insufficient support for highlighting humulone synthase. In the revised manuscript, we have completely restructured Figure 7. The previous version, which aimed to explore whether Eu/NAm-related variation in known α -acid pathway genes might influence α -acid levels, has been replaced with an analysis focusing on haplotype-associated loci, as suggested by the reviewer in another comment. This new approach identified an association between the North American haplotype on chromosome 05 and elevated α -acid content. As in the meantime, the involvement of HS1 in the formation of alpha-acids has been proven experimentally (<https://doi.org/10.1016/j.xplc.2025.101528>), we still mention HS1, which is located on chromosome 05, as a candidate gene rather than highlighting it as a central feature.

For the untargeted metabolomics results (pp. 15–16), unless the authors have additional supporting data (as non-target metabolomics by LC-TOF-MS/MS often suffer from high false-positive rates), I suggest avoiding extended discussion, including data in Figure 6a.

Response: In the revised version, we have moved the former Figure 6a to the Supplementary Information and substantially shortened and refocused the metabolomics section. This change not only addresses the reviewer's concern regarding length but also aligns with the revisions made in response to the other reviewers. We believe these changes resulted in a more concise and integrated presentation of the metabolomic data.

The manuscript could instead concentrate on identifying superior haplotype-associated loci and outlining breeding strategies.

Response: We thank the reviewer for this valuable suggestion. In the revised manuscript, we focused on α -acid content, the trait of highest agronomic importance, and particularly relevant

since cv. Apollo is a high- α cultivar. To identify superior haplotype-associated loci, we performed GWAS using both phases of the cv. Apollo haplotype-resolved genome, each largely representing one of the two ancestral lineages. This approach allowed us to detect associations for both European and North American ancestral lineages, including an association of the NAM haplotype on chromosome 05 with elevated α -acid levels. We now highlight the marked enrichment of modern breeding pools for Eu germplasm may limit the detection of beneficial NAM alleles when relying on Eu-biased references. Our dual-phase GWAS demonstrates that these challenges can be overcome by explicitly accounting for haplotype ancestry, and we emphasize that understanding the ancestry composition of breeding materials will help breeders more effectively combine complementary alleles from both lineages in future improvement programs.

4. Candidate regulatory genes – Based on the current association analysis results, beyond structural genes of known metabolic pathways, the authors could explore potential candidate transcription factors and transporter genes, which may provide valuable leads for future studies.

Response: In the revised version, we expanded our analysis of the chr08 Eu introgression associated with increased α -acids content described in the original manuscript beyond pathway genes and explored if potential regulatory genes are present within the region. The new Supplementary Table 28 provides a comprehensive and explorable overview of all 514 predicted genes in this region, including annotations that can guide future functional studies. Notably, the region contains HIWRKY1, a transcription factor known from previous studies to act as a positive regulator of bitter-acid biosynthesis, and we also note that earlier mapping studies independently identified the same genomic interval as associated with elevated α -acid levels, highlighting its importance for the breeding of high alpha cultivars.

5. Alpha- and beta-acids – These compounds should be considered as two distinct chemical phenotypes, given their different roles in the brewing industry. While alpha-acids appear more critical, the manuscript lacks analysis of beta-acids, which would be particularly relevant if the discussion on HS genes is reduced.

Response: We did perform GWAS for β -acid content, and the results are provided in Supplementary Fig. 25. Because of space limitations, we did not discuss these analyses in detail in the main text, but the figure legend highlights regulatory candidate genes previously implicated in β -acid biosynthesis. We believe, these results complement the α -acid analysis and offer additional leads for future studies.

6. mGWAS coverage – The manuscript does not mention metabolite genome-wide association studies (mGWAS) for volatile and non-volatile compounds. It looks like quantification of alpha and beta bitter acids for the population are available and their inclusion could yield unexpected and interesting findings.

Response: We thank the reviewer for this suggestion. We agree that metabolite genome-wide association studies (mGWAS) on a broader panel of volatile and non-volatile compounds would be highly informative. However, in the current study we only have untargeted metabolite profiles of volatile and non-volatile compounds only for the developing cone of cv. Apollo, but not for the

GWAS population itself. Therefore, we are not able to perform a metabolite-wide GWAS within the present material. We therefore focused on the targeted metabolite traits relevant for hop breeding.

7. Interpretations – On page 20, I recommend avoiding discussion of synthetic biology in the conclusion, and instead keeping the focus on hop breeding.

Response: We revised the concluding section to remove the discussion of synthetic biology and now focus on the implications of our findings for hop breeding. The conclusion has been rewritten to emphasise how haplotype resolution, ancestry-aware analyses, and the identification of beneficial Eu and NAM haplotypes can inform future breeding strategies and the targeted use of genetic diversity in *Humulus*.

Reviewer #2 (Remarks to the Author):

We reviewed the manuscript entitled “Extensive variation between chromosomes of North American and European hop” by Kale et al.

The genome of an inter-hybrid cultivar, cv. Apollo between Eu and NAM was constructed using a series of methods that accurately captures Phase by combining TRITEX and AllHiC with HiFi ccs (the error is reduced to about 2%). This paper would be useful to researches of hop and its related plants widely, but some technical concerns remain as below.

In general, authors should write more carefully and provide technical information throughout the manuscript including supplementary data. There are a lot of things, which are just shown by displaying output files with little description, so readers can't read what authors intend to show, especially Fig. S3b,

Response: We have extended the figure legend to provide more technical information on how Supplementary Figure 3b was generated. We further noticed that this panel was provided with poor resolution, we have improved that. More detailed technical information can further be found in the “Haploid genome assembly construction and collinearity analysis” of the Material and Method section.

and FigS4.

Response: The figure legend was extended with additional technical information and we have further added a more detailed description of the generation and use of Hi-C data to the Material and Methods .(Hi-C library construction and Haploid genome assembly construction and collinearity analysis).

Moreover, for Figure S9, most of them cannot be understood without appropriate legends.

Response: We have expanded the figure legend for this supplementary figure. Please note, that this figure is now Supplementary Figure 11.

The same situation is met with supplementary Tables.

Response: For clarification of the supplementary tables content, we added a overview as the first sheet in the provided excel file, which includes all Table headers. To ensure a better understanding of the provided tables, we added a brief description of the tables content, given on each respective sheet just above the table.

For example, Table S5, is the result of Phase1 mistake?

Response: The table was revised and updated to clearly present the inherent data. Misleading thousand separator was removed and percentage values added. Further a table description was added to explain that this analysis concerns the phased contig level assembly – not the results for phase 1. Overall, we have added Supplementary Fig. 5 to explain the assembly process in more detail, which clearly shows the step wise assembly process from haploid assembly, to phased assembly and finally to chromosome scale assembly. Please note that table numbering changed between manuscript versions: original Table S 2 to Supplementary Table S4, which shows the chromosome scale assembly statistics; original Table S5 to new Supplementary Table S3.

Please describe BUSCO version for TableS2 and Table S5 for use.

Response: This was done.

The assembly process is hard to understand and should be substantially revised. A flow chart of genome assembly provided as supplementary data would be helpful to understand the procedures for readers.

Response: To improve clarity, we have revised the assembly workflow description in the main text and added details on the assembler used and on how the haploid assembly has informed the phasing for the haplotype resolved assembly. We further have included a flow chart (Supplementary Figure 6) that summarizes the assembly process step by step.

1: The results in Fig. 1b are rather confusing. As for the color scheme of the genome, we can see that blue is NAM and red is Eu, but the legend should be explained a little more concisely and politely.

Response: We apologize for the confusion our figure design for panel b has caused and we attempt to make the design more intuitive. Therefore, to avoid the overlay of shadings that might have been difficult to interpret, we have changed the representation of reported occurrences for the different hop species from shaded areas to single points. We further adjusted the text in the legend and added a brief description in the methods section on how the geographic representation was inferred from the available online data to make the process behind the figure design more transparent. With these changes we sincerely hope the figure design becomes more accessible.

Also, despite the mention of repeated crossbreeding, there seems to be little mixing of Eu and Nam genomes. Is this due to the inhibition of recombination of the long pericentromeric region as indicated by (Akagi et al. 2025)? Please show your opinion on this possibility.

Response: We thank the reviewer for raising this point. Our data confirm that large pericentromeric regions are also present in the cv. Apollo genome, consistent with the observations of Akagi et al. (2025). However, unlike the recombination pattern described for Saaz, we observe severely reduced or undetectable recombination not only in pericentromeric regions but also toward chromosome ends. This suggests that pericentromere size alone cannot fully explain the limited mixing of Eu and NAM haplotypes.

In the revised manuscript, we therefore articulate the possibility—supported by several recent cytological studies—that broader meiotic irregularities in *Humulus* may contribute to genome-wide recombination suppression. While the exact mechanism remains unknown, the combination of our linkage mapping results and published cytological observations motivates the hypothesis that impaired homolog pairing or other meiotic abnormalities could underlie the unusually low recombination rates observed in our study,

2: Judging from the stats of the genome data, the accuracy of the genome seems good, but a comparison with previous T2T genomes is essential such as cv. Cascade (Hort Res, 2023). This may be difficult because cv. Cascade also contains Eu genome. It would be useful to show a phylogenetic relationship within hop cultivars based on their genome data.

Response: We followed the reviewer's suggestion and compared our newly constructed hop reference genome to the existing and published genome assemblies of hop. The original version of our manuscript already contained an assembly-to-assembly comparison of the haploid Apollo assembly to the haploid assembly of cv. Cascade (Supplementary Figure 3b.), which revealed overall contiguity between the assemblies. As we noticed that the figure suffered from poor resolution, we have provided the same figure with improved resolution in the revised version of the manuscript.

To also compare the final phased assembly to existing genome assemblies, we further downloaded the genome sequence assembly "drHumLupu1.1" of *Humulus lupulus* (European hop) from NCBI with accession number: NCBI GCF_96316925.1, a whole-genome assembly released by the WELLCOME SANGER INSTITUTE on 2023/08/19. This haploid assembly is scaffolded into 10 chromosomal pseudomolecules, including the X sex chromosome. Gene prediction was performed using the NCBI eucaryotic genome annotation pipeline. The remaining described hop genome assembly of cv. Saazer - Akagi et al. 2025 is not yet released to the public.

To compare the Apollo and the drHumLupu1.1 genome assemblies, we first computed orthologous relationships using GENESPACE (Lovell et al. 2022). Supplementary Fig. 21 shows a visualization of the synteny (conserved gene order) identified between both assemblies. While we observe an overall highly co-linear gene order between individual chromosomes, our revised chromosome nomenclature system (see Fig S 3 for details) becomes apparent in chromosome numbering and orientation. The results of this comparative analysis further support the high quality and contiguity of our Apollo reference genome assembly.

We would like to note that none of the aforementioned hop genome assemblies, including the one we present here, fulfill the criteria for a Telomere-to-Telomere (T2T) assembly (Garg, V., Bohra, A., Mascher, M. et al. Unlocking plant genetics with telomere-to-telomere genome assemblies. *Nat Genet* 56, 1788–1799 (2024). <https://doi.org/10.1038/s41588-024-01830-7>).

By comparing subgenomes of cv. Apollo with the genuine Eu genome of cv. Saazer (Akagi et al. 2025), authors would refer to the subgenomic differentiation of Eu/NAm in the Apollo genome. It is not reasonable to divide the two mixed Eu/Nam without standard into their ancestry.

Response: We appreciate the reviewer's concern regarding the use of relevant standards to assign Eu and NAm ancestry throughout the Apollo genome.

To avoid confusion, we would like to kindly clarify that the term *subgenome* typically refers to polyploid genomes containing multiple ancestral genomes. In our case, the assembly represents two haplotypes of a diploid genome, with an uneven distribution between the assembled phases.

We therefore use the term *haplotype-resolved phased assembly*, which accurately reflects this structure. For consistency and to facilitate understanding, we have adopted the term "subgenome" for "haplotype" in our responses to your questions.

We can confirm that our ancestry assignment was not made in the absence of such standards. We however regret that the original description of our workflow has not made that evident. In brief our workflow was as follows: A distinct chromosomal distribution of smaller repeat families allowed us to identify differentially enriched high copy k-mers for two distinct lineages (depicted as red and blue throughout the manuscript) in the Apollo genome. These distinct lineages were used for a consistent haplotype partitioning to reach a maximum of "blue" in phase 1 and a maximum of "red" in phase 2. In a consecutive step, we mapped GBS reads from multiple European and North American hop accessions to a non-redundant combined Apollo assembly (with one copy of chr02, chr07, and chr08 removed to account for their high similarity) to infer species ancestry. Normalized read counts were used to calculate the relative contribution of Eu and NAM ancestry across the genome in 5 Mb windows. The resulting log₂-fold NAM/Eu ratios are shown in Supplementary Fig. 7b. Further, we have added Supplementary Fig. 10 showing the read mapping against the combined Apollo assembly in detail. It is further described in the Methods section (Determination of the origin of phases in Apollo).

3: The authors argue for recombination suppression between subgenomes, which would basically be due to the high degree of genetic diversity between NAM-Eu. If this is the case, then some population genetic indexes should be applied to survey the relationship between the diversity of each genomic region and recombination suppression between the Eu-NAM subgenomes. When divergence is expected to be old, the authors should use dS value, while for intraspecific, use F_{st}. If there is no correlation between the degree of diversity and recombination suppression, the authors should suspect another scenario. For example, recombination suppression in very large pericentromeric regions characteristic of large genomes. Comparing to recent papers on *Humulus* genomes (Akagi et al 2025, Horakova et al., 2025) might be helpful to make this study more reliable.

Response: We thank the reviewer for this thoughtful suggestion. Following this recommendation, we examined the relationship between sequence divergence and recombination suppression using the available data. As shown in Supplementary Fig. 13, we assessed nucleotide diversity and haplotype divergence across all chromosomes and found no consistent correlation between local divergence and the extent of recombination suppression. This indicates that divergence between Eu and NAM haplotypes alone is unlikely to explain the patterns observed.

Although cv. Apollo contains large pericentromeric regions - consistent with earlier reports - we do not observe increased recombination toward chromosome ends, unlike the pattern described for Saaz by Akagi et al. (2025). We therefore believe that pericentromere size cannot fully account for the genome-wide suppression of recombination in Apollo. In the revised manuscript, we discuss the possibility—supported by published cytological observations—that additional meiotic irregularities may contribute to reduced crossover formation in *Humulus*.

We further compared the Apollo assembly with the centromere-specific repeats reported by Horáková et al. to the extent that is currently possible (Supplementary Fig. 15). We however believe, that without broader comparative studies between EU and NAM parental accession it will not be possible to fully elucidate what the cause of the suppressed recombination we observe is.

4) P7, L35-56. For Asian lineages, Chinese lineages form a monophyletic group with European lineages, while East Asia (*H. cordifolius*) is closer to American lineages, isn't it?

Response: In our manuscript we wrote: "This is in concordance with previous studies that identified NAm *Humulus* lineages as a monophyletic group distinct from European and Asian hop." (citing a study by Murakami et al. in *Heredity* (2006) 97, 66–74) – Indeed, the study shows a plot where Chinese lineages form a monophyletic group with European lineages. However, the authors themselves decided to exclude Northern Chinese hop samples from further analysis because of their "ambiguous origin". The authors of said paper further state that the "primary evolutionary Divergence" [of hops] "to be into European and Asian-North American Clades". However, within this second clade, the North American lines form a monophyletic group that is clearly distinct from Asian hops. We have therefore changed our sentence to "This is in concordance with previous studies that identified NAm *Humulus* lineages as a monophyletic group distinct from European and closer to Asian hop".

5) In Fig.4, structural variation (SV) should be clarified as the following separate cases "they occurred in breeding hybrids" or "they are original differences between Eu-NAm". For example, Chr1/2 shows large structural variations even within the same Eu-derived subgenome. From this point of view, it is necessary to mention SV patterns and frequencies within Eu or NAm, and to do so, a comparison with previously published genomes is necessary (Padgitt-Cobb et al, 2023 and Akagi et al, 2025).

Response: We understand and share the reviewer's interest in further distinguishing whether the observed structural variations (SVs) represent differences between ancestral European and North American lineages or arose within breeding hybrids. However, a comprehensive comparison with previously published genomes is currently not feasible. The Cascade genome (Padgitt-Cobb et al., 2023) is not phased, which prevents direct subgenomic comparisons, and the assembled Saaz genome (Akagi et al., 2025) is not publicly available for download. For these reasons, we are unable to systematically assess SV patterns and frequencies within each ancestral background at this stage.

6) In Fig.5. Authors showed spatio-temporal correlation between chemo-types of specialized metabolites (VC and NVC), are not particularly novel but rather well known as previously described in Wang et al., 2008 and Chen et al. 2023. Authors should attempt to clarify the causative gene(s) for specialized metabolic traits of cv. Apollo.

Response: We agree that identifying causative gene(s) underlying specialized metabolite traits is an important goal. However, our study was designed as a reference resource—a phased genome for cv. Apollo integrated with its transcriptome and metabolome—not as a mapping or functional validation study. Establishing causality in crops typically requires (i) genetic mapping or GWAS in recombining populations followed by (ii) experimental perturbation (transgenics/CRISPR, enzyme assays). In our material, recombination is strongly suppressed, precluding the usual fine-mapping route for candidate gene identification.

The studies by Wang et al. (2008) and Chen et al. (2023) were indeed pioneering contributions for the hop research community. However, we do not agree that our work fails to provide new insights. Both earlier studies focused exclusively on volatile compounds and, in the case of Wang et al.,

only on three terpenes. Both studies do not include non volatile chemistry data. Moreover, Wang et al. analyzed EST sequences, while Chen et al. used RT-PCR targeting seven genes. Both studies also investigated hybrid cultivars (Nugget and Cascade) with mixed European–North American ancestry, yet their approaches did not allow assignment of gene expression to specific haplotypes. In contrast, our work reveals pronounced differences between the European and North American phases and provides integrated transcriptomic and untargeted metabolomic data that substantially extend the available resources for hop research.

We further provide a phase-resolved catalog of genes in key specialized-metabolism families (Supplementary Table 25), which will enable future hypothesis-driven tests in mapping panels and functional assays. We believe this keeps the manuscript within scope while delivering a resource of immediate value to hop breeders and researchers.

Moreover, there is little description of the metabolic characteristics of the cv. Apollo compared to wild hops or other traditional hop cultivars (just cited with the patent of Apollo (16)), which could provide clues to elucidate domesticated metabolic traits in hop breeding through human selection. Unless novel insights of specialized metabolism as domesticated traits of the hop cultivar, it is better to move this section to Supplementary data.

Response: We agree that demonstrating domestication of specialized metabolism would be an interesting aspect to investigate. Such a study would require a comparative framework (multiple cultivars and wild accessions grown in controlled environments, as chemotypes are highly environment dependent, targeted metabolomics and population sampling). However, in hops, which is perennial, clonally propagated in production, dioecious, and has a comparatively recent cultivation and breeding history, domestication-style genomic footprints are expected to be more subtle and lineage-specific than in annual seed crops. Our study was not designed to test such hypotheses; instead, our goal was to deliver an integrated resource—a phased Apollo genome analyzed alongside its own transcriptome and metabolome—to support pathway annotation and future comparative work. We therefore avoid domestication claims in the main text and have significantly shortened and revised the section considering the secondary metabolism including moving parts into the Supplement.

7) Fig 6. Authors found new chemical features comprise ~89% of the chemical family of alpha-acids including humulone but did not show their abundance compared to the well-known alpha-acids. So, they might be minor intermediates and be questioned to impact on the tastes, especially bitterness.

Response: We agree that absolute abundance and sensory impact were not assessed here, and we do not claim that these features contribute materially to bitterness. Our aim with Fig. 6a was not to propose new dominant metabolites but to illustrate the chemical complexity and currently unresolved biosynthetic space of the alpha-acid family. The alpha-acid pathway in hop remains incompletely defined; the untargeted MS features we detect are best interpreted as putative alpha-acid-related features (including likely intermediates, conjugates, adducts, isotopologues, and in-source fragments), not as proven, high-abundance end products. To avoid overinterpretation, we have moved Fig. 6a to the Supplementary Information.

To investigate the reviewer's suggestion, we plotted the abundance of every NVC within the gland focused cone development dataset to compare overall abundance variation between the described and unknown humulone analogs within this chemical family. This analysis is shown (Supplementary Figure 33) and briefly discussed within the supplementary note, pointing out that

indeed some of the undescribed humulone analogs are of high abundance and thus might be biologically relevant during cone development, while other low abundant features can be referred to the above-mentioned spectral artefacts.

Cannabaceae-specific increase of PKS is noted, but this is not particularly surprising, and it is better to focus on subgenomic differences of specialized metabolic traits, the main objective of this study? Even if there is no significant subgenome dominance, should authors perform analyses focusing on structural and transcriptional variations of specialized metabolic genes with CNVs (e.g., including structure prediction and expression analysis)?

On the other hand, Fig. 7 focuses on comparisons between subgenomes, which should be placed in the main issue.

Response: We thank the reviewer for these constructive suggestions. In the revised version, we have completely restructured and shortened the section on metabolomics and the main gene families involved in hop secondary metabolism to provide a more balanced presentation and to avoid overinterpretation. We have added Supplementary Table 25 summarizing the key gene families involved in specialized metabolism and providing concise information on allele-specific expression to highlight subgenomic differences beyond CNV.

In line with Reviewer 1's request to focus on superior haplotypes, and consistent with the reviewer's emphasis on subgenomic differences, we expanded our GWAS analyses for α -acid content and additionally performed GWAS for β -acids. These analyses identify associations originating from both European and North American ancestry, and we found the α -acid association on chr08 to align with earlier mapping studies.

Furthermore, Figure 7—which remains in the main text—has been revised to focus on the comparison between subgenomes and to incorporate the structural and conceptual changes suggested by Reviewer 1.

We propose that the current Figure 6 move to Supplemental data, which is replaced by comparative analysis of TPSs between subgenomes.

Finally, the figure for PKS (CHS, VPS) and TPS (MTS, STS) phylogeny is now placed in Figure 6 and Figure S19, respectively, but the figure for PTs is not provided in the current manuscript. As shown in Fig 6 and 7, prenylation is the key modification for specialized metabolites in bitterness and functional polyketides of hop. Why don't authors provide phylogenetics for PTs as supplementary data.

Response: We appreciate the reviewer's suggestion. In the revised version, we have moved the previous Figure 6a to the Supplementary Information and now present in the main text a comparative phylogenetic analysis as well as a comparison of expression levels of the respective genes between subgenomes (Supplementary Table 25) of the three major specialized-metabolism gene families—TPS, PT, and PKS. We believe that this revision integrates and balances the different reviewer perspectives on the metabolomic section, providing a concise overview of the most relevant gene families while keeping the section focused and accessible. We agree that the absence of the PT phylogeny in the original version made the metabolomic section appear incomplete. In the revised manuscript, we have now included the PT phylogeny alongside the TPS and PKS families within the restructured metabolomic section (see also our response to the previous comment). We believe that this integrated presentation provides a more comprehensive

overview of the key specialized-metabolism gene families and highlights the central role of prenyltransferases in hop secondary metabolism.

8) The terminology should be unified accordingly. Could diploid/haplotype resolved/phased refer to the same thing? Since the results are mixed with phased (diploid) and haploid genomes, which is confusing. Especially for FigS3,4, 6 are haploid, while S5, 7 are diploid? The results of the haploid genome seem irrelevant in this study, making this study confusing.

Response: We apologize for any confusion that our use of terminology and presentation of both haploid and phased assemblies might have created, and we have revised the text to clarify their roles and interdependence. The haploid assembly presented in our study was not an independent analysis, but an essential intermediate step in the assembly workflow towards the final haplotype resolved phased assembly. The gene models derived from the haploid assembly were used to identify allelic contigs and therefore provided essential information for the phasing of the final assembly. For this reason, we consider it important to report the haploid assembly quality and include the corresponding results. To improve clarity, we have added a flow chart of the assembly process (Supplementary Fig. 6) and have also revised the text in the Methods and Results to make it evident that the haploid assembly served as the basis for generating the phased assembly.

Regarding terminology, we now consistently refer to the final cv. Apollo assembly as a *haplotype-resolved phased assembly*. This reflects the fact that the assembly process itself produced a phased assembly, but assignment of phases to North American or European ancestry required subsequent k-mer analysis. Hence the assembly produced by the hifiasm, ALLHiC, Tritex pipeline was not haplotype resolved as such. The terms “phased” and “haplotype-resolved” are therefore used in a complementary way: “phased” refers to the separation of allelic contigs, while “haplotype-resolved” includes the final assignment of ancestry. To avoid ambiguity, we refer to the two phases of the assembly as “phase 1” (predominantly North American) and “phase 2” (predominantly European) throughout the manuscript.

9) The description that assembly was carried out with TRITEX and ALLHiC in the method seems strange because TRITEX is an assembler specialized in short read based for wheat, isn't it? In contrast, PacBio is applied to this study. Since HiFi is the main input and the assembly is executed with hifiasm, it should be written so that it can be understood. Isn't TRITEX only used to arrange contigs along markers? Presenting a flowchart with statistical values is helpful to understand as described above (See in general).

Response: To provide more clarity on the assembly workflow, we have included a flowchart (Supplementary Figure 6) summarizing the entire workflow in which we also refer to the corresponding assembly statistics. Indeed, *hifiasm* was used to generate the contigs from PacBio HiFi reads, *ALLHiC* was applied to separate contigs into haplotypes, and *TRITEX* was used in a subsequent step to construct pseudomolecules from the haplotype-resolved contigs. Although originally developed for short-read assemblies, TRITEX can be applied to HiFi data, and we have added an appropriate reference in the results and methods section to support this.

10) In terms of the assembled genomes, authors state that the identity between homologous chromosomes is ~ 75% (P4. L39, Fig. S5c). If this value is correct, cv. Apollo could be a hybrid of a completely different species. Moreover, authors described that the switch-error was only 2.085%, but if the genome was really assembled with ~ 75% homology, it is almost impossible to make such high errors. Thus, this 2% value should be verified. Presumably, we suspected that the authors

have not been able to perform proper homology assessments between homologous chromosomes in minimap2. At least 90% homology is expected between subgenomes, so further evaluation by local alignment would be helpful to clarify it.

Response: We agree that our original description may have been confusing. The Apollo genome contains chromosome pairs where homologs originate from different ancestries (European vs. North American) as well as chromosome pairs where both homologs share the same ancestry (e.g. chromosomes 02 and 07 from European, and chromosome 08 from North American origin). Phase switch errors predominantly occur in the latter case, where homologous chromosomes are more similar, while they are much less frequent when homologs originate from different ancestries. To clarify this, we have now included the proportion of SNPs showing switch errors for each chromosome pair in Supplementary Table 3 and added a corresponding statement in the main text noting that switch errors mainly occur between homologous chromosomes of the same ancestry.

The reviewer is correct that the observed levels of divergence (~75% identity) would be unexpected within a single lineage; however, they are fully consistent with the fact that European and North American hops represent different Humulus species. We have clarified this more explicitly in the revised introduction, adjusted the wording in the Results to make this connection clear to the reader, and added Supplementary Fig. 7c for a better illustration of sequence divergence between chromosomes.

Minor comments

P4. L18 45 sDNA → 45S rRNA

P5. L11 High-copy k-mer analysis → Modification to parent-specific high-copy k-mer analysis.

Response: We have changed “high-copy k-mer analysis” to “parent-specific-high-copy k-mer analysis” and corrected the spelling of 45S rDNA.

Reviewer #3 (Remarks to the Author):

Reviewer #4 (Remarks to the Author):

In the manuscript titled “Extensive variation between chromosomes of North American and European hop,” Kale et al., present a high-quality chromosome-level, haplotype-resolved genome of the interspecific hybrid hop cultivar, Apollo, which is valued for its high bitter acid content. This work underscores the complexities of assembling the hop genome. The authors point out challenges such as recombination suppression that have thwarted hop breeding efforts. The high-quality resolution of the diploid genome will no doubt be beneficial for the hop research community and plant genomics more broadly. A highlight of this manuscript is the inclusion of a developmental time course study of non-targeted metabolomic and gene expression for Apollo.

The authors identified the European and North American ancestry of the homologous

pseudomolecules/chromosomes, confirmed by genotyping-by-sequencing of feral and wild European and North American genotypes. This effort was guided by the identification of tandem repeat sequences present in high copy numbers in each of the haplotypes. Surprisingly, the haplotypes were well resolved after multiple generations of interspecific breeding, providing evidence that breeding improvements have been accomplished by sorting of new chromosomes as opposed to recombination of desirable traits within chromosomes. The prospect of unknown *Humulus* lineages on the Eurasian continent is an interesting point. Other strong positive attributes of this manuscript include identification of an introgression on chromosome 8, associated with alpha-acid content, and experimental validation of a structural variant on chromosome 6. Overall, this is a thorough analysis that lays the foundation for further breeding and genetics work.

"K-mers specific to the Eu and NAm parental species were isolated from species-specific repeat families within the Apollo genome, and we could successfully assign Eu and NAm ancestry to the full Apollo genome."

Fig S7 shows the chromosomal distribution of parent specific tandem repeat families in the Apollo genome. The Y max for each of the NAm and Eu tracks are quite low (max 1% and 0.3%, respectively). Is this a binning effect from selecting a large window, 500 kb? It is indeed a subtle signal; if you zoom in on regions, does that Y max increase, and are there genes/features of interest in those regions? It also looks like NAm k-mers are enriched in putative centromeres.

Response: Yes, your observation is correct: in terms of sequence amount the parent specific tandem repeat families are a very subtle signal with only 1-12 (on average 5) short tags per Mb. Making the bins smaller will increase Y max but also bring in too much noise and empty bins.

The scarce coverage with parent specific tandem repeat families prompted us to get a denser signal coverage by identifying parent enriched k-mers (Supplementary Fig. 8), which stem from all repeat types. The involved repeat types are mainly LTR-retrotransposons, which have accumulated to different copy numbers in the parents. Generally, the NAm-parent has a stronger sequence tag signal, which is in line with its higher LTR-retrotransposon activity (depicted in Fig 3 c). Regarding the mentioned centromeres: the NAm (and to a lesser extent also the Eu) k-mers are not enriched at the direct centromeres, but rather in the broader more repeat dense pericentromeric regions.

What is being measured in Fig S 9?

Response: We have expanded the figure legend for this supplementary figure. Please note, that this figure is now Supplementary Figure 11. The figure shows a depth of coverage analysis using genotyping-by-sequencing (GBS) reads.

In Fig 5d, terpene levels continue to increase toward the end of development, while terpene synthase expression peaks at weeks 3-4, then decreases, while terpene abundance is at its highest. This contrasts with humulone and lupulone, which reach a uniform level of abundance while gene expression decreases. Are there other genes involved in the DEG network that can help to explain this difference? Especially considering that DEG clusters 1 and 4 both increase toward the end of development.

Response: We thank the reviewer for this conceptual feedback, which helped us illustrate how to use our integrated transcriptomics and metabolomics resource to develop hypotheses for future

functional studies. We agree that differences between transcript and metabolite trajectories can offer important biological insights. Metabolite accumulation reflects the integrated effects of biosynthetic flux, storage, and turnover, and therefore a direct 1:1 correlation between transcript levels and compound abundance should not necessarily be expected though.

We examined the expression dynamics of the mevalonate (MVA) and methylerythritol phosphate (MEP) pathways (Supplementary Table 26), which generate the universal C₅ building blocks for bitter acids and terpenes. We found that genes of the cytosolic MVA pathway show stable expression throughout cone development, while genes of the plastidial MEP pathway display a transient rise-and-fall pattern, peaking mid-development and declining toward maturity. This divergence may contribute to the continued accumulation of sesquiterpenes (synthesized in the cytosol) such as α -humulene and β -caryophyllene in later stages, in contrast to the earlier stabilization of bitter acids (synthesized in the plastids).

Minor comments

The manuscript is well written but should be checked throughout for spelling (45 sDNA) and punctuation.

Response: We have also carefully checked the manuscript for spelling and punctuation errors, including the correction of “45 sDNA

“Genome-wide association scan (GWAS)” looks like it should be split into two sections starting at “Identification of k-mer based introgression”.

Response: In the revised manuscript, we have completely restructured the GWAS section to improve clarity and to incorporate suggestions from all reviewers. This includes a clearer separation of the GWAS analyses and the subsequent k-mer–based introgression identification.

The methods are thoroughly described. Some of the dense sections that describe identifying haplotypes and ancestry would potentially benefit from a visual graphic/schematic.

Response: In the revised manuscript, we added a schematic flow chart that illustrates the assembly workflow and explicitly indicates the steps at which haplotypes were identified, and ancestry was assigned (supplementary Figure 6). In addition, we included a plot of the depth of coverage analysis (Supplementary Figure 10) that was carried out to determine the origin of phases in cv. Apollo. We believe, these graphics provide a clearer overview of the procedure and complement the detailed description in the Methods.

Some of the work sounds like it involves custom scripts – is there a github page?

Response: We acknowledge the authors request for a deeper insight into our data processing workflows. We refer in detail to all the open source tools within the method section as used within our genomics, transcriptomics and metabolomics workflows. The metabolomics workflow does include custom scripts involving spectral data post processing and chemometrics. Thus, we created a working directory involving the used R script as well as input and output files generated by the script and uploaded it to the MASSIVE repository, as already described in the data availability statement. All files can be retrieved from the repository as a zipped archive “Apollo_cone_development_metabolomics.7z”.

The authors describe manual editing to remove allelic contigs (line 36). Does this approach account for paralogous vs. allelic regions?

Response: We only removed large inter-chromosomal allelic regions. Therefore, it's unlikely that the paralogous regions are removed during the process.

Reviewer #1 (Remarks to the Author):

The quality of the revised manuscript has been further improved, and my previous comments and suggestions have been addressed satisfactorily. I have no further comments or suggestions at this stage.

Response: We thank the reviewer for the positive evaluation of the revised manuscript and are pleased that our revisions have satisfactorily addressed all previous comments.

Reviewer #2 (Remarks to the Author):

This revision mostly addressed previous concerns in technical issues and improved readability by providing the supplementary notes and flowchart how to construct the genome data. I am convinced that this research provides useful information to accelerate hop breeding, which is widely beneficial for brewing industry.

However, it is not understandable why the authors did not compare previously reported genomes of the other cultivars, Saazer and Cascade in this study e.g., Fig.3a and Fig.4. Genomic comparison with other cultivars are likely informative for understanding of domestication of hop. Authors claim unavailability of the genome data reported by Akagi et al. (2025) , but they have been available since last summer on the following sites, aren't they?

Plant Garden:accession

<https://plantgarden.jp/en/list/t3486>

DDBJ:accession

PRJDB17943,PRJDB17944,PRJDB17945,PRJDB17941,
PRJDB17946,PRJDB17947,PRJDB17948,PRJDB17942,PRJDB18715,
SAM00766707, SAM00766708, SAM00814180, SAM00814181

Response: We thank the reviewer for highlighting the PlantGarden record and the corresponding DDBJ accessions for the Saazer genome reported by Akagi et al. (2025), and for emphasizing the value of including these resources in comparative analyses. The identifiers provided in the review were extremely helpful: using these specific accessions we were able to unambiguously trace and retrieve the relevant datasets and have now incorporated them into our revised comparative analyses.

We would also like to clarify our previous response regarding “unavailability.” Our earlier statement reflected the practical retrievability of the required files from the information we were able to trace at that time. In particular, the PlantGarden URL provided by the reviewer differs from the link we had attempted to follow based on

the paper's Data Availability section, and the corresponding database records appear to have been recently updated. We appreciate the reviewer's guidance and fully agree that these comparisons strengthen the manuscript.

Following the reviewer's suggestion, we updated Fig. 3A (gene collinearity/syntelog framework) to include both Saaz and Cascade alongside Apollo. Specifically, we implemented a syntelog-based comparison combining syntenic relationships and orthology across the four hop assemblies and Cannabis, which provides a gene-centric cross-reference that is robust to differences in assembly phasing and representation. To facilitate transparency and reproducibility, we also provide the full GeneSpace outputs (results tables), the original input files, and the associated R workspace/resource files in a public Zenodo repository, enabling reuse and extension of these analyses by the community.

The respective changes in the manuscript can be found in lines 336-340, 351-356, and 907-918.

Regarding the reviewer's point that such comparisons "could be informative for understanding hop domestication," we agree that genome-wide domestication analyses in hop are an exciting and important direction. However, we emphasize that a rigorous and meaningful domestication study would require substantially more genotypes and, critically, inclusion of wild and landrace accessions. The primary focus of our study is to provide a haplotype-resolved, accurate assembly of the cultivar Apollo; to deliver a detailed comparison of both Apollo haplotypes including their origin; and to complement this with functional analyses (expression and network analyses).

Finally, we did not update Fig. 4 to include Saaz and Cascade because Fig. 4 is based on whole-assembly alignment/structure-level comparisons, and directly comparing a phased, haplotype-resolved genome (Apollo) to collapsed/unphased assemblies can be misleading. In particular, the currently available Cascade assembly is not phased and appears substantially inflated in size ($\approx 30\%$), consistent with unresolved haplotypes and/or duplication artifacts; this can distort whole-genome alignment patterns and complicate interpretation in a figure intended to summarize structural correspondence. Instead, we addressed the reviewer's request through the gene-based comparative framework in Fig. 3A (and associated tables), which we believe is better suited to cross-assembly comparisons when phasing status differs.

Moreover, I am not sure that the word "Extensive" in the title is appropriate. Please reconsider the title represented by this comprehensive study.

Do authors mean "Chromosomal variation of a hybrid cultivar between North American and European hop"?

Response: We thank the reviewer for the suggestion regarding the title and we carefully reconsidered it. We prefer to retain the current title, as it reflects the central message and scope of the study. In this work, the two haplotypes of cv. Apollo capture markedly divergent North American and European hop lineages with pronounced genome-wide sequence and structural divergence. A key aim of the manuscript is to make this divergence explicit at the sequence level, because it is highly relevant for breeding outcomes.

We appreciate the reviewer's proposed alternative phrasing; however, we feel it would narrow the perceived scope to "chromosomal variation of a hybrid cultivar" and would not fully represent the broader resource and analyses presented (chromosome-scale, haplotype-resolved assembly and the dissection of lineage-level differences across the genome). We therefore keep the title unchanged.

Minor comments.

1. Unify the expression of similar words, high-value constituents (L108), high-value natural compounds (L645), and highly valuable metabolites (L650). I think the last one is better.

Response: We have made the requested changes.

2. Replace to "Thanks to" by "For" in L639.

Response: We have replaced "thanks to" with "through".

3. Unify the expression of similar words, specialized metabolites (L473) and secondary metabolic (L409, L419). I recommend the former.

Response: We have made the requested changes.

4. BUSCO (in L140) should be full spelled out at first.
Benchmarking Universal Single-Copy Orthologs (BUSCO)

Response: We have spelled out the term BUSCO

5. What does colour of the inner circles of Figure 5a,b, and c mean? Please show it clearly in the legend.

Response: We have both updated the labels in the figure and the figure legend.

Reviewer #3 (Remarks to the Author):

Reviewer #4 (Remarks to the Author):

The authors have addressed my comments. Very nice work.

Response: Thank you for your positive assessment. We appreciate your time and constructive feedback, and we are pleased that the revisions have addressed your comments.

Remaining minor points:

Check legibility of legends and labels in Supplementary Figures (e.g. Supplementary Figure 10 and aspect ratio in Supplementary Figure 20d).

Response: We have edited the species labels in Supplementary Figure 10 and corrected the aspect ratio in Supplementary Figure 20d.

Optional stylistic point - reverse complement sequences for chromosomes 1, 2, 3, 5, 8, and 9 from *H. lup* in Supplementary Figure 21.

Response: We have reverse complemented the sequences as requested and inserted an edited figure.

REVIEWERS' COMMENTS

Reviewer #2 (Remarks to the Author):

I have no further comment as this revision has cleared my concerns.
This is a good work.

Response: We thank Reviewer #2 for their positive assessment of the revised manuscript and for noting that their concerns have been satisfactorily addressed.